# A maternal brain hormone that builds bone

Muriel E. Babey[1,12], William C. Krause[2,12], Kun Chen[3], Candice B. Herber[2,9], Zsofia Torok[2], Joni Nikkanen[2,10], Ruben Rodriguez[2,11], Xiao Zhang[4], Fernanda Castro-Navarro[2], Yuting Wang[5], Erika E. Wheeler[3,6], Saul Villeda[7], J. Kent Leach[3,6], Nancy E. Lane[8], Erica L. Scheller[4], Charles K. F. Chan[5,13], Thomas H. Ambrosi[3✉] & Holly A. Ingraham[2✉]

In lactating mothers, the high calcium ($Ca^{2+}$) demand for milk production triggers significant bone loss[1]. Although oestrogen normally counteracts excessive bone resorption by promoting bone formation, this sex steroid drops precipitously during this postpartum period. Here we report that brain-derived cellular communication network factor 3 (CCN3) secreted from KISS1 neurons of the arcuate nucleus ($ARC^{KISS1}$) fills this void and functions as a potent osteoanabolic factor to build bone in lactating females. We began by showing that our previously reported female-specific, dense bone phenotype[2] originates from a humoral factor that promotes bone mass and acts on skeletal stem cells to increase their frequency and osteochondrogenic potential. This circulatory factor was then identified as CCN3, a brain-derived hormone from $ARC^{KISS1}$ neurons that is able to stimulate mouse and human skeletal stem cell activity, increase bone remodelling and accelerate fracture repair in young and old mice of both sexes. The role of CCN3 in normal female physiology was revealed after detecting a burst of CCN3 expression in $ARC^{KISS1}$ neurons coincident with lactation. After reducing CCN3 in $ARC^{KISS1}$ neurons, lactating mothers lost bone and failed to sustain their progeny when challenged with a low-calcium diet. Our findings establish CCN3 as a potentially new therapeutic osteoanabolic hormone for both sexes and define a new maternal brain hormone for ensuring species survival in mammals.

Osteoporosis significantly affects healthy ageing and is commonly experienced by more women than men. Females leverage oestradiol (E2) to increase energy expenditure[3] and preserve bone mass[4] as an anabolic hormone by regulating bone remodelling through osteocytes[5], osteoblasts[6] and osteochondral skeletal stem cells (ocSSCs)[7], which are fated for bone and cartilage[8,9]. For women, oestrogen depletion following menopause or anti-hormone therapies degrades bone mass, an effect that underscores the anabolic features of oestrogen on bone. However, the intimate association between oestrogen and bone is uncoupled during lactation when the E2 surge in late-stage pregnancy drops precipitously. Moreover, bone remodelling increases sharply in rodents[10] and in primates[11,12] to meet the high calcium demand by progeny[13]. Parathyroid hormone-related protein (PTHrP), a close orthologue of parathyroid hormone (PTH) from mammary glands, is the main driver for stripping calcium from maternal bones for milk production[14,15]. The continuous demand for calcium by newborns eventually leads to significant bone loss in mothers, dropping nearly 30% in rodents owing to large litter sizes[10] and 10% in humans[12,16,17]; these losses mostly normalize after lactation[10,17]. Presumably, the maternal skeleton (and that of pups) would be severely compromised without a concomitant lactational anabolic phase, as inferred by the increased bone mass in lactating mothers after conditional knockout of PTHrP[18,19]. This raises the possibility that dedicated mechanisms drive the anabolic pathway of bone remodelling during lactation.

In addition to the direct actions of E2 on bone, we and others have shown that central oestrogen signalling exerts a sex-dependent restraint on bone formation, alongside its role in promoting thermogenesis and spontaneous activity[20–23]. Females exhibit high bone mass following the deletion of oestrogen receptor-α (ERα) in the ARC of the medial basal hypothalamus[2,24]. Eliminating ERα in $ARC^{KISS1}$ neurons, which regulate metabolism and reproduction[25], confirmed the central origins of this skeletal phenotype independent of high E2 levels[2].

Here using question-driven and discovery-based approaches, we set out to identify an osteoanabolic hormone in mutant female mice after first showing that this factor circulates in the blood. CCN3 (also known as NOV) emerged as the top candidate, meeting all predicted benchmarks. That is, it is secreted, its appearance in the ARC coincides with the onset and loss of the bone phenotype and it enhances bone formation and fracture repair. The relevance of brain-derived CCN3 in female physiology was revealed after demonstrating its essential role in lactating mothers.

[1]Department of Medicine, Division of Endocrinology and Metabolism, University of California, San Francisco, San Francisco, CA, USA. [2]Department of Cellular and Molecular Pharmacology, University of California, San Francisco, San Francisco, CA, USA. [3]Department of Orthopaedic Surgery, University of California, Davis, Sacramento, CA, USA. [4]Department of Medicine, Washington University, St Louis, MO, USA. [5]Institute for Stem Cell Biology and Regenerative Medicine and Department of Surgery, Stanford University School of Medicine, Stanford, CA, USA. [6]Department of Biomedical Engineering, University of California, Davis, Davis, CA, USA. [7]Department of Anatomy, University of California, San Francisco, San Francisco, CA, USA. [8]Department of Medicine, Division of Rheumatology, University of California, Davis, Sacramento, CA, USA. [9]Present address: Denali Therapeutics, South San Francisco, CA, USA. [10]Present address: Department of Integrative Biology, University of California, Berkeley, Berkeley, CA, USA. [11]Present address: Carmot Therapeutics, Berkeley, CA, USA. [12]These authors contributed equally: Muriel E. Babey, William C. Krause. [13]Deceased: Charles K. F. Chan. ✉e-mail: thambrosi@ucdavis.edu; holly.ingraham@ucsf.edu

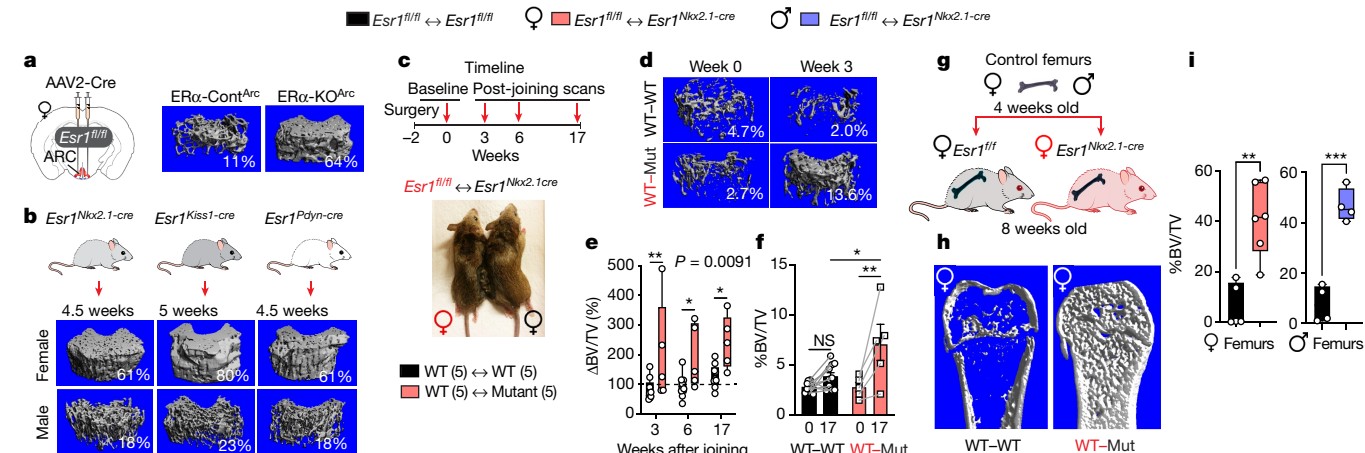

Fig. 1 | A brain-dependent circulatory factor builds bones in mice. a,b, Viral and genetic mouse models. a, Left, schematic of the stereotaxic deletion of ERα in the ARC using an AAV2-Cre vector. Right, representative μCT scans of the distal femur from females injected with AAV2 control virus (ERα-Cont^Arc) and of a ERα-knockout^Arc (ERα-KO^Arc) female, as previously reported[2]. %BV/TV values are indicated. b, μCT images obtained from *Esr1^Nkx2.1-cre*, *Esr1^Kiss-cre* and *Esr1^Prodynorphin-cre* (*Esr1^Pdyn-cre*) 4.5–5-week-old mice with %BV/TV values indicated in the lower right-hand corner. c–f, Parabiosis models. c, Timeline of in vivo μCT imaging after surgical pairing of *Esr1^fl/fl* (WT) and *Esr1^Nkx2.1-cre* (mutant) female mice. d, Representative in vivo μCT imaging of distal femurs at baseline (week 0) and 3 weeks later (week 3) with %BV/TV values indicated. e, Per cent change in %BV/TV at weeks 3, 6 and 17 compared with week 0 (biological sample sizes used (N) = 5). f, Absolute %BV/TV values for *Esr1^fl/fl* females in WT–WT and WT–Mut pairs, showing values for each animal at baseline (0) and 17 weeks later (17). g–i, Bone transplants. g, Schematic of WT female and male femur transplants into *Esr1^fl/fl* (WT) and *Esr1^Nkx2.1-cre* (mutant) female mice for a duration of 6 weeks. h, Representative μCT images of control femurs transplanted WT–WT and WT–Mut females. i, Fractional BV (%BV/TV) of excised control femurs transplanted into *Esr1^fl/fl* females (black bars) or female (red bar) or male (blue bar) bones transplanted into *Esr1^Nkx2.1-cre* females (N = 4–6). Two-way analysis of variance (ANOVA) in e with repeated measures (Šidák's multiple-comparisons test). One-way ANOVA in f (Tukey's multiple-comparisons test). Unpaired Student's *t*-test, two-tailed in i. *P < 0.05, **P < 0.01, ***P < 0.001, NS, not significant. Error bars ± s.e.m. Graphic in b was adapted from BioRender (https://www.biorender.com). Graphic in g was adapted from Mind the Graph (https://mindthegraph.com) under a Creative Commons licence CC BY-SA 4.0.

## A brain-derived humoral factor builds bone

Our previous viral-mediated and genetic deletions of ERα in the ARC found that a subset of KISS1 neurons regulates bone mass and bone strength in females but not males[2] (Fig. 1a,b). That KISS1 neurons participate in this brain–bone axis was further supported after deleting ERα with the *Prodynorphin-Cre* driver (Fig. 1b and Extended Data Fig. 1a,b). To identify the molecular origins of the high bone mass phenotype, we relied exclusively on the *Esr1^Nkx2.1-cre* female mouse model, which exhibits this unusual phenotype by 4 weeks of age (Extended Data Fig. 1c–e). Given the privileged position of the ARC as a circumventricular organ of the brain that lies dorsal to the median eminence, we asked whether a humoral factor accounts for this female-specific high bone mass.

Using classical parabiosis coupled with in vivo micro-computed tomography (μCT) imaging, female mice were surgically joined to generate either control–control wild-type (WT–WT) pairs or control–*Esr1^Nkx2.1-cre* (WT–Mut) pairs (Fig. 1c and Extended Data Fig. 2a). Shortly after surgery (2 weeks), baseline bone parameters on the contralateral femur opposite the surgical side were established for each paired animal. Females in the WT–WT pairing exhibited a net decrease in bone mass that was readily observed beginning at 3 weeks after baseline (Fig. 1d,e). This reduction normalized by week 17, increasing by an average of about 37%. In the WT–Mut pairings, higher fractional bone volume (per cent bone volume/total volume (%BV/TV)) was observed at all time points in control females, increasing by about 152% by 17 weeks. Mutant females also regained significant bone mass with pairing (Fig. 1f and Extended Data Fig. 2b–d). Other gross parameters were unchanged in WT–Mut pairings, including uterine weights, a result consistent with the notion that increased oestrogen levels are not involved in generating the bone phenotype in mutant females (Extended Data Fig. 2e).

To confirm that a humoral factor accounts for the high bone mass in mutant females, femurs of both sexes from 4-week-old control donors were subcutaneously implanted into 8-week-old control and mutant females (Fig. 1g and Extended Data Fig. 3a). Significant increases in fractional BV (about sixfold) were detected in mutant females 6 weeks after implantation with femurs from female and male mice (Fig. 1h,i and Extended Data Fig. 3b,c). This result demonstrates that although the origins of this brain-dependent osteoanabolic hormone are female-specific, it functions in male and females.

## Skeletal stem cells confer high bone mass

Bone homeostasis is tightly regulated by skeletal stem cell (SSC)-based bone formation and osteoclast-based bone resorption. It has been demonstrated that stem cells with distinct lineage hierarchies facilitate new bone formation[26]. In particular, ocSSCs form bone and cartilage and are present in the growth plate and periosteum of bones[8,9,27]. Conversely, perivascular SSCs (pvSSCs) give rise to unilateral committed adipogenic progenitor cells (APCs) that generate all bone marrow adipose tissue (BMAT)[26,28] (Fig. 2a). Given the increased bone formation in mutant females[2], we reasoned that the brain-dependent osteoanabolic hormone might enhance ocSSC activity. OcSSCs from female WT mice were isolated by flow cytometry and transplanted beneath the renal capsule of control and mutant females (Fig. 2b). WT ocSSCs transplanted into control *Esr1^fl/fl* female littermates formed an ectopic bone graft with a host-derived haematopoietic compartment over 6 weeks (Fig. 2c). By contrast, when grafted into *Esr1^Nkx2.1-cre* females, significantly higher mineralization with sparse haematopoietic marrow was detected (Fig. 2c–e), which suggested that the osteoanabolic hormone present in mutant females alters the ocSSC lineage to promote bone formation. Consistent with this hypothesis, the higher bone mass observed in WT–Mut parabionts or WT bone transplants correlated with increased ocSSC frequency (Extended Data Fig. 4a). The potency of this circulatory bone-building hormone was further verified by stereotaxic delivery of prospectively isolated GFP-positive WT ocSSCs to the vicinity of the ARC (Fig. 2f). Notably, μCT imaging of mutant *Esr1^Nkx2.1-cre* hypothalami revealed mineralized ossicles overlapping with transplanted GFP-positive cells 6 weeks after injection, whereas

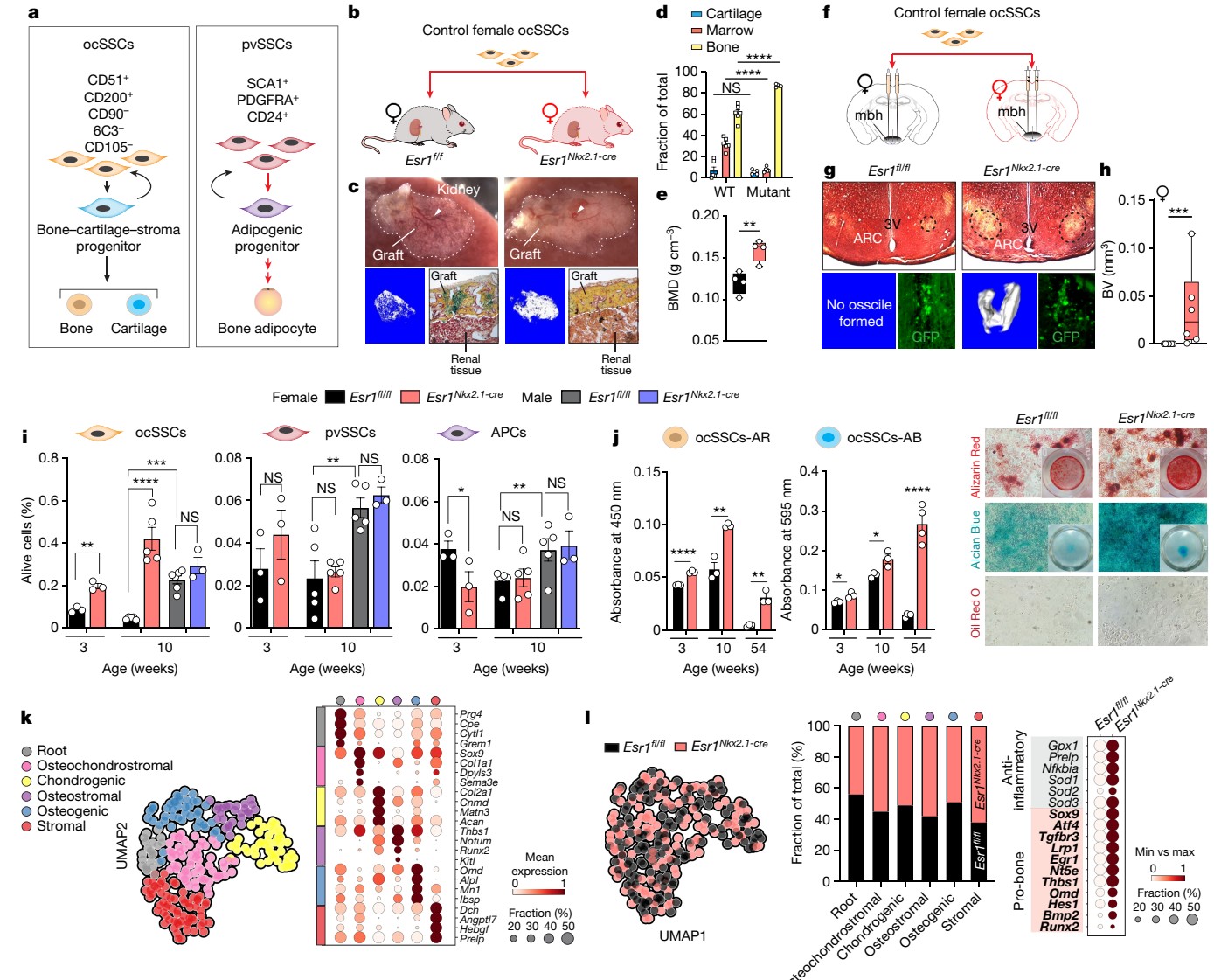

**Fig. 2 | A brain-dependent circulatory factor increases the osteogenic capacity of ocSSCs. a**, Schematic of fluorescence-activated cell sorting (FACS) isolation from non-haematopoietic, non-endothelial cell fraction and the fate of ocSSCs (left) and pvSSCs (right). **b**–**e**, SSC kidney capsule transplants. **b**, Schematic of WT female ocSSC kidney capsule transplants into *Esr1^fl/fl* and *Esr1^Nkx2.1-cre* females. **c**, Representative images of the graft region with host-derived haematopoiesis (top, white arrowheads), BV and Movat's pentachrome staining of bone (yellow), cartilage (blue) and marrow (red). **d**,**e**, Fractional areas from stained kidney graft sections (*N* = 6, 5) (**d**) and bone mineral density (BMD) from grafts by μCT (*N* = 4, 4; *Esr1^fl/fl* (black) and *Esr1^Nkx2.1-cre* (red) females) (**e**). **f**–**h**, SSC transplants into the medial basal hypothalamus (mbh). **f**, Schematic of stereotaxic bilateral delivery of control ocSSCs (about 550 live cells) from *Esr1^fl/fl-CAG-Luc-GFP* into the mbh of *Esr1^fl/fl* and *Esr1^Nkx2.1-cre* females (*N* = 7, 6). **g**, Representative images of pentachrome-stained brain sections (top) with ossicles (lower left) and anti-GFP (lower right) 6 weeks after injection. 3V, third ventricle. **h**, BV in the mbh of *Esr1^fl/fl* (black) and *Esr1^Nkx2.1-cre* (red) females. **i**, Per cent ocSSCs, pvSSCs and APCs (Methods) in femurs of *Esr1^fl/fl* and *Esr1^Nkx2.1-cre* 3-week-old (*N* = 3, 3) and 10-week-old females (*N* = 5, 5) and males (*N* = 5, 3). **j**, Quantification (left) of Alizarin Red (AR; osteogenesis) or Alcian Blue (AB; chondrogenesis) staining of differentiated ocSSCs from 3-week-old, 10-week-old and 54-week-old females with representative images (right) including Oil Red O (adipogenesis) staining (technical replicates in cell culture assays (*n*) = 3–4 per group). **k**, Uniform manifold approximation and projection (UMAP) plot (Leiden clustering) of Smart-Seq2 scRNA-seq data of high-quality filtered single ocSSCs from 7-week-old females (left) with dot plot of cluster-specific markers (right). **l**, UMAP (left), distribution of genotypes within cluster populations (middle) and dot plot of anti-inflammatory and pro-osteogenic markers (right) of *Esr1^fl/fl* and *Esr1^Nkx2.1-cre* ocSSCs. One-way ANOVA in **d** and **i** (Tukey's multiple-comparisons test). Mann–Whitney test, two-tailed in **h**. Unpaired Student's *t*-test, two-tailed for **e**, **i** (for the 3-week time point) and **j**. *$P < 0.05$, **$P < 0.01$, ***$P < 0.001$, ****$P < 0.0001$. Error bars ± s.e.m. Graphic in **b** (kidney) was reproduced from BioRender (https://www.biorender.com). Graphic in **b** (mouse) was adapted from Mind the Graph (https://mindthegraph.com) under a Creative Commons licence CC BY-SA 4.0.

no ossicles were detected in WT brains (Fig. 2g,h). These data provide further support for the existence of a circulatory anabolic bone factor in mutant females, possibly originating from the ARC.

That a circulating factor affects WT ocSSC activity motivated us to compare the differentiation capacity of mutant and control ocSSCs. Flow cytometry analyses revealed a sex-dependent increased frequency of ocSSCs in pre-pubertal (3 weeks old) and young adult mutant (10 weeks old) females (Fig. 2i). This alteration was limited to ocSSCs, as pvSSCs and their progeny (APCs) fated for BMAT[26,28] were equivalent in controls and mutants, although APCs were reduced in younger mutants (Fig. 2i). Functional assays revealed that ocSSCs from both genotypes showed no differences in colony-forming ability (CFU-F) at 10 weeks of age (Extended Data Fig. 4b), whereas mutant ocSSCs exhibited a higher intrinsic potential for bone and cartilage formation, even

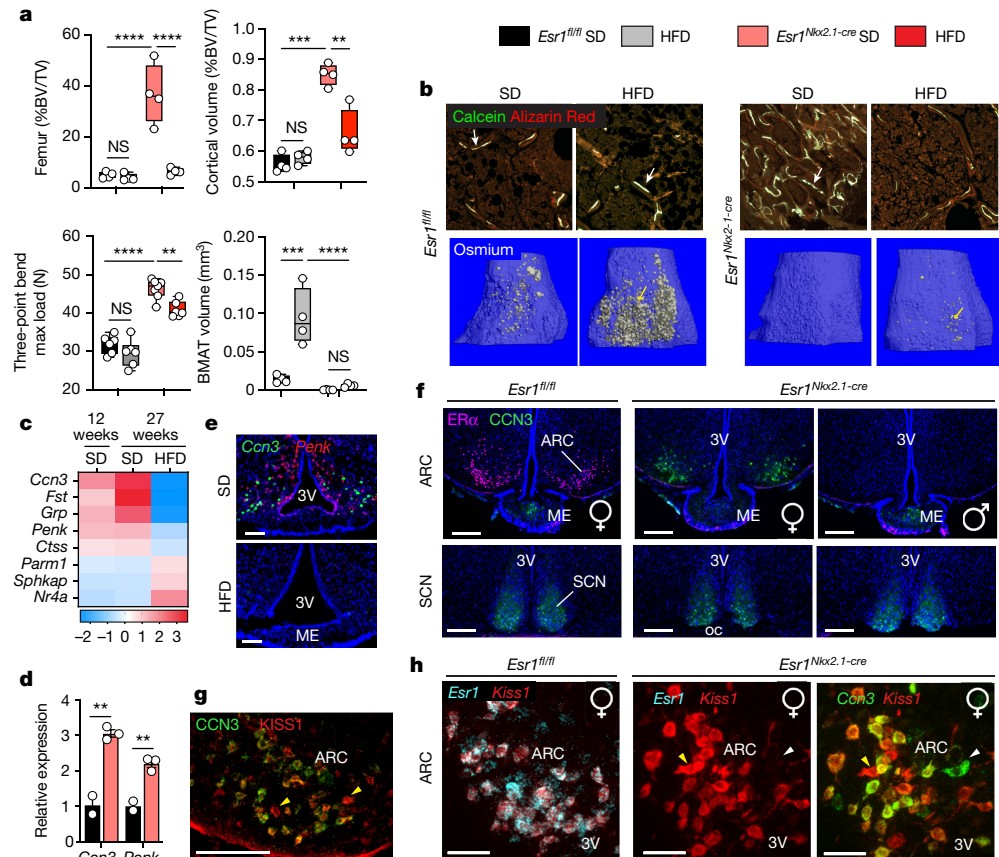

**Fig. 3 | Identification of CCN3 as a brain-derived osteoanabolic factor.**
**a**, Trabecular and cortical fractional BV, mechanical strength (three-point bend) and BMAT levels in long bones of *Esr1fl/fl* and *Esr1Nkx2.1-cre* females fed standard diet (SD) or HFD for 17 weeks (*N* = 4–6 per group). **b**, Representative images of tibia from female mice (aged 27 weeks) fed SD or HFD for 17 weeks labelled for calcein and Alizarin Red (top, white arrows and magnified from Extended Data Fig. 6c) and osmium stained with lipid droplets (bottom, yellow arrows). **c**, Heatmap of top DEGs changed in the ARC of *Esr1Nkx2.1-cre* females at 12 weeks of age (adapted from ref. 2) and at 27 weeks of age fed SD or HFD. Scale is log₂ fold change. **d**, Transcript levels of *Ccn3* and *Penk* in the ARC of 3.5-week-old mutant females, measured by quantitative PCR (qPCR). *N* = 2–3 per group. **e**, *Ccn3* and *Penk* expression by RNAscope of the ARC in mutant female *Esr1Nkx2.1cre* mice fed either SD or HFD. Scale bar, 100 μm. ME, median eminence. **f**, Staining for ERα (pink) and CCN3 (green) in brain sections from posterior ARC and SCN regions of *Esr1fl/fl* female (10-week-old) and *Esr1Nkx2.1-cre* female and male (12-week-old) mice. Scale bar, 200 μm. oc, optic chiasm. **g**, CCN3 and KISS1 overlapping expression in *Esr1Nkx2.1-cre* female medial basal posterior ARC (yellow arrowheads). Scale bar, 100 μm. **h**, *Ccn3* (green) *Esr1* (cyan) and *Kiss1* (red) transcripts from posterior ARC brain sections in control and mutant *Esr1Nkx2.1-cre* females (*Kiss1* only, yellow arrowheads; *Ccn3* only, white arrowheads). Scale bar, 50 μm. One-way ANOVA in **a** (Tukey's and Šidák's multiple-comparisons test). Unpaired Student's *t*-test, two-tailed for **d**. \*\**P* < 0.01, \*\*\**P* < 0.001, \*\*\*\**P* < 0.0001. Error bars ± s.e.m.

when collected from aged (54 weeks old) females (Fig. 2j and Extended Data Fig. 4c). These data are consistent with the attenuated bone loss observed in aged mutant females (Extended Data Fig. 4d,e) and the known linkage between ocSSC dysfunction and age-related bone loss[29].

## CCN3 is a brain-derived bone factor

Based on the expansion and enhanced osteogenic capacity of mutant ocSSCs, Smart-Seq2 single-cell RNA sequencing (scRNA-seq) data of freshly sorted cells were used to identify downstream signalling components of the osteoanabolic factor in mutant females. Although the differentiation dynamics of mutant ocSSCs were primed towards bone formation, only modest transcriptomic differences were detected between samples from control and mutant mice. These results provided few clues regarding the identity of the osteoanabolic hormone in *Esr1Nkx2.1-cre* female mice (Fig. 2k,l and Extended Data Fig. 5). In a second approach, we asked whether a chronic high-fat diet (HFD) challenge, which is known to affect ARC^KISS1 neurons[30], might disrupt the production of this brain-derived bone-building factor. Notably, the bone phenotype in *Esr1Nkx2.1-cre* females was reversed with the HFD while leaving body weights, fat mass, blood triglycerides, glucose homeostasis and

bone resorption unchanged (Fig. 3a,b and Extended Data Fig. 6a–d). Bone parameters in *Esr1Nkx2.1-cre* males were unaffected (Extended Data Fig. 6e,f). This HFD-induced bone loss in mutant female mice was specific and could not be recapitulated by chronic hyperglycaemia following treatment with the insulin receptor antagonist S961 (ref. 31) (Extended Data Fig. 6g). Notably, although dense bones in *Esr1Nkx2.1-cre* mutant female mice readily degraded with the HFD, they remained strong and resisted fat accumulation (Fig. 3a,b), as quantified by osmium staining[32], thus defying the anticipated coupling between BMAT expansion and bone loss[33].

Gene changes associated with the dietary-induced loss of bone mass in *Esr1Nkx2.1-cre* female mice were captured by profiling microdissected ARC and other tissues. Bulk RNA-seq of the ARC revealed a small set of upregulated differentially expressed genes (DEGs) encoding neuropeptides or secreted proteins, including *Ccn3*, *Fst*, *Grp* and *Penk*, which were significantly reduced after HFD feeding (Fig. 3c and Extended Data Fig. 7a,b). Of these DEGs, only *Ccn3* and *Penk* were detected in the ARC at 3.5–4 weeks of age, before the appearance of the bone phenotype (Fig. 3d and Extended Data Fig. 7c,d), with both disappearing following HFD feeding (Fig. 3e). CCN3 was detected in mutant female mice in close proximity to the third ventricle and was absent in intact female

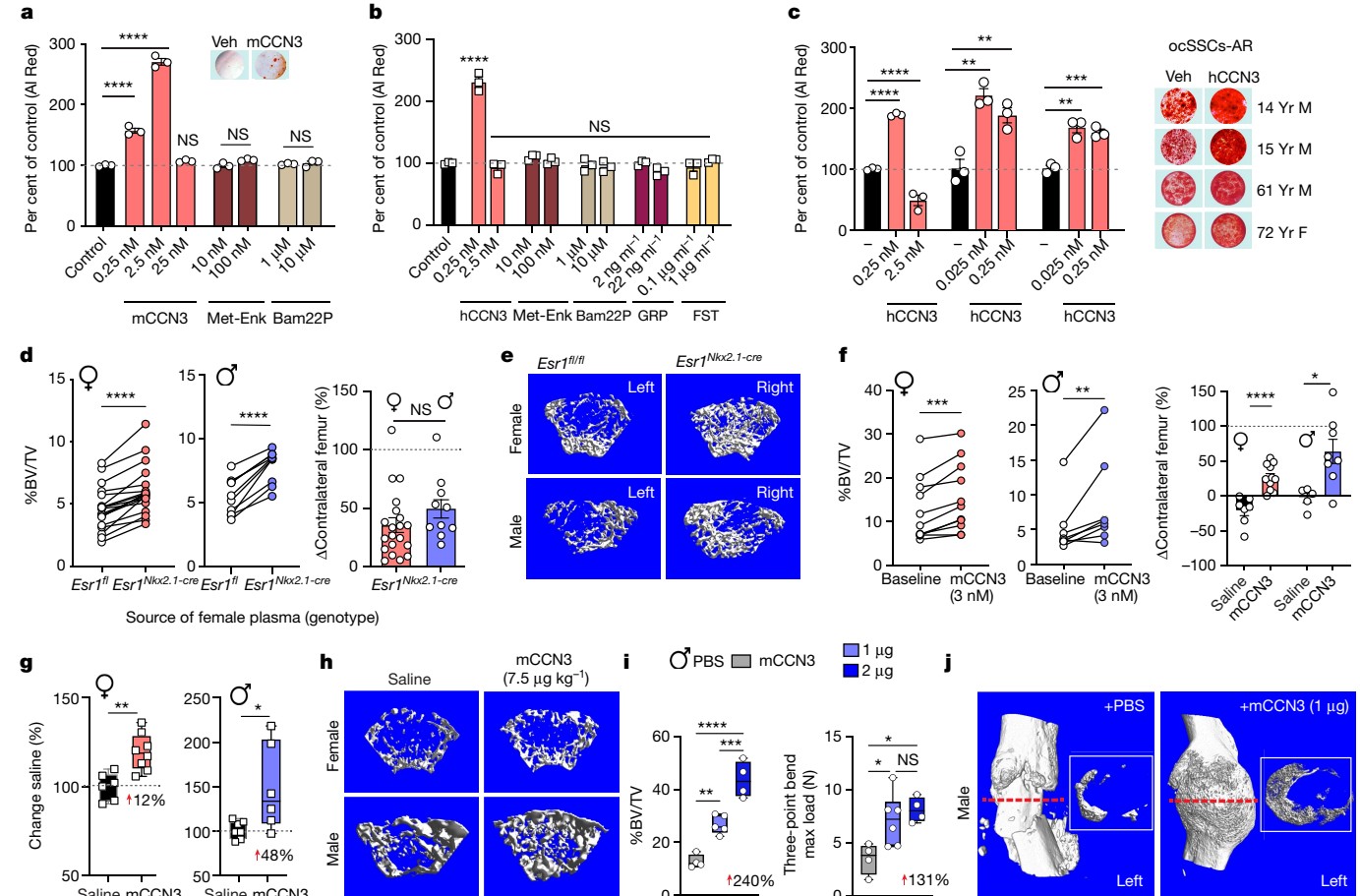

**Fig. 4 | CCN3 enhances osteogenesis, bone formation and fracture repair.**
**a**–**c**, Osteogenic differentiation assays (14 days). **a**, Differentiation of mouse ocSSCs (from 2-week-old male and female mice) treated with mCCN3, met-ENK and Bam22P. Inset, cells stained with Alizarin Red (Al Red) (*n* = 3). Veh, vehicle. **b**,**c**, Human ocSSCs treated with human CCN3 (hCCN3) during in vitro osteogenesis. **b**, Human ocSSCs from a 14-year-old male treated with hCCN3, met-ENK, Bam22P, gastric-related peptide (GRP) and follistatin (FST). **c**, Additional ocSCCs from 15-year-old (left), 72-year-old (middle) and 61-year-old (right) patients treated with hCCN3 (red bars) (*n* = 3), with representative images of wells stained with Alizarin Red to the right of the graph. F, female; M, male. **d**–**f**, Whole femur bone cultures treated with plasma or mCCN3 daily for 5 days. **d**, Left, %BV/TV for control *Esr1^{fl/fl}* female (red) and male (blue) 6–8-week-old femurs treated with plasma from *Esr1^{Nkx2.1-cre}* females; contralateral femurs treated with plasma from *Esr1^{fl/fl}* mice (*N* = 19, 10). Right, per cent change in %BV/TV of contralateral female (red bar) or male (blue bar) femurs. **e**, Representative μCT images from treated femurs. **f**, Left, %BV/TV control

female (*N* = 11) and male (*N* = 8) 10–11-week-old femurs treated daily for 5 days with mCCN3 (3 nM) compared with untreated baseline contralateral femur control. Right, per cent change in female (red bar) or male (blue bar) %BV/TV treated with CCN3 or saline normalized to baseline (*N* = 5–11). **g**, Per cent change in %BV/TV in *Esr1^{fl/fl}* mice following daily CCN3 injections (i.p. 7.5 μg kg⁻¹) or saline for 21 days, normalized to mean %BV/TV of saline controls (*N* = 6, 8 females and 7, 6 males). **h**, Representative μCT images from treated femurs. **i**, %BV/TV (left) and mechanical strength of callus (right) 21 days after fracture in aged male mice after slow-release mCCN3 (1 or 2 μg) treatment (phosphate-buffered saline (PBS) *N* = 4; 1 μg *N* = 5, 6; and 2 μg *N* = 4). **j**, Representative μCT images and cross-sections of callus from 24-month-old C57BL/6 male femurs. One-way ANOVA in **a**–**c** and **i** (Dunnett's (**a**, **b**, **i**) and Tukey's (**c**) multiple-comparisons test). Paired Student's *t*-test, one-tailed for left panels in **d**, **f**, and unpaired Student's *t*-test, two-tailed for right panels in **d**, **f** and **g**. *P < 0.05, **P < 0.01, ***P < 0.001, ****P < 0.0001. Error bars ± s.e.m.

and male mice and in mutant male mice, which is in contrast to the constitutive expression of CCN3 in the suprachiasmatic nucleus (SCN)[34] (Fig. 3f and Extended Data Fig. 7e). No viable candidates emerged after profiling the pituitary and liver, two common tissue sources of secreted proteins (Extended Data Fig. 7a,b). As expected from our previous genetic models[2], CCN3 colocalized with nearly all KISS1 neurons in the mutant female mice (Fig. 3g,h).

## CCN3 is an osteoanabolic hormone in mice

High expression of CCN3 in mutant ARC neurons that are positive for KISS1 and negative for ERα prompted us to test this founding member of the CCN family[35] as an osteoanabolic factor. This secreted protein is postulated to antagonize CCN2 to inhibit osteogenesis[35,36,37], although a single report suggests the opposite[38]. Culturing primary ocSSCs

isolated from postnatal day 14 WT mice with mouse CCN3 (mCCN3) increased mineralization by around 200% (Fig. 4a). Two major peptides encoded by *Penk*, met-ENK and BAM-22P, failed to induce any changes. Similarly, the high bone mass in mutant female mice resisted chronic treatment with the μ-opioid receptor antagonist naloxone (Extended Data Fig. 8a). Using primary human ocSSCs from pubertal and older individuals, the osteogenic effects of CCN3 were readily observed in both male and female ocSSCs (Fig. 4b,c and Extended Data Fig. 8b).

We then evaluated the anabolic potential of CCN3 using ex vivo whole long-bone cultures. This was done because we initially found that bone mass of freshly dissected control femurs was increased when treated with plasma collected from *Esr1^{Nkx2.1-cre}* females compared with plasma from control mice (Fig. 4d,e and Extended Data Fig. 8c,d). Using this simple but effective assay, low doses of mCCN3 (3 nM) induced an upwards dynamic shift (around 50–60%) in bone mass in young and

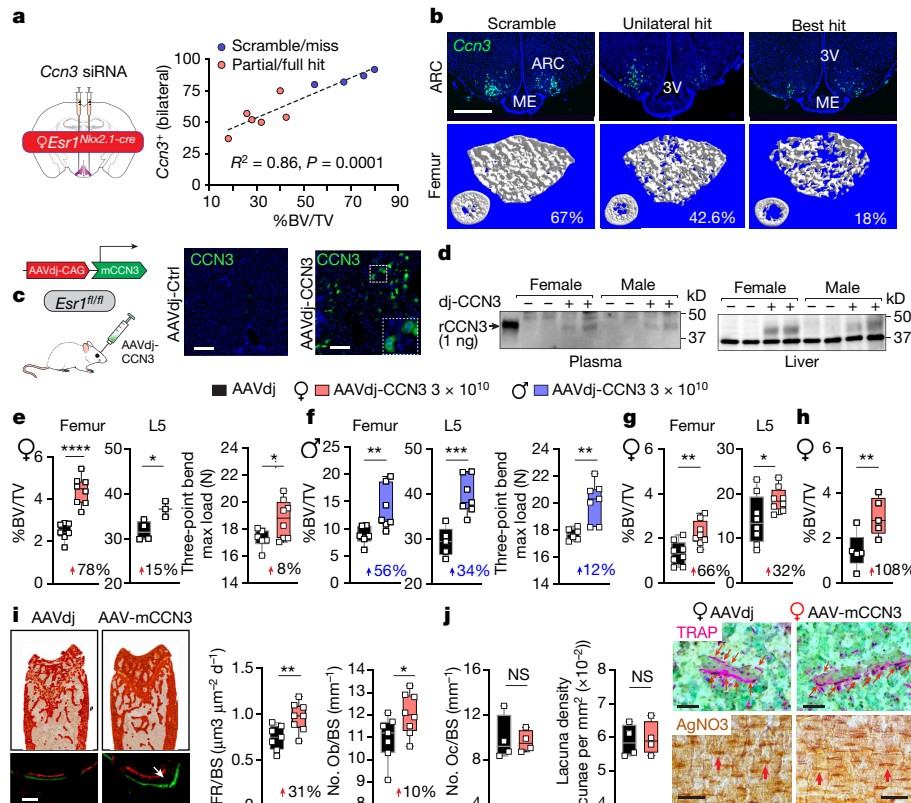

**Fig. 5 | Exogenous CCN3 drives higher mass, strength and formation of bone in vivo. a**, Left, schematic of experiment to induce loss-of-function of CCN3 (*Ccn3* knockdown) in the ARC in mice. Right, *Ccn3*-positive neurons in female *Esr1^Nkx2.1-cre* ARC versus %BV/TV after *Ccn3* siRNA injections. **b**, *Ccn3* expression in control, unilateral and bilateral hit with corresponding μCT scans of distal femurs. Scale bar, 500 μm (*N* = 6, 4). **c**, Left, schematic of experiment to induce gain-of-function CCN3 in the liver. Right, ectopic mCCN3 expression in *Esr1^fl/fl* female hepatocytes following retro-orbital injection of AAVdj-CAG-CCN3 (AAVdj-CCN3) or control (AAVdj-Ctrl) vectors. Inset shows double nuclei. Scale bar, 100 μm. **d**, mCCN3 immunoblot of heparin–agarose-purified plasma (left) and liver extracts (right, 10 μg total protein) from mice 5 weeks after injection with AAVdj-Ctrl (−) or AAVdj-mCCN3 (+). Recombinant mCCN3 (rCCN3) shown in far left lane. **e**, %BV/TV (left) of femurs and L5 and mechanical strength (right) of femurs from 3-4-month-old *Esr1^fl/fl* female mice 5 weeks after injection (*N* = 7, 8 femurs and *N* = 4, 3 L5). **f**, %BV/TV (left) of femurs and L5 and mechanical strength (right) of femurs from 3–4-month-old *Esr1^fl/fl* males 5-weeks after injection (*N* = 6, 7 femurs and *N* = 5, 7 L5). **g**, %BV/TV of femurs and L5 in 5-month-old OVX *Esr1^fl/fl* females 9 weeks after injection (*N* = 8, 8 femurs and *N* = 8, 8 L5). **h**, %BV/TV of 20–23-month-old *Esr1^fl/fl* female femurs 9 weeks after injection (*N* = 5, 5). **i**, Bone formation rate and bone surface (BFR/BS), and number of osteoblasts per bone surface (No. Ob/BS) determined by histomorphometry (*N* = 7, 8). Scale bar, 50 μm. **j**, Number of osteoclasts per bone surface (No. Oc/BS) and lacunar density per bone area as determined by static histomorphometry (*N* = 4, 4). Representative images of TRAP (top) and silver nitrate (AgNO3, bottom) staining for femoral osteoclasts and lacunae, respectively. Scale bar, 50 μm. Simple linear regression for **a**. Unpaired Student's *t*-test, one-tailed or two-tailed for **e**–**g**, **i** and **j** as indicated in Supplementary Table 3. Mann–Whitney test, one-tailed for **h**. *\**P* < 0.05, \*\**P* < 0.01, \*\*\**P* < 0.001, \*\*\*\**P* < 0.0001. Error bars ± s.e.m. Graphic in **c** was adapted from BioRender (https://www.biorender.com).

aged femurs compared with a significant degradation with saline (Fig. 4f and Extended Data Fig. 8e,f). Adult WT mice injected with mCCN3 (7.5 μg kg⁻¹, intraperitoneally (i.p.)) or saline daily over 3 weeks also showed a significant per cent increase in bone mass when treated with mCCN3 (Fig. 4g,h and Extended Data Fig. 8g). Notably, in a stabilized fracture model carried out in 2-year-old male mice, callus BV and strength exhibited dose-dependent increases (Fig. 4i,j and Extended Data Fig. 8h). These findings highlight the utility of mCCN3 to accelerate and improve fracture repair, and when taken together with other assays, confirm that CCN3 drives osteogenesis in human SSCs and higher bone mass in mice.

To establish a link between brain CCN3 and increased bone mass in mutant female mice, we examined how transient knockdown of *Ccn3* in ARC neurons of *Esr1^Nkx2-1* females would affect bone mass. The degree of *Ccn3* suppression induced by short interfering RNA (siRNA) tracked well with the fractional BV (%BV/TV) (Fig. 5a,b), thereby establishing a reliance on brain CCN3. We then leveraged the secretory capacity of hepatocytes to ectopically increase circulating CCN3 in control *Esr1^fl/fl* females through systemic delivery of AAV-dj-CCN3. CCN3 expression in hepatocytes was detected as early as 2 weeks after injection and increased

in a dose-dependent manner (Fig. 5c and Extended Data Fig. 9a,b). Plasma CCN3 was detected only after heparin–agarose purification from the highest levels of hepatic CCN3 expression (15 × 10¹⁰ genome copies (GC) per mouse), a result that underscores the poor specificity of existing anti-mCCN3 antibodies (Fig. 5d and Extended Data Fig. 9c). Nonetheless, at these and lower levels (3 × 10¹⁰ GC), CCN3 increased bone mass and bone strength in intact adult mice of both sexes (Fig. 5e,f and Extended Data Fig. 9e). The potency of CCN3 as an osteoanabolic hormone was further established by the 1.5-fold and 2-fold increase in bone mass after expressing CCN3 in ovariectomized (OVX) mice and in an older group of *Esr1^fl/fl* females, respectively (Fig. 5g,h). Even at exceedingly low levels of hepatic CCN3 expression, modest increases in bone formation were observed in young mice, as also observed with higher doses in aged females (Extended Data Fig. 9d,e). CCN3-induced bone formation did not lead to compensatory upregulation in osteoclast number or osteocyte number, as reflected by tartrate-resistant acid phosphatase (TRAP)-positive staining cells and lacunae density values, respectively (Fig. 5i,j and Extended Data Fig. 9f–i). This result implies that CCN3 not only increases bone mass but also promotes healthy bone remodelling in both sexes.

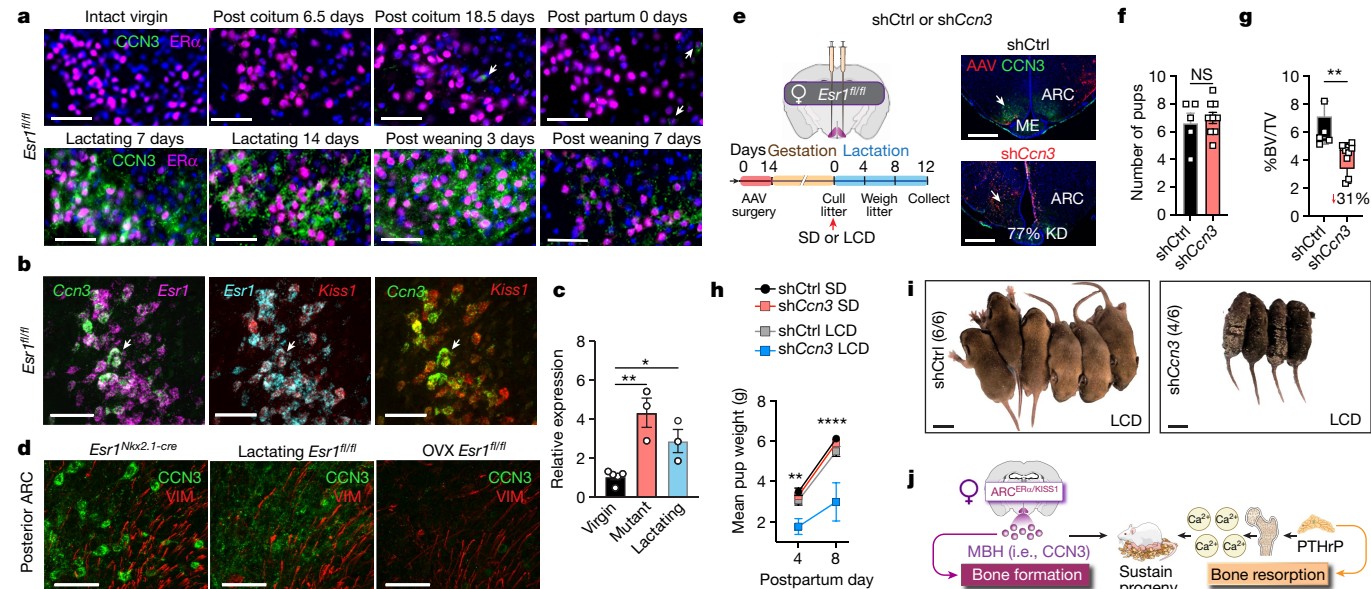

**Fig. 6 | Lactating females require maternal brain CCN3 to sustain progeny.**
**a**, Representative images of coronal brain sections from *Esr1^fl/fl* females stained for ERα (magenta) and CCN3 (green) in the posterior medial basal ARC during pregnancy and postpartum stages (lactation and post weaning) (N ≥ 2 for each time point). Scale bar, 50 μm. **b**, Colocalization of *Ccn3* (green), *Kiss1* (red) and *Esr1* (magenta and cyan) transcripts in the ARC of a lactating control (*Esr1^fl/fl*) female at 7 days. Scale bar, 50 μm. **c**, Relative *Ccn3* quantified from microdissected ARC tissue obtained from *Esr1^fl/fl* virgin, *Esr1^Nkx2.1-cre* mutant virgin and *Esr1^fl/fl* lactating (7 DPP) female mice (N = 5, 3, 3). **d**, CCN3 (green) and vimentin (red, VIM) in the posterior ARC of virgin *Esr1^Nkx2.1-cre* mutant and lactating or OVX *Esr1^fl/fl* control mice (1 week after surgery). **e–i**, *Ccn3* knockdown (KD) in the ARC of lactating female mice. **e**, Schematic of injection of shRNA *Ccn3* (sh*Ccn3*) or shRNA control (shCtrl) vectors into the ARC of control *Esr1^fl/fl* females with experimental timeline (left) and representative images of CCN3 at 12 DPP (right). Scale bar, 500 μm. LCD, low-calcium diet. **f,g**, Litter sizes (**f**) and

BV of femurs (**g**) from mothers injected with shCtrl or sh*Ccn3* (N = 5, 9) fed SD (0.8%Ca²⁺). **h**, Average pup weight (5–6 pups per litter) nursed by dams injected with shCtrl or sh*Ccn3* and fed SD (N = 5,9) or LCD (0.01% Ca²⁺, N = 4,2). **i**, Litters at 8 DPP nursed by dams injected with shCtrl or sh*Ccn3* and fed a LCD, with survival values in parentheses. Scale bar, 1 cm. **j**, Brain-derived MBH (that is, CCN3) replaces E2 as an osteoanabolic hormone during lactation and counteracts the catabolic actions of mammary-gland PTHrP to promote healthy bone formation, thereby ensuring adequate calcium supplies for milk and maternal skeleton integrity during lactation. One-way ANOVA for **c** (Šidák's multiple-comparisons test). Unpaired Student's *t*-test, two-tailed for **f**, and one-tailed for **g**. Three-way ANOVA for **h** (Tukey's multiple-comparisons test). *P < 0.05, **P < 0.01, ****P < 0.0001. Error bars ± s.e.m. Graphics in **j** (mammary, bone and calcium) were reproduced or adapted from BioRender (https://www.biorender.com). Graphic in **j** (dam and litter) was reproduced from Mind the Graph (https://mindthegraph.com) under a Creative Commons licence CC BY-SA 4.0.

## Brain CCN3 in lactating dams sustains pups

Moving beyond genetically engineered mouse models, we asked whether brain-derived CCN3 might function in the female life cycle, focusing on the postpartum period when maternal bone formation increases to maintain the skeletal calcium reservoir[10,39] and when circulating E2 is reduced[13]. As in virgin intact females, CCN3 expression was absent in ARC^ERα neurons during the early and later stages of pregnancy (Fig. 6a). However, by 7 days postpartum (DPP), CCN3 was present in ARC^ERα/KISS1 neurons of lactating dams (Fig. 6a,b), reaching near equivalent levels as found in *Esr1^Nkx2.1-cre* mutant females (Fig. 6c). Forced weaning reduced CCN3 in ARC^ERα neurons when examined 3 or 7 days after removal of pups (10 or 14 DPP, respectively), which suggested that the need for bone-promoting CCN3 lessens at the cessation of lactation when calcium demand is reduced (Fig. 6a). CCN3-positive neurons reside in close proximity to tanycytes lining the third ventricle in both mutant and lactating females, but were notably absent in OVX females (Fig. 6d). This result implied that oestrogen depletion by itself is insufficient to induce CCN3 production in ARC^ERα/KISS1 neurons.

Finally, to verify that CCN3 is an anabolic brain hormone during lactation, viral vector delivery of short hairpin RNA (shRNA) targeting *Ccn3* (sh*Ccn3*) was used to knockdown CCN3 in the ARC of adult virgin females before pregnancy (Fig. 6e and Extended Data Fig. 10a,b). sh*Ccn3* in the ARC did not affect fertility (time to plug), fecundity (litter size) or milk provision (Fig. 6f and Extended Data Fig. 10c,d). However, these dams experienced a 31% reduction in bone mass when fed calcium-rich breeder chow (0.8% Ca²⁺). Furthermore, when lactating mothers with

CCN3 knocked down (sh*Ccn3* mothers) were challenged postpartum with a low-calcium diet (0.01% Ca²⁺), the role of brain CCN3 in facilitating inter-generational resource transfer became evident. Despite their ability to suckle, pups nursed by a sh*Ccn3* mother failed to thrive, eventually leading to increased mortality (Fig. 6h,i). Pup viability depended on the status of brain CCN3 in mothers, as transfer of pups to a sh*Ccn3* mother resulted in 10% weight loss compared with 30% weight gain when nursed by dams injected with control shRNA (shCtrl) (Extended Data Fig. 10e). In summary, our data showing that ARC^ERα/KISS1 neurons produce CCN3 to maintain the maternal skeleton and viability of offspring establish a newly discovered role for this factor as an osteoanabolic maternal brain hormone (MBH) (Fig. 6j).

## Discussion

The role of ARC^ERα/KISS1 neurons as the gatekeeper of female reproduction and energy allocation is well established[25,40,41]. These neurons control multiple facets of physiology, including regulating pubertal onset and the hypothalamic–pituitary–gonadal axis[25]. Here we discovered another crucial function for ARC^ERα/KISS1 neurons in females in controlling bone homeostasis during lactation through the brain-derived osteoanabolic hormone MBH, that is, CCN3. Shutting down oestradiol production during lactation[42] poses a dilemma with respect to how osteoblast numbers increase and how mineralized bone surfaces are maintained while being 'plundered' for calcium[13] when ERα signalling is reduced[5]. This problem is especially acute in the trabecular-rich spine, which is susceptible to lactational osteoporosis[16]. Through MBH, ARC^ERα/KISS1 neurons solve this problem by

lifting the usual restraints on energetically costly bone formation. An outstanding question is how *Ccn3* expression is triggered in the ARC in females during this life stage. We note that unlike the marked increase in prolactin signalling during lactation[43], levels of this hormone are only modestly increased in mutant female mice[2], which indicates that alternative mechanisms may involve calcium sensing. Although depleting oestrogen signalling seems to prime *Ccn3* expression in the ARC, the absence of CCN3-positive neurons in OVX females implies that secondary events, whether shared or distinct, must drive *Ccn3* in both mutant and lactating females. Regardless, we posit that MBH plays a vital part during the anabolic phase of healthy, postpartum rapid bone remodelling[44]. Without MBH, bone loss is even greater during lactation owing to high PTHrP levels and low E2 levels. Thus, in mutant females, the early onset of CCN3 expression, in combination with E2 and absent PTHrP, rapidly generates strong, dense bones.

Our data across several models established that CCN3-mediated formation of new bone is coupled with higher bone quality and healthy bone remodelling. Although our study is seemingly at odds with suggestions that CCN3 inhibits osteogenesis and bone regeneration[37], these differences could reflect a dose effect of CCN3, with higher levels resulting in compensatory cellular responses or nonspecific receptor activation in bone niches that are anti-osteogenesis. In fact, we also observed inhibitory effects in both mouse and human ocSSC differentiation assays at higher CCN3 doses. Identifying the molecular target of CCN3 in ocSSCs and possibly other cellular populations, including osteocytes that reversibly remodel their perilacunar and canalicular matrix during lactation[45], will help resolve these discrepancies. Based on the presence of antiparallel β-strands and the carboxy-terminal cystine knot domain that mediates disulfide-linked dimerization[46], we predict that CCN3 circulates at low doses as a tightly held homodimer, binding its cognate receptor with high affinity, similar to other growth factors such as NGF.

Our study provides a new outlook in brain–body crosstalk[47], whereby hypothalamic neurons bypass the canonical hypophyseal portal route for transporting hypothalamic neuropeptides and, instead, release maternal hormones directly into the blood. For ARC^ERα/KISS1 neurons, this informational exchange is made possible by their juxtaposition with the tanycytes and fenestrated blood–brain barrier of the median eminence. In reverse, circulatory hormones such as leptin[48] and prolactin[49] exploit this weak barrier, shuttling into the brain to act directly on ARC neurons. Whether MBH is eventually exported to milk is unclear. We suggest that hormones, such as MBH, are in place to coordinate adaptive physiological responses in peripheral tissues, including the skeleton and gut[50], to meet the high demands of motherhood. Future directions of research include the potential translation of MBH in genetic and chronic bone diseases.

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

## Methods

### Ethics

Experiments were approved and performed in accordance with the guidelines of the University of California, San Francisco (UCSF) Institutional Animal Care Committee (IACUC) or the University of California, Davis (UCD) Animal Ethics Committee, the National Institutes of Health Guide for Care and Use of Laboratory Animals, and the recommendations of the International Association for the Study of Pain. The animals used in this study ranged from 1 to 70 weeks of age and included both male and female mice.

### Mice

The origin of the *Esr1*[fl/fl] allele (official allele: *Esr1*[tm1Sakh]) on a 129P2 background and used to generate *Esr1*[Nkx2.1-cre] mice have been previously described[2] and were maintained on a CD-1;129P2 mixed background. Primer sequences used for genotyping are listed in Supplementary Table 1. *Esr1*[Nkx2.1-cre-CAG-Luc-GFP] mice were generated by crossing male mice harbouring the *CAG-Luc-GFP* allele (official allele: *L2G85Chco/J*) to female mice homozygous for the *Esr1*[fl/fl] allele, followed by an additional cross to generate a *Esr1*[fl/fl;Nkx2.1-cre;Luc-GFP] colony, which was maintained on a mixed FVB/N, CD-1, 129P2 and C57BL/6 genetic background. *Esr1*[Prodynorphin-cre] mice were generated by crossing homozygous *Esr1*[fl/fl] females to *Prodynorphin-cre* (B6;129S-*Pdyn*[tm1.1(cre)Mjkr]/LowlJ, purchased from Jackson Laboratory) males. Mice were maintained on a 12-h light–dark cycle with ad libitum access to a standard breeder chow diet (PicoLab 5058; LabDiet, 4 kcal% fat, 0.8% Ca[2+]) and sterile water and housed under controlled and monitored rooms for temperature and humidity. Eighteen-month-old and 24-month-old C57BL/6 female and male mice were obtained through the NIA Aged Rodent Colony Program, available to NIA NIA-funded projects. To create cohorts of OVX females, ovariectomy was performed at 4 months of age, followed by 4 weeks of surgical recovery. All animal procedures were performed in accordance with UCSF and UCD institutional guidelines under the Ingraham and Ambrosi laboratories' Institutional Animal Care Committee protocol of record.

### Parabiosis

Parabiosis surgery followed previously described procedures[51]. In brief, 6-week-old *Esr1*[fl/fl] and *Esr1*[Nkx2.1-cre] females underwent mirror-image incisions at the left and right flanks through the skin, and incisions were made through the abdominal wall. The peritoneal openings of the adjacent parabionts were sutured together. Additionally, elbow and knee joints from each parabiont were sutured together. The skin incision of each mouse was then stapled. Each mouse was subcutaneously injected with enrofloxacin (Baytril, Bayrer) antibiotic and buprenorphine (Butler Schein) and monitored during surgical recovery. To monitor the health of pairs, their body weights and grooming behaviours were monitored weekly. The pairing was checked by the presence of Evans Blue dye in blood collected 2 h after injection of 200 µl of 0.5% Evans Blue from the submandibular of the uninjected parabiont.

In vivo µCT was performed to determine changes in trabecular bone mass over time using a Scanco Viva CT40 high-speed µCT preclinical scanner. Before scanning, mice were anaesthetized and placed in a 3D-printed nose cone to accommodate surgically paired mice. Distal femurs of each mouse in the parabiont group (opposite to the incision side) were imaged 2 weeks after surgery at baseline (0) and then at 3, 6 and 17 weeks from baseline imaging at intervals that preserve bone mass[52]. Parabionts were anaesthetized, and a 2 mm region of the contralateral distal femur (opposite to the surgically paired side) was used for assessing the trabecular bone compartment of 1 mm length proximal to the epiphyseal plate and cortical parameters at the diaphysis in an adjacent 0.4 mm region of the femur. Imaging was performed at the UCSF Skeletal Biology and Biomechanics Core supported by the NIAMS Award P30AR075055.

### Bone, ocSSC kidney and brain transplants

In brief, long bones were dissected from 4-week-old female or male *Esr1*[fl/fl] mice, cleaned of excess tissue and immediately implanted under the skin after creating a horizontal incision and small pocket posterior to the scapula in 8-week-old acceptor *Esr1*[fl/fl] or *Esr1*[Nkx2.1-cre] female mice. After 6 weeks of incubation, donor femurs were subsequently removed and analysed by ex vivo µCT. In brief, volumetric bone density and volume were measured at the distal femur using a Scanco Medical µCT 50 specimen scanner calibrated to a hydroxyapatite phantom or Bruker SkyScan1276 (Bruker Preclinical Imaging). For Scanco imaging, samples were fixed in 10% phosphate-buffered formalin and scanned in 70% ethanol using a voxel size of 10 mm and an X-ray tube potential of 55 kVp and X-ray intensity of 109 µA. Scanned regions included a 2 mm region of the femur proximal to the epiphyseal plate. For Bruker Skyscan 1276 imaging, a source voltage of 85 kV, a source current of 200 µA, a filter setting of AI 1 mm and a pixel size of 12–20 µm (a set number was used for samples of a specific experiment) at 2,016 × 1,344 were used. Reconstructed samples were analysed using CT Analyser and CTvox software (Bruker). Standard best practices were used to quantify trabecular and cortical bone parameters[52]. Image acquisition of an implanted femur was captured using an iPhone 13 Pro and then edited in Photoshop CC.

For kidney transplant assays, primary ocSSCs were isolated as described below from 4-week-old *Esr1*[fl/fl] female mice. Approximately 20,000 live cells were resuspended in 2 µl Matrigel (Corning, 356234), and the entire mixture was injected into the renal capsule of 8-week-old recipient mice. Six weeks after transplantation, animals were then euthanized and kidney grafts were processed as described below. Excised kidneys with renal capsule grafts and brain tissue with injected stem cell grafts were scanned using the same settings. For the processing of kidneys after SSC transplants, kidneys were dissected out and cleaned of soft tissue, fixed in 4% paraformaldehyde (PFA) and embedded in OCT for cryosectioning. Sections (5 µm) were subsequently stained using a standard Movat's pentachrome staining kit (Abcam, ab245884). Bright-field images were taken using a Luminera Infinity-3 and quantified using ImageJ software. Excised kidneys containing renal capsule grafts injected with SSCs were scanned using the same settings as above for bone transplant assays.

For brain transplants, *Esr1*[fl/fl-CAG-Luc-GFP] ocSSCs were isolated by FACS as described below and kept on ice in sterile artificial cerebral spinal fluid (Tocris Bioscience, 3525). Approximately 2 h after cell isolation, 400–700 ocSSCs in 1 µl of solution were delivered bilaterally above the third ventricle by the ARC nucleus by stereotaxic injection (anterior–posterior (AP): −1.58; medial–lateral (ML): ±0.3; dorsal–ventral (DV): −5.95 from the skull) into 12–18-week-old *Esr1*[fl/fl] or *Esr1*[Nkx2.1-cre] females. Six weeks after implant, mice were perfused, and immunohistochemistry was performed on cryosections (20 µm) collected from brains fixed in 4% PFA using standard procedures. GFP staining used a polyclonal chicken anti-GFP antibody (Novus Biologicals, NB100-1614) at 1:2,500. Images were taken using a Keyence B2-X800. Excised medial basal hypothalamic brain tissues from female mice injected with stem cells were scanned using the same settings as described above for bone transplant assays.

### Flow cytometry isolation of primary SSCs

Flow cytometry and cell sorting were performed on a FACS Aria II cell sorter (BD Biosciences) and analysed using FlowJo software. Mouse long bones and callus samples from human patients (UC Davis Institutional Review Board (IRB) 1997852; SCRO-1199) were dissected and freed from the surrounding soft tissue, which was then followed by dissociation with mechanical and enzymatic steps as previously described[53,54]. For isolation of human SSCs, six samples of callus tissue from individuals (age range 14–72 years) with fractured bones were acquired during the process of open reduction internal fixation of fractures at UCD

Medical Center. Collection adhered to IRB guidelines (IRB 1997852-3, without restrictions based on the race, sex or age of the donors of the specimens). The isolation of SSCs performed in this study does not meet the criteria for human research according to the IRB assessment. Hence, informed consent was not sought. Only fractures treated with an open approach and direct realignment of fracture fragments were included, and haematoma or callus tissue hindering satisfactory realignment of displaced fragments was excised and preserved for research purposes during the surgery. Excised tissues were promptly placed on ice, and human SSCs were isolated within 5 h after surgery, as detailed below.

In brief, the tissue was placed in collagenase digestion buffer supplemented with DNase and incubated at 37 °C for 60 min under constant agitation. After collagenase digestion and neutralization, undigested materials were gently triturated by repeated pipetting. Total dissociated cells were filtered through a 70-µm nylon mesh and pelleted at 200$g$ at 4 °C for 5 min. Cells were resuspended in ammonium–chloride–potassium lysing buffer to eliminate red blood cells and centrifuged at 200$g$ at 4 °C for 5 min. The pellet was resuspended in 100 µl staining medium (2% FBS/PBS) and stained with antibodies for at least 30 min at 4 °C (antibody information can be found in Supplementary Table 2). Living cells were gated for lack of propidium iodide (1:1,000 diluted stock solution: 1 µg ml$^{-1}$ in water; mouse cells) signal or DAPI (human cells). Compensation, fluorescence-minus-one control-based gating and FACS isolation were conducted before analysis or sorting using established antibody cocktail combinations. A complete list of antibodies used for FACS purification of SSCs is presented in Supplementary Table 2.

For mouse SSC lineages, the following antibodies were used: CD90.1 (Thermo Fisher, 47–0900), CD90.2 (Thermo Fisher, 47–0902), CD105 (Thermo Fisher, 13–1051), CD51 (BD Biosciences, 551187), CD200 (Thermo Fisher, MA5-17980), CD45 (BioLegend, 103110), Ter119 (Thermo Fisher, 15–5921), Tie2 (Thermo Fisher, 14–5987), 6C3 (BioLegend, 108312), streptavidin PE-Cy7 (Thermo Fisher, 25–4317), Sca-1 (Thermo Fisher, 56-5981), CD45 (Thermo Fisher, 11–0451), CD31 (Thermo Fisher, 12-0311), CD140a (Thermo Fisher, 17–1401) and CD24 (Thermo Fisher, 47–0242).

For human SSC isolation, the following antibodies were used: CD45 (BioLegend, 304029), CD235a (BioLegend, 306612), CD31 (Thermo Fisher Scientific, 13-0319), CD202b (TIE-2) (BioLegend, 334204), streptavidin APC-AlexaFlour750 (Thermo Fisher, SA1027), CD146 (BioLegend, 342010), PDPN (Thermo Fisher Scientific, 17-9381), CD164 (BioLegend, 324808) and CD73 (BioLegend, 344016).

## Cell culturing and differentiation assays of primary mouse and human SSCs

Only freshly sorted primary mouse or human ocSSCs were used in this study. After cell isolation by FACS, primary cells were cultured as described above. Mouse cells were cultured in minimum essential medium-α (MEMα) with 10% FBS and 1% penicillin–streptomycin (Thermo Fisher, 15140–122) and maintained in an incubator at 37 °C with 5% CO$_2$. Human cells were cultured in MEMα (Fisher Scientific, 12561-056) with 10% human platelet-derived lysate (Stem Cell Technologies, 06960) and 1% penicillin–streptomycin solution (Thermo Fisher Scientific, 15140-122). To induce osteogenic differentiation, pre-confluent cells were supplemented with osteogenesis-inducing factors, 100 nM dexamethasone, 0.2 mM L-ascorbic acid 2-phosphate and 10 mM β-glycerophosphate for 14 days.

For testing of candidate factors (mCCN3 (Novus Biologicals, NBP2-35100), hCCN3 (Novus Biologicals, NBP2-35084), mFST (Novus Biologicals, NBP2762685U), hFST (Stem Cell Technologies, 50-197-6487), BAM-22P (Sigma, SCP0057), met-ENK (Sigma, M6638) and hGRP (RayBiotech, 230-00695-10)), indicated concentrations were added to defined medium and changed every second day with fresh medium. Cells were then formalin-fixed and stained with 2% Alizarin Red S (Roth) in distilled water. Wells were washed twice with PBS and once with distilled water. Oil Red O staining was performed by fixing cells with 4% PFA for 15 min

at room temperature using an Oil Red O working solution prepared from a 0.5% stock solution in isopropanol and diluted with distilled water at a ratio of 3:2. The working solution was filtered and applied to fixed cells for at least 1 h at room temperature. Cells were washed four times with tap water before evaluation. CFU-F assays were conducted by freshly sorting a defined number of cells of desired cell populations into separate culture dishes containing expansion medium. The medium was changed twice a week. Cells were fixed and stained with crystal violet (Sigma) on day 10 of culturing.

For in vitro osteoclastogenesis assays, bone marrow macrophages were isolated from 3-month-old and 24-month-old C57Bl/6 male mice as previously described[55]. For osteoclast generation, cells were cultured with 30 ng ml$^{-1}$ M-CSF and 10 ng ml$^{-1}$ Rankl (R&D systems) treated with and without recombinant mCCN3 (0.25 M and 2.5 M) (R&D Systems, 1976-NV-050) for 4 days with medium exchange performed daily. On day 4, cells were fixed with 4% PFA, and TRAP staining was performed according to the manufacturer's protocol (Sigma-Aldrich). TRAP-positive cells with two or more nuclei per well were counted. Bone marrow macrophages were obtained from four mice per treatment group.

## HFD, LCD challenges and metabolic parameters
HFD was purchased from Research Diets (D12492, 60 kcal% fat, 0.78% Ca$^{2+}$). $Esr1^{fl/fl}$ and $Esr1^{Nkx2.1-cre}$ mice were maintained on a HFD for 17 weeks starting from 10 weeks of age. Glucose tolerance tests were conducted after a 6-h fast with glucose administered (i.p., 1.0 g kg$^{-1}$ of body mass). For glucose tolerance tests, mice were subjected to 6 h of fasting (starting at about ZT2) and injected with glucose (i.p., 1 g kg$^{-1}$). Tail-blood samples were collected at baseline at 15, 30, 45, 90 and 120 min after glucose injection. Blood glucose levels were quantified using a hand-held glucometer (Roche, Accu-Check Compact). For non-fasting triglyceride measurements, whole blood was collected from 27-week-old $Esr1^{fl/fl}$ and $Esr1^{Nkx2.1-cre}$ mice into EDTA-treated tubes (Microvette CB 300 K2E) and placed directly on ice. To isolate plasma, whole blood was spun down at 2,000$g$ for 15 min at 4 °C, and the supernatant was collected. Non-fasting plasma triglycerides levels were then measured using a commercially available kit (Cayman Chemicals, 10010303) as per the manufacturer's protocols. All plasma samples were stored at −80 °C before analysis. Body composition to determine per cent lean and fat mass was obtained by dual-energy X-ray (DEXA, GE Lunar PIXImus). LCD was purchased from Teklad (TD.95027, 14.7 kcal% fat, 0.01% Ca$^{2+}$). For lactation studies, a subset of $Esr1^{fl/fl}$ females were switched from standard breeder chow after parturition and maintained on LCD for 12 days before collection of long bones and quantification of bone using parameters described above for Bruker Skyscan 1276 imaging.

## Bone parameters
Volumetric bone density and BV for mice fed SD and HFD were measured at the right femur using a Scanco Medical µCT 50 specimen scanner calibrated to a hydroxyapatite phantom. In brief, samples were fixed in 10% phosphate-buffered formalin and scanned in 70% ethanol. Scanning was performed using a voxel size of 10 mm and an X-ray tube potential of 55 kVp with an X-ray intensity of 109 µA. Scanned regions included a 2 mm region of the femur proximal to the epiphyseal plate and a 1 mm region of the femoral mid-diaphysis. Scanned femurs were performed with 10 µm resolution at 70 kV, 57 µA, 4 W and an integration time of 700 ms. The analysis threshold for cortical and trabecular bone was 0.8 sigma, 1 support and 260 (lower) and 1,000 (upper) permille. Volumes of interest were evaluated using Scanco evaluation software. Representative 3D images were created using Scanco Medical mCT Ray (v.4.0) software.

## Bone histomorphometry analysis
Mice were injected with 20 mg kg$^{-1}$ calcein (Sigma-Aldrich) 7–9 days before euthanasia and with 15 mg kg$^{-1}$ of Alizarin (Sigma-Aldrich)

2 days before euthanasia. Bones were fixed in 4% PFA, dehydrated in 30% sucrose and embedded in OCT or embedded in MMA plastic. Standard undecalcified sections (5 mm) were cut using a microtome (Leica CM1950) together with the CryoJane Tape-Transfer System. Mounted sections were imaged with an ECHO REVOLVE R4 using FITC (Calcein) and Texas Red (Alizarin Red) channels. A standard sampling site with an area of 2.5 mm$^2$ was established in the secondary spongiosa of the distal metaphysis. Before histomorphometry analyses, mosaic-tiled images of distal femurs were acquired at ×20 magnification with a Zeiss Axioplan Imager M1 microscope (Carl Zeiss MicroImaging) fitted with a motorized stage. The tiled images were stitched and converted to a single image using Axiovision software (Carl Zeiss MicroImaging) before blinded analyses was performed using two image-analysis software programs: Bioquant OSTEO or ImageJ. The following variables were analysed: %BV/TV, mineral apposition rate, mineral surface/bone surface (MS/BS), bone formation rate/bone surface (BFR/BS), osteoblast number/bone surface (No. Ob/BS), lacunar density (N Lacunae/BA), TRAP-positive osteocytes (%), and osteoclast number/bone surface (No. Oc/BS). Adjacent sections were stained with Alizarin Red for overview bright-field images.

### Haematoxylin and eosin, TRAP, silver nitrate staining and osmium staining for BMAT quantification

Femoral or tibial samples were fixed in 4% PFA and demineralized in 10% EDTA for 10–14 days before being embedded in MMA plastic. Sections (5 μm) were cut using a Leica RM2165 and subsequently stained with haematoxylin and eosin (H&E) or stained with TRAP. Photoshop software removed the background in non-tissue areas for images of the proximal tibias. Silver nitrate staining of cortical osteocytes was conducted on sections of undecalcified frozen femurs. In brief, after removal of OCT, sections were incubated in 10% EDTA for 1 h at room temperature. Subsequently, slides with sections were incubated in a silver nitrate–gelatin solution, mixed using 2 parts of 50% w/v silver nitrate and 1 part 2% gelatin in 1% formic acid, at room temperature for 55 min, followed by a 2 min water wash. Sections were then incubated in 5% sodium thiosulfate for 10 min, followed by another 2 min water wash step. Dehydrated sections, protected with a cover slip, were imaged. Images were used for counting osteocytes in defined cortical areas.

Quantification of BMAT followed a published protocol[32]. In brief, femurs were decalcified in 14% EDTA, pH 7.4 for 2 weeks, followed by incubation with a PBS solution containing 1% osmium tetroxide (Electron Microscopy Sciences 19170) and 2.5% potassium dichromate (Sigma-Aldrich 24–4520) for 48 h. After washing for 2 h with water, osmium-stained bones were embedded in 2% agarose before scanning at 10 μm voxel resolution with a Scanco μCT 40 scanner. Regions of interest were contoured and analysed with a threshold of 400 for BMAT quantification. Specifically, a region of 2 mm immediately above the growth plate in distal metaphysis was used for the quantification of regulated BMAT in femurs.

### Biomechanical strength testing and stabilized bicortical femoral fracture model

Femurs underwent a three-point bend test using mechanical load frames (Instron E100 or EnduraTEC, ELF3230). A span of 7 mm separated the lower supports to support two ends of the specimen. The testing head was aligned at the midpoint between the supports. Femurs were preloaded to a force of 1 N and then loaded at a rate of 0.1 mm s$^{-1}$. Loading was terminated after mechanical failure, determined by a drop in force to 0.5 N. Force displacement data were collected every 0.01 s. All tests were performed at room temperature using an electromechanical load frame as specified above.

Femurs of anaesthetized mice were exposed following muscle distraction and lateral dislocation of the patella. A 25-gauge needle was inserted between the femoral condyles to provide relative intramedullary fixation before creating a transverse, mid-diaphysis fracture using

micro-scissors. Hydrogels were then immediately placed at the fracture site. The patella was relocated, and 6-0 nylon suture (Ethicon) was used to re-approximate the muscles. Mice were euthanized 21 days after surgery, femurs with fracture calluses were dissected and intramedullary pins removed for subsequent analyses. For hydrogel fabrication, eight-arm poly(ethylene glycol) vinyl sulfone (PEG-VS) (10 kDa) (JenKem) was dissolved in HEPES (25 mM, pH 7.2) at a 2× concentration. Murine recombinant CCN3 (R&D Systems) was added to the precursor solution to reach a final concentration of 1 or 2 μg per hydrogel. Hydrogels without growth factors served as controls and were loaded with HEPES of equal volume. GPQ-A (GCRDGPQGIAGQDRCG, GenScript), a protease-cleavable crosslinking peptide, was mixed at a 1:2 volume ratio with PEG-dithiol (PEG-DT) (3.5 kDa) (JenKem) at a 2× concentration in medium (pH 8.3) to permit matrix metalloproteinase-mediated degradation[56]. The precursor solutions were mixed at a 1:1 volume ratio and pipetted onto a silicone mould for a final volume of 6 μl per hydrogel. Fracture calluses were scanned using a SkyScan1276 (Bruker Preclinical Imaging) with settings described above and analysed by selecting 50 sections in both directions of the fracture site, producing a total area of 100 sections. For analysis of callus mineralization, CTAn software was used to select a region of interest spanning the fracture callus area outside the intramedullary space, excluding cortical bone tissue.

### Plasma collection and whole bone assays

In brief, 300 μl of whole blood was collected from the submandibular vein from pre-pubertal $Esr1^{fl/fl}$ and $Esr1^{Nkx2.1-cre}$ female mice into EDTA-treated tubes (Microvette CB 300 K2E) and placed directly on ice. To isolate plasma, whole blood was spun down at 2,000$g$ for 15 min at 4 °C, and the supernatant was collected. The right and left femurs of 10–11-week-old control female and male mice ($Esr1^{fl/fl}$) were collected and cleaned of soft tissue. Femurs collected from 18-month-old, aged female mice (strain C57BL/6) provided by the National Institutes of Aging were also tested in whole bone assays. The left femur was immediately fixed in 4% PFA and then transferred into PBS at 4 °C for histological assessment to obtain baseline measurements. The right femur was cultured in a 12-well plate containing 1.4 ml primary culture medium (α-MEM; containing L-glutamine and nucleosides; Mediatech), supplemented with 10% FBS (Atlanta Biologicals) and 100 U ml$^{-1}$ penicillin–streptomycin (Mediatech). The left and right femurs were treated with 15 μl of plasma from $Esr1^{fl/fl}$ and $Esr1^{Nkx2.1-cre}$ females, respectively, or with mCCN3 (14 μl of 0.0125 μg μl$^{-1}$ of recombinant mCCN3, 1976-NV-050, R&D Systems in 1.4 ml of primary culture medium) or with vehicle (14 μl of 0.9% normal saline in 1.4 ml of primary culture medium), respectively. To assess the degradation of whole bone during culturing, the right tibia or femur was cultured in medium with 14 μl of 0.9% normal saline for 5 days (saline). These were compared with the baseline contralateral femur, which were immediately chilled and fixed in 4% PFA for analysis (baseline). Medium changes, including plasma, mCCN3 or vehicle treatments, were performed daily. Femurs were collected after 5 days of culture, fixed in 4% PFA and then transferred into PBS at 4 °C before μCT imaging. After μCT imaging, femurs were processed for histology as described below.

Femoral samples were cleaned of soft tissue, fixed in 4% PFA and demineralized in 10% EDTA for 10–14 days before embedding in paraffin wax. Sections measuring 5 μm were then cut using a Leica RM2165 and subsequently stained with a Movat's pentachrome staining kit (Abcam, ab245884). For H&E staining analysis, femurs were collected, fixed in 4% formalin, decalcified in Cal-Rite, dehydrated in 30% sucrose and embedded in OCT. Then, 5 μm standard sections were cut using a microtome (Leica CM1950).

### Western blotting

Hepatic protein lysates were prepared as previously described[57]. Plasma was isolated as described above, and CCN3 protein was enriched by heparin–agarose affinity purification[58]. Heparin–agarose beads (Sigma, H6508; 200 μl per sample) were washed and equilibrated in PBS with

protease inhibitors (Thermo, 78425), mixed with a volume of plasma equivalent to 1 mg of total protein, and incubated overnight at 4 °C with constant rotation. Beads were then washed four times with PBS, and proteins were eluted by boiling the beads for 10 min in Laemmli sample buffer (Bio-Rad, 1610747) containing 50 mM dithiothreitol. Hepatic (10 µg) and affinity-purified plasma proteins were separated by SDS–PAGE and transferred to nitrocellulose membranes (Bio-Rad, 170-4270). Protein loading levels were assessed by Ponceau S staining (Thermo, A40000279). Membranes were de-stained and blocked in TBS-T (0.1% Tween 20) with 5% normal donkey serum (Abcam, AB7475). Blots were then probed overnight at 4 °C with anti-CCN3 antibody (R&D Systems, AF1976; 1:3,000) in TBS-T with 5% serum. After washing in TBS-T, blots were incubated with HRP-conjugated secondary antibody (Invitrogen, A15999; 1:30,000) for 1 h at room temperature, washed in TBS-T, incubated with chemiluminescent substrate (Thermo, 34577) and imaged (Azure Biosystems).

### Brain RNAscope and immunohistochemistry
Fluorescent immunohistochemistry was performed using RNAscope (ACD, Multiplex Fluorescent V2) according to the manufacturer's protocol using the following probes: *Ccn3* (ACD, 415341-C2), *Esr1* (ACD, 478201), *Penk* (ACD, 318761) and *Kiss1* (ACD, 500141-C1).

Immunohistochemistry was performed using primary antibodies against ERα (EMD Millipore, C1355 polyclonal rabbit, 1:750 dilution), CCN3 (R&D Systems, AF1976 polyclonal goat, 1:1,000 dilution), VIM (Abcam, AB92547-1001 monoclonal rabbit, 1:1,000) and KISS1 (Abcam, ab19028 polyclonal rabbit, 1:200 dilution) diluted in PBS with 0.1% Triton-X100, 5% normal donkey serum and 5% BSA. For detection, sections were labelled with species-appropriate secondary Alexa Fluor-coupled antibodies (Invitrogen, A-21447, A10042 or A-11055; 1:1,000 dilution). Slides were imaged using a Keyence BZ-X800 wide-field fluorescence microscope. Confocal images were acquired at the UCSF Nikon Imaging Center using a Nikon CSU-22 with an EMCCD camera and MicroManager (v.2.0gamma). Images were processed and quantified using ImageJ Fiji (v.1.52i) and the Cell Counter plugin (v.2). Three representative views of each sample were selected. A complete list of all antibodies used in immunohistochemistry analyses is listed in Supplementary Table 2. Cryosections (20 µm) collected from brains fixed in 4% PFA were used for both fluorescent immunohistochemistry and immunostaining.

### siRNA and shRNA studies
Mice were secured in a stereotaxic frame (Model 1900, David Kopff Instruments), and 400 nl of *Ccn3* or non-targeting siRNA pools (Dharmacon, E-040684-00-0010 or D-001810-10-05, 0.4 mM) were injected bilaterally to the ARC at the following coordinates: AP: Bregma −1.58 mm, ML: Bregma ±0.25 mm, DV: skull −5.9 mm. For shRNA studies, female mice were injected bilaterally (200 nl per side, $2.53 \times 10^{13}$ GC ml$^{-1}$) and allowed to recover for 10–14 days before mating with male mice. At 12 days after injection (siRNA) or 12 DPP (shRNA), female mice were euthanized, and brain and bone samples were collected and processed as described above. Isolated femurs were then imaged by µCT as described above for bone transplant assays. Owing to the high bone phenotype of mutant female mice, thresholding and region of interest selection were adjusted between different experiments but kept consistent within each individual experiment.

### CCN3, S961 and naloxone in vivo treatments
For recombinant CCN3, 10-week-old control female and 13-week-old control male mice (*Esr1$^{fl/fl}$*) were injected daily with recombinant mCCN3 (i.p., 7.48 µg kg$^{-1}$; R&D Systems, 1976-NV-050) or vehicle control (i.p., 0.9% normal saline) for 21 days. Seven days before euthanasia, mice were first injected with 20 mg kg$^{-1}$ calcein (Sigma-Aldrich) and then 15 m kg$^{-1}$ of Alizarin (Sigma-Aldrich) 2 days before euthanasia. Right femurs were cleaned of soft tissue, fixed in 4% PFA and then stored

in PBS at 4 °C before imaging. Isolated femurs were imaged by µCT scanning as described above for bone transplant assays. After imaging, femurs were then processed for H&E histology and dynamic histomorphometry as described above. For S961 treatment, 10-week-old *Esr1$^{fl/fl}$* and *Esr1$^{Nkx2.1-cre}$* female mice were infused with continuous S961 (70 nM per osmotic pump), an insulin peptide receptor antagonist, as adapted from a previous study[31] and obtained from the Novo Nordisk Compound Sharing Program (NNC0069-0961) or vehicle solution through an ALZET mini osmotic pump (Model 1004), which was exchanged once after 4 weeks for a total infusion period of 8 weeks. S961 was reconstituted in 0.9% normal saline. The mini osmotic pumps were implanted into an interscapular subcutaneous pocket under isoflurane anaesthesia and exchanged once. For naloxone treatments, 10-week-old *Esr1$^{fl/fl}$* and *Esr1$^{Nkx2.1-cre}$* female mice were infused with continuous naloxone, a non-selective opioid antagonist (0.5 mg per 24 h over 4 weeks, Tocris, 0599/100) or vehicle solution through an ALZET mini osmotic pump (Model 1004). Naloxone was reconstituted in the vehicle solution containing 0.9% normal saline. The mini osmotic pump was implanted into an interscapular subcutaneous pocket under isoflurane anaesthesia.

### Hepatic viral transduction of mCCN3
Ectopic hepatocyte expression of mCCN3 protein was achieved using AAVdj viral vectors with high liver tropism[59] encoding mouse CCN3 under the control of the constitutive cytomegalovirus immediate-early enhancer/chicken β-actin promoter (CAG) promoter (Vector Biolabs, AAV-265951). Viral vectors were first diluted in sterile saline, and control (*Esr1$^{fl/fl}$*) mice were injected retro-orbitally with 100 µl of $0.5 \times 10^{10}$, $3 \times 10^{10}$ or $15 \times 10^{10}$ GC per mouse titres of AAVdj-CAG-CCN3 or the negative control. After 2 weeks, one animal from each group was euthanized, and its liver was checked for expression of CCN3 using an anti-CCN3 antibody as listed in Supplementary Table 2. After 5 weeks, mice were euthanized, and bone, plasma and liver were collected. Liver samples were divided and processed separately for analysis of *Ccn3* mRNA expression by qPCR or CCN3 protein expression by immunohistochemistry. Total liver RNA was isolated by phenol–chloroform extraction and purified using a RNeasy Mini kit (Qiagen, 74104). qPCR was performed as described below. For immunohistochemistry, liver samples were drop-fixed in 4% PFA, cryosectioned (10 µm) and antibody stained as described above. Isolated femurs were cleaned and then imaged as described above for bone transplant assays.

### RNA isolation, qPCR and bulk RNA-seq
Microdissected ARC or medial basal hypothalamic tissue was obtained from control and mutant female mice (1–27 weeks of age) using the optic chiasm as a reference point, and a 2 mm block of tissue containing the hypothalamus was isolated with a matrix slicer. For ARC, total RNA was purified using a RNA Mini kit (Invitrogen). For qPCR, cDNA was synthesized using an Applied Biosystems High-Capacity cDNA Reverse Transcription kit. Expression analysis was performed using SYBR Green. Values were normalized to either *36b4* or *mCyclo*. Sequences for primer pairs are provided in Supplementary Table 1.

For bulk RNA-seq analyses, barcoded sequencing libraries were prepared using a NEBNext Ultra II RNA Library Prep kit for Illumina from RNA samples after a quality check, and sequencing was performed on Illumina's NovaSeq 6000, S4 flow cell. Novogene carried out these steps. For all tissue samples, sequencing-generated reads were aligned to the mouse transcriptome (mm10) using Kallisto in gene mode[60]. Differential gene expression was evaluated using the likelihood-ratio test by Sleuth (qval < 0.05)[61]. All heatmaps were generated with the top 50 female/male-biased genes obtained from 27-week-old mice and were generated in R[62].

### Plate-based Smart-Seq2 scRNA-seq of ocSSCs
Single ocSSCs from 4-week-old *Esr1fl/fl* and *Esr1$^{Nkx2.1-cre}$* long bones were isolated by FACS using processing and flow cytometry protocol as

described above. Single-cell suspensions pooled from four mice per group was used, and individual cells were captured in separate wells of a 96-well plate containing 4 µl lysis buffer (1 U µl⁻¹ RNase inhibitor (Clontech, 2313B)), 0.1% Triton (Thermo Fisher Scientific, 85111), 2.5 mM dNTP (Invitrogen, 10297–018), 2.5 µM oligo dT30VN (IDT, custom: 5′–AAGCAGTGGTATCAACGCAGAGTACT30VN-3′) and 1:600,000 External RNA Controls Consortium ExFold RNA Spike-In Mix 2 (ERCC; Invitrogen, 4456739) in nuclease-free water (Thermo Fisher Scientific, 10977023) according to a modified Smart-Seq2 protocol[54,63]. Two 96-well plates per phenotype with a single ocSSC per well were sorted and processed. Plates were spun down and kept at −80 °C until cDNA synthesis, which was conducted using oligo-dT-primed reverse transcription with SMARTScribe reverse transcriptase (Clontech, 639538) and a locked nucleic acid containing template-switching oligonucleotide (Exiqon, custom: 5′-AAGCAGTGGTATCAACGCAGAGTACATrGrG+G-3′). PCR amplification was conducted using KAPA HiFi HotStart ReadyMix (Kapa Biosystems, KK2602) with in situ PCR primers (IDT, custom: 5′-AAGCAGTGGTATCA-ACGCAGAGT-3′). Amplified cDNA was then purified with Agencourt AMPure XP beads (Beckman Coulter, A63882). After quantification, cDNA from each well was normalized to the desired concentration range (0.05–0.16 ng µl⁻¹) by dilution and consolidated into a 384-well plate. Subsequently, this new plate was used for library preparation (Nextera XT kit; Illumina, FC-131–1096) using a semi-automated pipeline. The barcoded libraries of each well were pooled, cleaned-up and size-selected using two rounds (0.35× and 0.75×) of Agencourt AMPure XP beads (Beckman Coulter), as recommended by the Nextera XT protocol (Illumina). A high-sensitivity fragment analyser run was used to assess fragment distribution and concentrations. Pooled libraries were sequenced on NovaSeq6000 (Illumina) to obtain 1–2 million 2× 151 base-pair paired-end reads per cell.

## scRNA-seq data processing

Sequenced data were demultiplexed using bcl2fastq2 (v.2.18; Illumina). Raw reads were further processed using a skewer for 3′ quality trimming, 3′ adaptor trimming and removal of degenerate reads. Trimmed reads were then mapped to the mouse genome (v.M20) using STAR (v.2.4), and counts for gene and transcript reads were calculated using RSEM (v.1.2.21). Data were explored, and plots were generated using Scanpy (v.1.9). To select high-quality cells only, we excluded cells with fewer than 450 genes, and genes detected in fewer than three cells were excluded. Cells with a mitochondrial gene content higher than 5%, ERCC content higher than 30% and ribosomal gene content higher than 5% were excluded as well. Scrublet was then used to detect and remove residual duplicates. A total of 264 high-quality cells (122 control and 142 mutant mouse cells) were included in the final analysis. Raw counts per million (CPM) values were mean-normalized and log-normalized, and then data were scaled to a maximum value of 10. Combat batch correction was applied to account for potential biases through minor differences in cell processing. Principal component (PC) 'elbow' heuristics were used to determine the number of PCs for clustering analysis with UMAP and Leiden algorithm (leidenalg). Differential gene expression between $Esr1^{fl/fl}$ (WT) and $Esr1^{Nkx2.1-cre}$ (mutant), as well as Leiden clusters, was calculated using Wilcoxon rank-sum test. EnrichR was used to explore enrichment for pathways and ontologies of DEGs between WT and mutant groups[64].

## Statistics

Statistical tests, excluding RNA-seq analyses, were performed using Prism 10 (GraphPad). A description of the test and results are provided in Supplementary Table 3. Multiple comparisons correction for one-way, two-way and repeated-measures ANOVA were performed using the post hoc tests as indicated in each figure legend. For all panels in the main figures and Extended Data figures, $N$ indicates biological sample sizes used and $n$ indicates technical replicates in cell culture

assays. Outliers were identified using Grubbs test ($\alpha = 0.05$). Unless otherwise noted, data are presented as mean ± s.e.m. or box plots, in which whiskers represent minimum and maximum values, edges of the box are 25th and 75th percentiles, and the centre line indicates the mean. Sample sizes are based on previous work from our laboratories; however, no specific statistical calculation was performed to determine sample size. For AAVdj-CCN3 and siRNA injections, mice of identical genotypes were drawn at random from littermate pools to receive functional or control virus injections. Experimenters were blinded to the type of AAV received and genotype of the mice under study for all subsequent µCT and dynamic histomorphometry analyses. All raw data and processed data files for the bulk RNA-seq and scRNA-seq are publicly available at the Gene Expression Omnibus (GEO) under sample accession numbers GSE248882 and GSE241478, respectively. A list of reagents used in this study is provided in Supplementary Table 1. Graphics in Figs. 1b, 2b (kidney), 5c and 6j (mammary, bone and calcium) were reproduced or adapted from BioRender (https://www.biorender.com). Graphics in Figs. 1g, 2b (mouse) and 6j (dam and litter) were reproduced or adapted from Mind the Graph (https://mindthegraph.com) under a Creative Commons licence CC BY-SA 4.0.

## Reporting summary

Further information on research design is available in the Nature Portfolio Reporting Summary linked to this article.

## Data availability

All data generated or analysed during this study are included in the article and its supplementary information files. Source data are provided with this paper.

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

**Acknowledgements** We mourn the loss of our beloved colleague and collaborator Charles (Chuck) K. F. Chan, whose creative thinking and original scientific approach were crucial to this study. Chuck passed away during the resubmission process on 12 March 2024. We thank J. Argiris for assistance with animal husbandry and genotyping; T. Tsusaka for technical guidance; all members of our groups for suggestions and critical comments in the early stages of this project, including D. Julius, as well as R. Kalia for his structural insights into CCN3 and its potential mechanism of action; and the UC Davis orthopaedic surgeons A. Whitaker and A. Saiz for human specimen collection. This work was supported by NIH Transition Awards, including an NIDDK Emerging Physician Scientist Award R01DK121657-S1 (to M.E.B.), NIA-1K01AG065916 (to C.B.H.), NIGMS K12 IRACDA 5K12GM081266 (to R.R.), NIDDK K99DK129763 (to J.N.) and NIA K99/R00 AG066963 (to T.H.A.). Grants from the US NIH include R01 support NDDK R01DK132073 (to E.L.S.), R01AG067740 (to S.V.), NIA R01AG070647 (to N.E.L.) and R01AG062331, R01DK121657 (to H.A.I.). Non-NIH Support came from a Stanford Pilot Award

(to H.A.I. and C.K.F.C.), the CIRM training grant EDUC4-1279 (to E.E.W.) and a Senior Scholar Award GCRLE0320 (to H.A.I.). Core services were made possible by funding from NIH NIAMS Support for Cores P30AR075055 (at UCSF) and P30AR075055 (at Wash U) as well as a Shared Instrumentation Grant (1S10OD02349701, principal investigator T. C. Doyle, at Stanford University).

**Author contributions** M.E.B., W.C.K., C.B.H., T.H.A. and H.A.I. conceived and designed the experiments, interpreted the results and wrote the paper. M.E.B. designed and conducted ex vivo bone assays and in vivo treatments with drugs and HFD. W.C.K. performed all brain histology, stereotaxic surgery, viral CCN3 injections and lactation analyses with SD and LCD. K.C. conducted femoral fractures as well as bone histology and histomorphometry analyses on bone following viral CCN3 hepatic transduction. C.B.H. conducted ex vivo whole bone assays, parabiosis studies with S.V., and carried out whole bone transplants and ocSSC culturing with T.H.A. T.H.A. analysed all bone measurements except for mice treated with SD and HFD and performed scRNA-seq experiments and SSC assays. Z.T. generated confocal images from fluorescent in situ hybridization analyses of brain sections. J.N. performed analyses on bulk RNA-seq datasets. R.R. performed stereotaxic surgery of isolated ocSSCs into the brain. F.C.-N. conducted dynamic histomorphometry analyses. E.E.W. and J.K.L. manufactured slow-release gels for fracture repair studies. Y.W. provided technical support on multiple aspects of this project. X.Z. and E.L.S. determined BMAT levels in mouse models. N.E.L. and C.K.F.C. provided guidance on the project, including bone expertise and bone stem cell niches, and edited the manuscript.

**Competing interests** The authors declare no competing interests.

**Additional information**
**Correspondence and requests for materials** should be addressed to Thomas H. Ambrosi or Holly A. Ingraham.

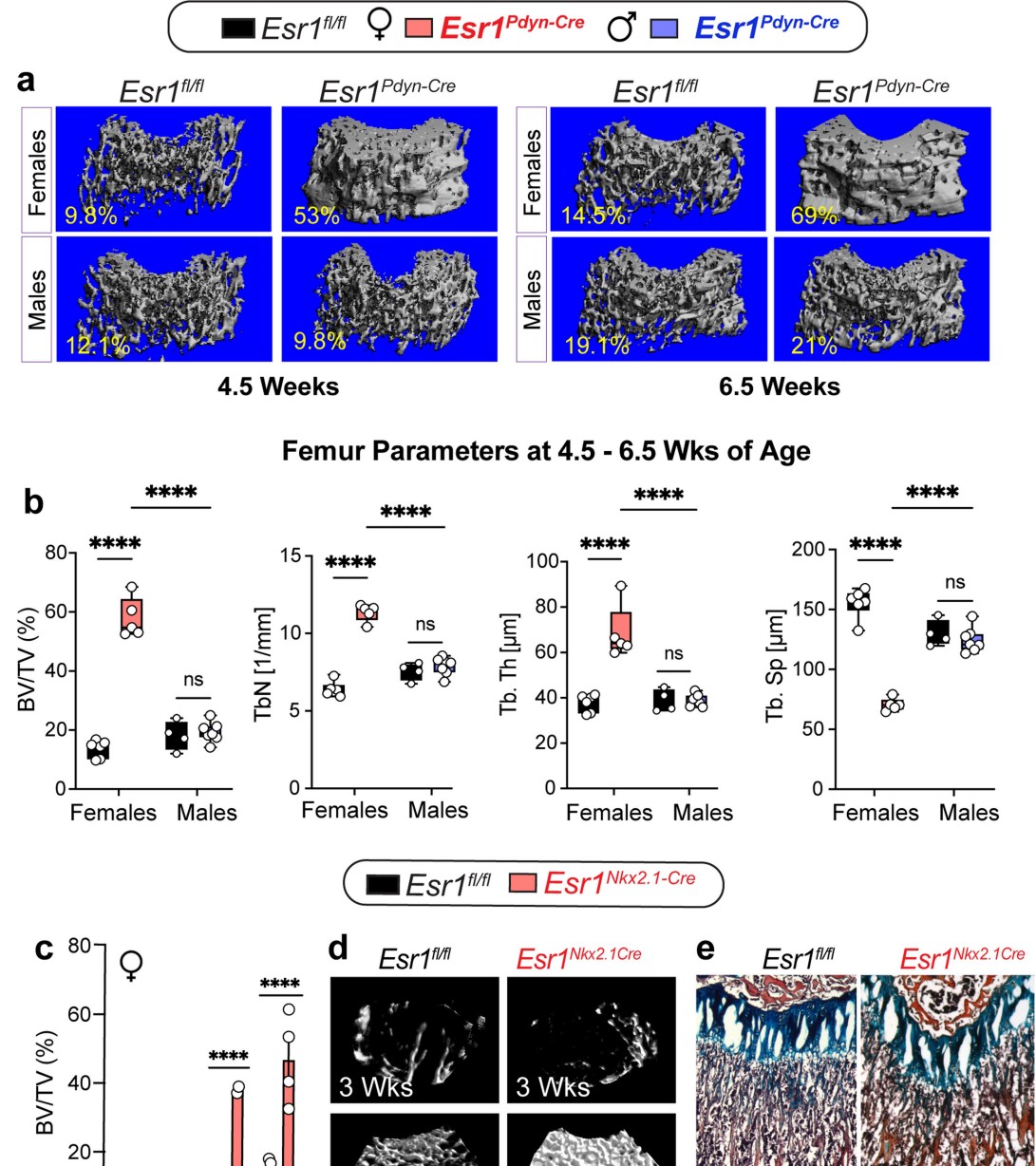

**Esr1**$^{fl/fl}$ ♀ **Esr1**$^{Pdyn-Cre}$ ♂ **Esr1**$^{Pdyn-Cre}$

**a**

Females / Males

Esr1$^{fl/fl}$ — Esr1$^{Pdyn-Cre}$ — Esr1$^{fl/fl}$ — Esr1$^{Pdyn-Cre}$

9.8% / 53% / 14.5% / 69%

12.1% / 9.8% / 19.1% / 21%

**4.5 Weeks** | **6.5 Weeks**

**Femur Parameters at 4.5 - 6.5 Wks of Age**

**b** BV/TV (%) — TbN [1/mm] — Tb. Th [µm] — Tb. Sp [µm]

**Esr1**$^{fl/fl}$ — **Esr1**$^{Nkx2.1-Cre}$

**c** BV/TV (%), Age (wks): 1, 3, 4, 4.5 ♀

**d** Esr1$^{fl/fl}$ — Esr1$^{Nkx2.1Cre}$ — 3 Wks / 3 Wks — 4 Wks / 4 Wks

**e** Esr1$^{fl/fl}$ — Esr1$^{Nkx2.1Cre}$ — 4 Wks

**Extended Data Fig. 1 | Sex-Dependent High Bone Mass in Genetic Models that Target KNDy ARC Neurons Occurs By 4 Weeks of Age. a**, Representative µCT images of female and male distal femurs at 4.5 and 6.5 weeks. **b**, Box and whisker plots of structural bone parameters of control *Esr1*$^{fl/fl}$ and mutant *Esr1*$^{Pdyn-Cre}$ (mutant) females (red) and males (blue); legend on top. **c**, Time course of high bone mass in *Esr1*$^{fl/fl}$ (black) or *Esr1*$^{Nkx2.1-Cre}$ (red) females beginning at 1 week of age, data for the 4.5-week time point are taken from[2] and re-graphed,

(N = 4 for all groups except N = 2 for mutant at 4 wks), legend on top. **d**, µCT imaging of females at 3 and 4 weeks of age. **e**, Modified Pentachrome staining of sections from control and mutant distal femur 4 weeks of age showing enhanced mineralized bone (red) in the mutant femur. One-Way ANOVA for panel b (Šidák's multiple-comparisons test). Two-Way ANOVA for panel c, $F_{(1, 21)}$ = 65.63, $P < 0.0001$. ****p < 0.0001, ns = not significant. Error Bars ± SEM.

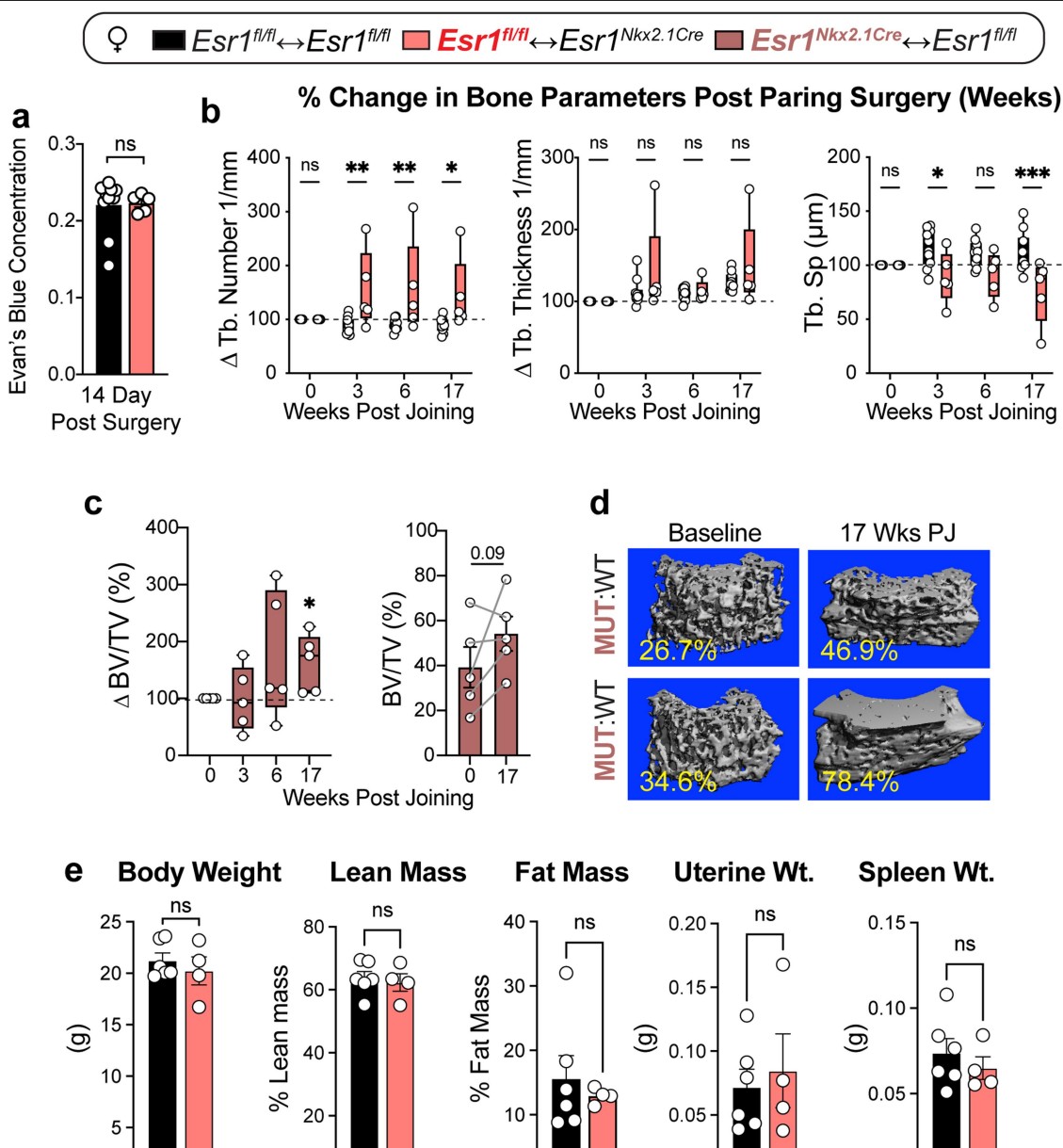

**Extended Data Fig. 2 | Increases in Trabecular Bone in WT:MUT and MUT:WT Parabiosis Without Weight Changes in Whole Body or Other Tissues. a**, Bar graph of Evan's Blue concentration in blood injected into Control or Mutant female mice 14 days post-surgery to assess pairing. **b**, Box and whisker plots of percent changes in structural bone parameters of the *Esr1^fl/fl^* distal femur after WT:MUT pairing for the time indicated as determined by in vivo μCT imaging. **c**, Per cent change (left panel) and fractional bone volume (%BV/TV. right panel) in *Esr1^Nkx2.1Cre^* femurs (N = 5) in MUT:WT parabionts, as determined by in vivo

μCT scans. **d**, Representative μCT images of *Esr1^Nkx2.1Cre^* distal femur from two different MUT:WT pairings. Legend is shown on top. Unpaired Student T-Test 2-tailed for panel a. **e**, Body weights and other tissue measurements obtained 17 weeks post-joining after euthanasia (N = 6, 4). Two- and One-Way ANOVA for repeated measures for panels in b and c (left), respectively (Šidák's multiple-comparisons test). Ratio Paired Student T-Test for panels a, c (right) and e. *p < 0.05, **p < 0.01, ***p < 0.001, ns = not significant. Error Bars ± SEM.

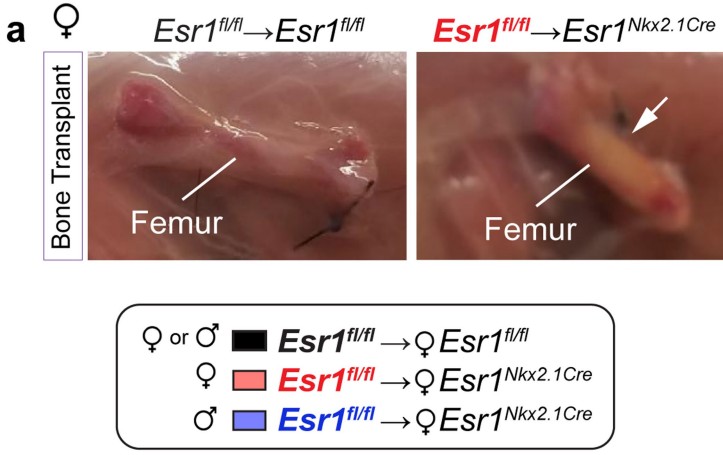

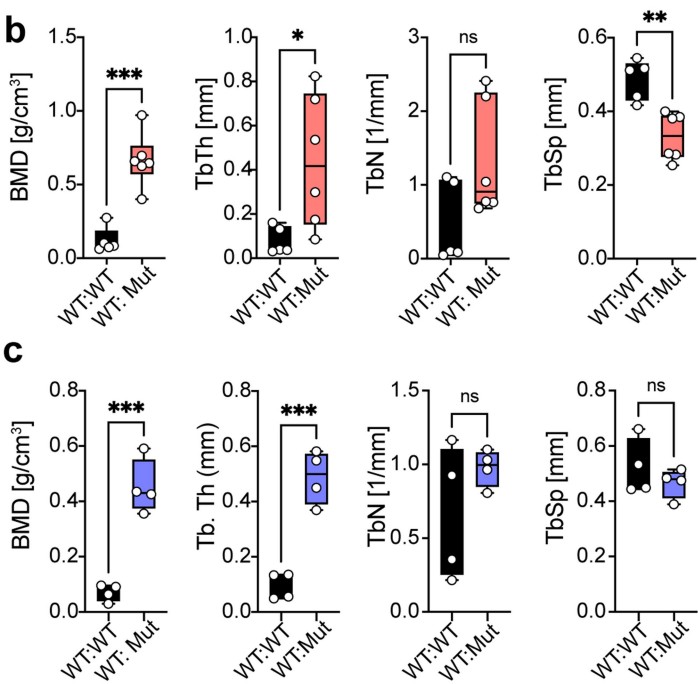

**Extended Data Fig. 3 | Higher Bone Mass of Wild-type Femurs Transplanted into Mutant *Esr1^{Nkx2.1-Cre}* Females. a**, Images of wild-type female bones 6-weeks post-implantation into control or mutant females. **b**, Box and whisker plots of μCT structural parameters of wild-type female femurs into *Esr1^{fl/fl}* (black) or *Esr1^{Nkx2.1-Cre}* (red, N = 5, 6) females 6-weeks post-implantation. **c**, Box and whisker

plots of μCT structural parameters of wild-type male bones into *Esr1^{fl/fl}* (black) or *Esr1^{Nkx2.1-Cre}* (blue, N = 4, 4) females 6-weeks post-implantation. Unpaired Student T-Test 2 tailed for panels in b and c. *p < 0.05, **p < 0.01, ***p < 0.001, ns = not significant. Legends to plots on top.

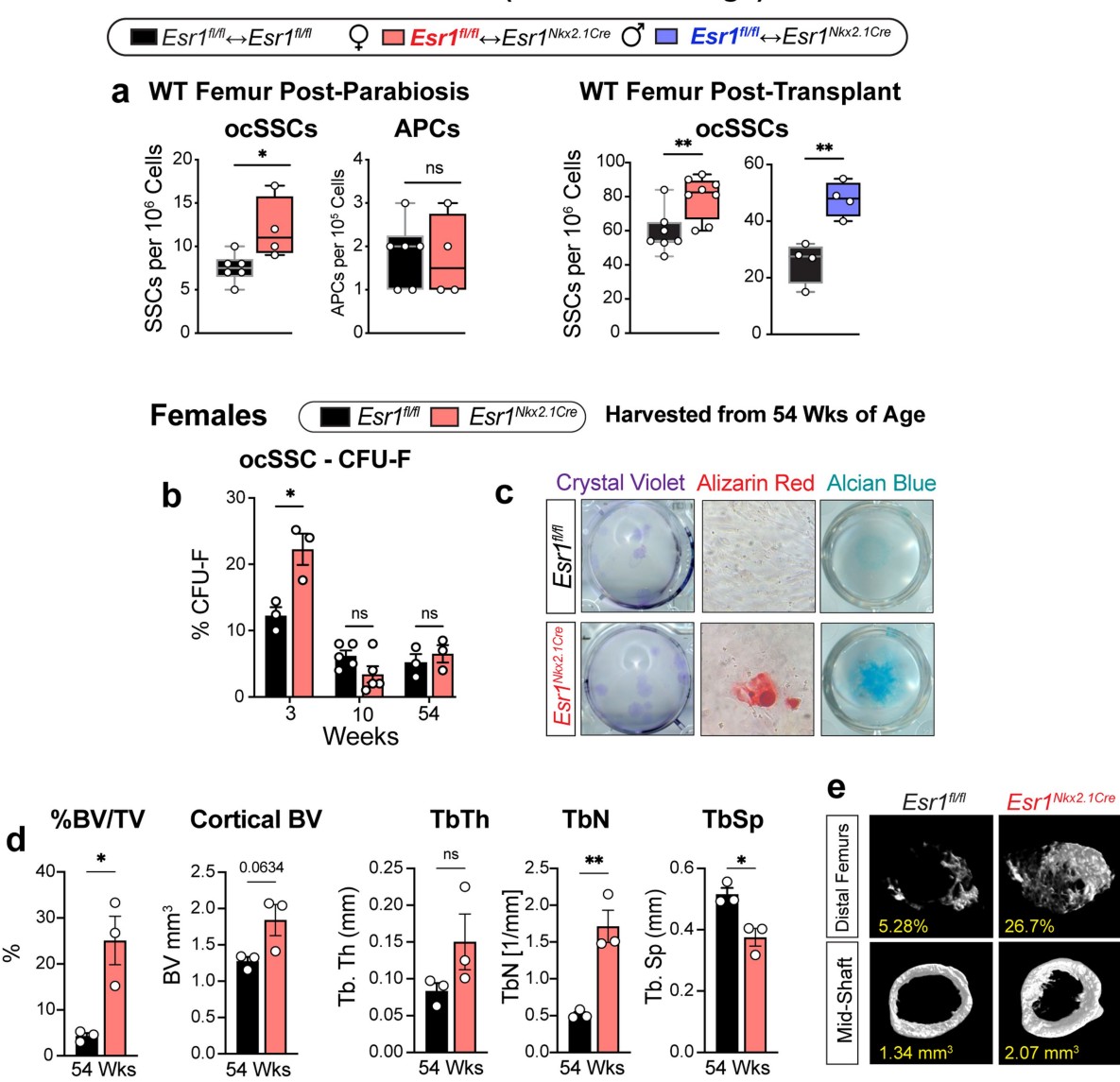

**Female and Males (14-21 Wks of Age)**

$Esr1^{fl/fl} \leftrightarrow Esr1^{fl/fl}$ ♀ $Esr1^{fl/fl} \leftrightarrow Esr1^{Nkx2.1Cre}$ ♂ $Esr1^{fl/fl} \leftrightarrow Esr1^{Nkx2.1Cre}$

**a** **WT Femur Post-Parabiosis**

**ocSSCs** **APCs**

**WT Femur Post-Transplant**

**ocSSCs**

**Females** $Esr1^{fl/fl}$ $Esr1^{Nkx2.1Cre}$ **Harvested from 54 Wks of Age**

**ocSSC - CFU-F**

**b**

**c** Crystal Violet  Alizarin Red  Alcian Blue

$Esr1^{fl/fl}$

$Esr1^{Nkx2.1Cre}$

**d** **%BV/TV**  **Cortical BV**  **TbTh**  **TbN**  **TbSp**

**e** $Esr1^{fl/fl}$  $Esr1^{Nkx2.1Cre}$

Distal Femurs — 5.28% / 26.7%

Mid-Shaft — 1.34 mm³ / 2.07 mm³

**Extended Data Fig. 4 | Increased ocSSCs from Control Bones After Sharing Circulation with Mutant $Esr1^{Nkx2.1-Cre}$ Females, who Exhibit Youthful ocSSCs and Bone Mass. a**, Box and whisker plots of live cells obtained following FACS-purification as described in Methods isolated from control femurs obtained from WT:WT or WT:MUT parabionts (N = 6, 4) or from control female (red, N = 6, 7) or male (blue, N = 4, 4) femurs transplanted into $Esr1^{Nkx2.1-Cre}$ females as indicted by legend on top. **b**, Bar graphs of CFU-F from ocSSCs purified from control or mutant female long bones at age indicated (N = 3–6). **c**, Functional in vitro assays of ocSSCs from 54-week-old females in defined media and stained with dye as indicated at the top of representative images of culture wells (n = 3-4 replicates); Crystal Violet (CFU-F), Alizarin Red (osteogenesis), and Alcian Blue (chondrogenesis). **d**, Fractional bone volume of trabecular and cortical bone as well as other trabecular parameters obtained from μCT scanning of femurs from aged females (≥ 52 weeks of age) of distal and midshaft regions (N = 3, 3). **e**, Representative images of μCT-scans. Unpaired Student T-Test 2-tailed for panels a-c. *p < 0.05, **p < 0.01, ns = not significant. Error Bars ± SEM.

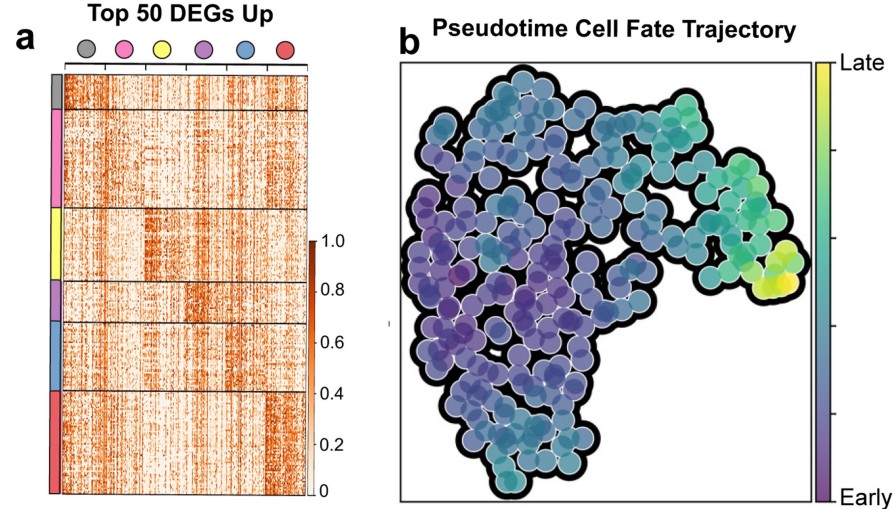

**a**, Top 50 DEGs Up

**b**, Pseudotime Cell Fate Trajectory

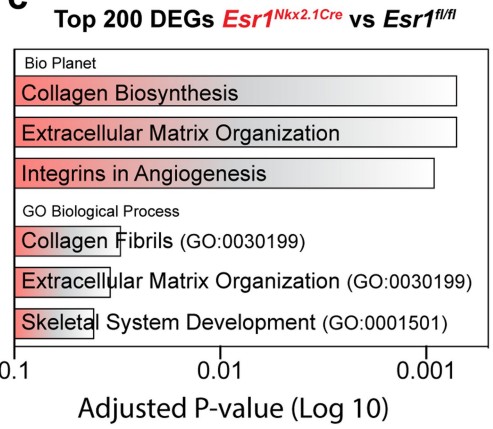

**c**, Top 200 DEGs *Esr1^Nkx2.1Cre* vs *Esr1^fl/fl*

Bio Planet
- Collagen Biosynthesis
- Extracellular Matrix Organization
- Integrins in Angiogenesis

GO Biological Process
- Collagen Fibrils (GO:0030199)
- Extracellular Matrix Organization (GO:0030199)
- Skeletal System Development (GO:0001501)

Adjusted P-value (Log 10)

**Extended Data Fig. 5 | ScRNA-Sequencing of Prospectively Isolated Mouse OcSSCs. a**, Heatmap of top fifty upregulated genes per cluster of isolated mouse ocSSCs. **b**, PAGA pseudo time cell maturation state trajectory inference. **c**, BioPlanet 2019 pathway enrichment (top three bars) and GO Biological Process 2023 ontology (bottom three bars) displaying overexpression in mutants versus wild type based on top 200 DEGs.

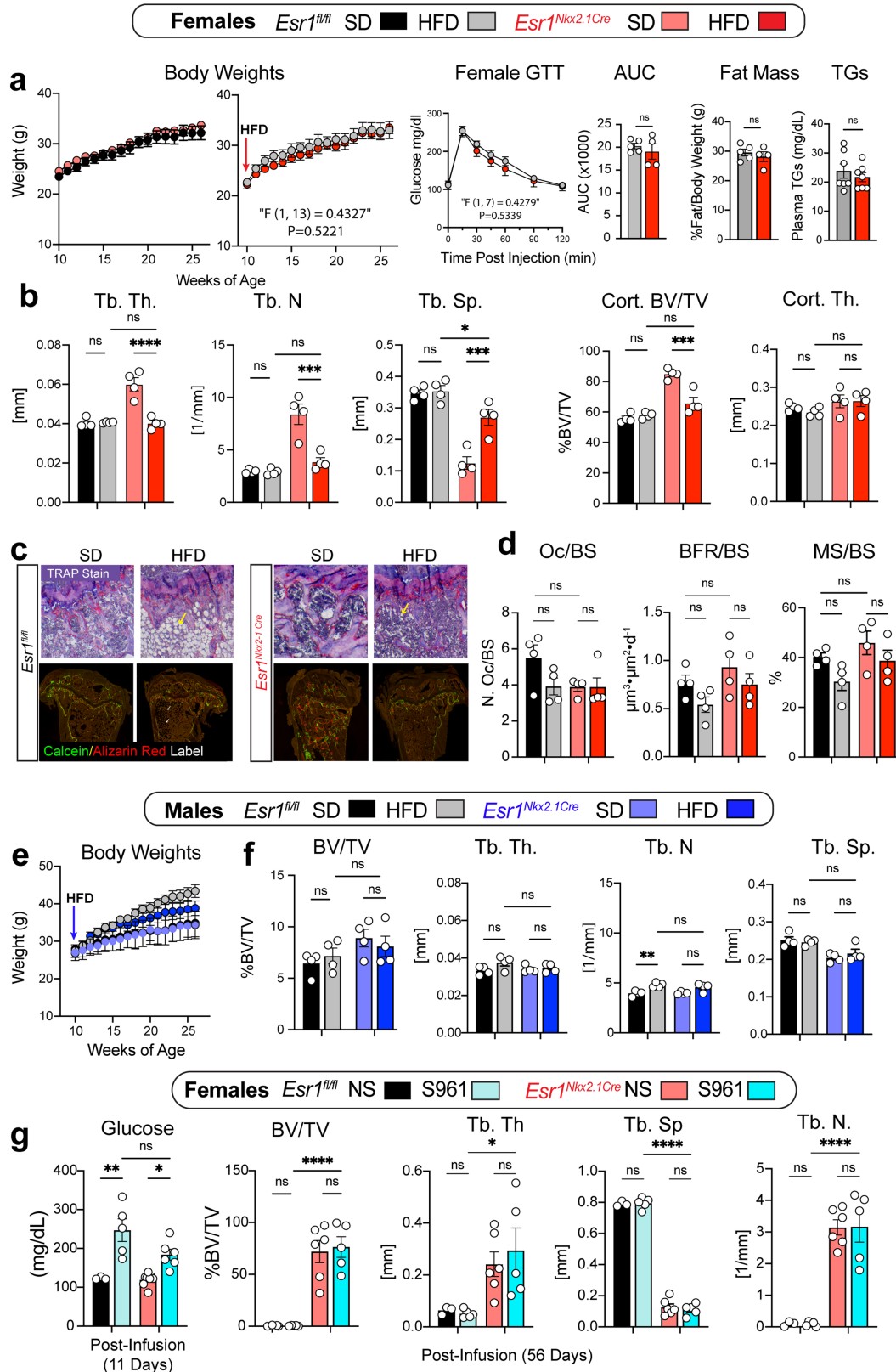

**Extended Data Fig. 6** | See next page for caption.

**Extended Data Fig. 6 | Dietary Challenge Degrades Bone Only in Mutant Females Without Altering Other Metabolic Parameters. a**, XY plot of body weights versus age for *Esr1*$^{fl/fl}$ and *Esr1*$^{Nkx2.1-Cre}$ age-matched female littermates maintained on standard breeder chow (SD) or high-fat diet (HFD) for 17 weeks starting at 10 weeks of age (N ≥ 4 per group). Blood glucose levels after GTT (i.p.), area under the curve (AUC), fat mass by DEXA fed HFD (fed for 12 weeks), and serum triglycerides (fed for 17 weeks) plotted for control and mutant females, legend on top. **b**, Trabecular and cortical bone parameters obtained after μCT scans for four experimental female cohorts, note that fractional bone volume for cortical bone is regraphed from main Fig. 3a. **c**, Representative images of sections of TRAP-stained and double-labeled with Calcein green (green) and Alizarin red (red) femurs for *Esr1*$^{fl/fl}$ and *Esr1*$^{Nkx2.1-Cre}$ cohorts. **d**, Dynamic histomorphometry obtained from tibias for four different experimental cohorts: osteoclasts per bone surface (Oc/BS), bone formation rate/bone surface (BFR/BS), and mineralized surface/bone surface (MS/BS), N = 4 per group. **e**, XY plot of body weights versus age for *Esr1*$^{fl/fl}$ and *Esr1*$^{Nkx2.1-Cre}$ age-matched male experimental cohorts, (N = 4 per group). **f**, Trabecular and cortical bone parameters obtained by μCT imaging for four male experimental cohorts. Legend on top. **g**, Blood glucose and structural bone parameters obtained by μCT imaging for *Esr1*$^{fl/fl}$ and *Esr1*$^{Nkx2.1-Cre}$ female cohorts treated with vehicle (N = 3, 6) or S961 (N = 5, 5) delivered by implanted osmotic pumps at 17.5 nM/week over a period of 8 weeks (N = 5-6 per group). Two-way ANOVA for repeated measures for panels in a and e (BW curve and GTT), respectively (Šidák's multiple-comparisons test). One-way ANOVA for panels b, d, f, and g (Šidák's multiple-comparisons test), Unpaired Student's T-test 2-tailed for three right-hand panels in a. *p < 0.05, **p < 0.01, ***p < 0.001, ***p < 0.0001, ns = not significant. Error Bars ± SEM.

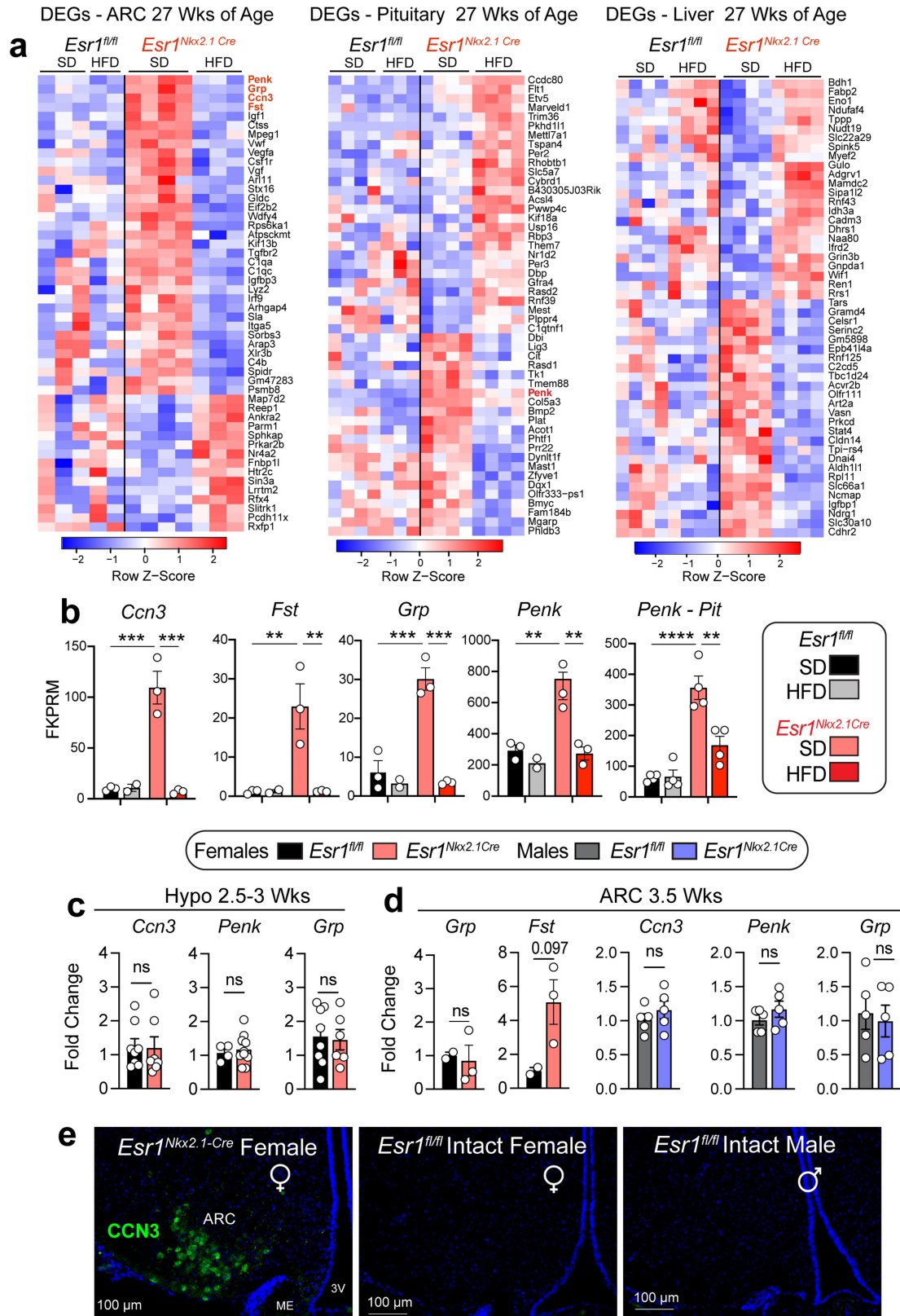

**Extended Data Fig. 7** | See next page for caption.

**Extended Data Fig. 7 | A Cluster of ARC DEGs Correlates with Changing Bone Mass in Female _Esr1$^{Nkx2.1-Cre}$_ Mutants. a**, Heatmaps of top 50 DEGs listed to the right following analyses of bulk RNA-Seq datasets of _Esr1$^{fl/fl}$_ and _Esr1$^{Nkx2.1-Cre}$_ age-matched female littermates maintained on standard breeder chow (SD) or high-fat diet (HFD) for 17 weeks starting at 10 weeks of age; samples include microdissected ARC (left panel), whole pituitary glands (middle panel) or liver tissue (right panel). The cluster of secreted proteins/peptides for the ARC attenuated by HFD are highlighted in red text. Legend for each heatmap shows relative Z-Scores. **b**, Normalized reads for candidate genes from the ARC; _Penk_ in pituitary with either SD or HFD (N = 2–4). **c**, Relative expression of transcripts as listed in the female hypothalamus at 2.5 weeks of age shown in bar graphs with individual points (N = 4–8). **d**, Relative expression of transcripts in microdissected ARC harvested from control _Esr1$^{fl/fl}$_ and mutant _Esr1$^{Nkx2.1-Cre}$_ age-matched females (red) or males (blue), (N = 2–5). **e**, CCN3 (green) expression in the posterior ARC of mutant female, control virgin female and control intact male. Scale bars = 100 μm. One-Way ANOVA for panel b, Unpaired Student's T-test 2-tailed for panels c and d, **p < 0.01, ***p < 0.001, ****p < 0.0001, ns = not significant. ns = not significant. Error Bars ± SEM. Abbreviations: ARC arcuate nucleus. ME median eminence.

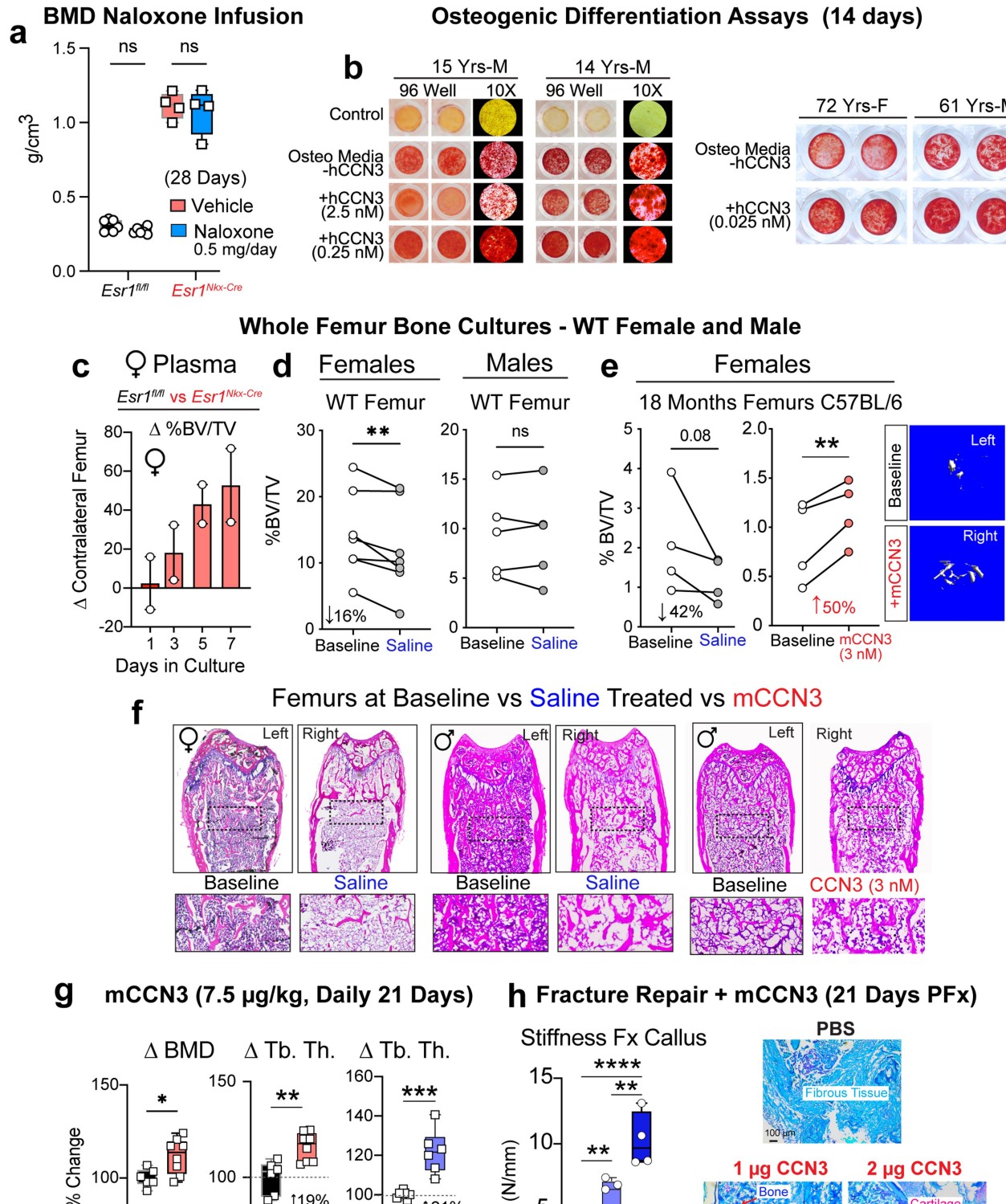

**Extended Data Fig. 8** | See next page for caption.

**Extended Data Fig. 8 | CCN3 Increases Osteogenesis in Human ocSSCs and Bone Mass in Vitro, in Vivo and in Fracture Repair. a**, Effects of chronic infusion of Naloxone over 28 days with fractional bone volume plotted for control *Esr1*$^{fl/fl}$ and mutant *Esr1*$^{Nkx2.1-Cre}$ age-matched females. The ages of female mice at the beginning of treatment were 10–12 weeks of age, which was delivered via an implanted mini-osmotic pump (0.5 mg/24 hrs) over 28 days. Legend in bar graph (N = 4 per group). **b**, Representative images of duplicate wells of Alizarin staining in Control media, osteogenic media minus or plus different doses of human CCN3 with magnified images of one well in far-right images of each panel. Representative images of duplicate wells of Alizarin staining in culture wells with osteogenic defined media minus or plus human CCN3. Some images from panels c and d are duplicated from Main Fig. 4a,c. **c**, Bar plots of change in fractional bone volume from whole femurs harvested from control females and then cultured with isolated plasma from *Esr1*$^{fl/fl}$ and *Esr1*$^{Nkx2.1-Cre}$ age-matched female littermates. Plasma (15 µl) was added daily for 1–7 days of culture as described in the Methods Section. **d**, Plots of fractional bone volume were determined after culturing the right femur (females) or right femur (males) in media treated with 0.9 % NS (Saline). Baseline values were obtained for freshly isolated left femur from the same mouse and immediately fixed in 4% PFA for analysis without culturing (Baseline). **e**. Plots of fractional bone volume were determined after culturing the right femur from 18-month-old C57BL/6 female mice in media treated with 0.9 % NS (Saline) or 3 nM mCCN3 compared to baseline. **f**, Representative images of H&E stained sections of the contralateral left and right femurs from the same female and male mouse at Baseline, Saline, or after treatment with mCCN3. **g**, Box and whiskers plots of bone parameters after saline (black) or mCCN3 (red) daily treatments of control females. **h**. Stiffness of callus 21 days post-fracture with images from Modified Periodic Acid-Schiff (PAS) staining shown for callus region. One Way ANOVA for panels a and h (Tukey's multiple-comparisons test). Paired Student's T-test 1-tailed for panels d and e. Unpaired Student's T-test 2-tailed for panel g. *p < 0.05, **p < 0.01, ns = not significant. Error Bars ± SEM. *p < 0.05, ***p < 0.001, ****p < 0.0001, ns = not significant. Error Bars ± SEM.

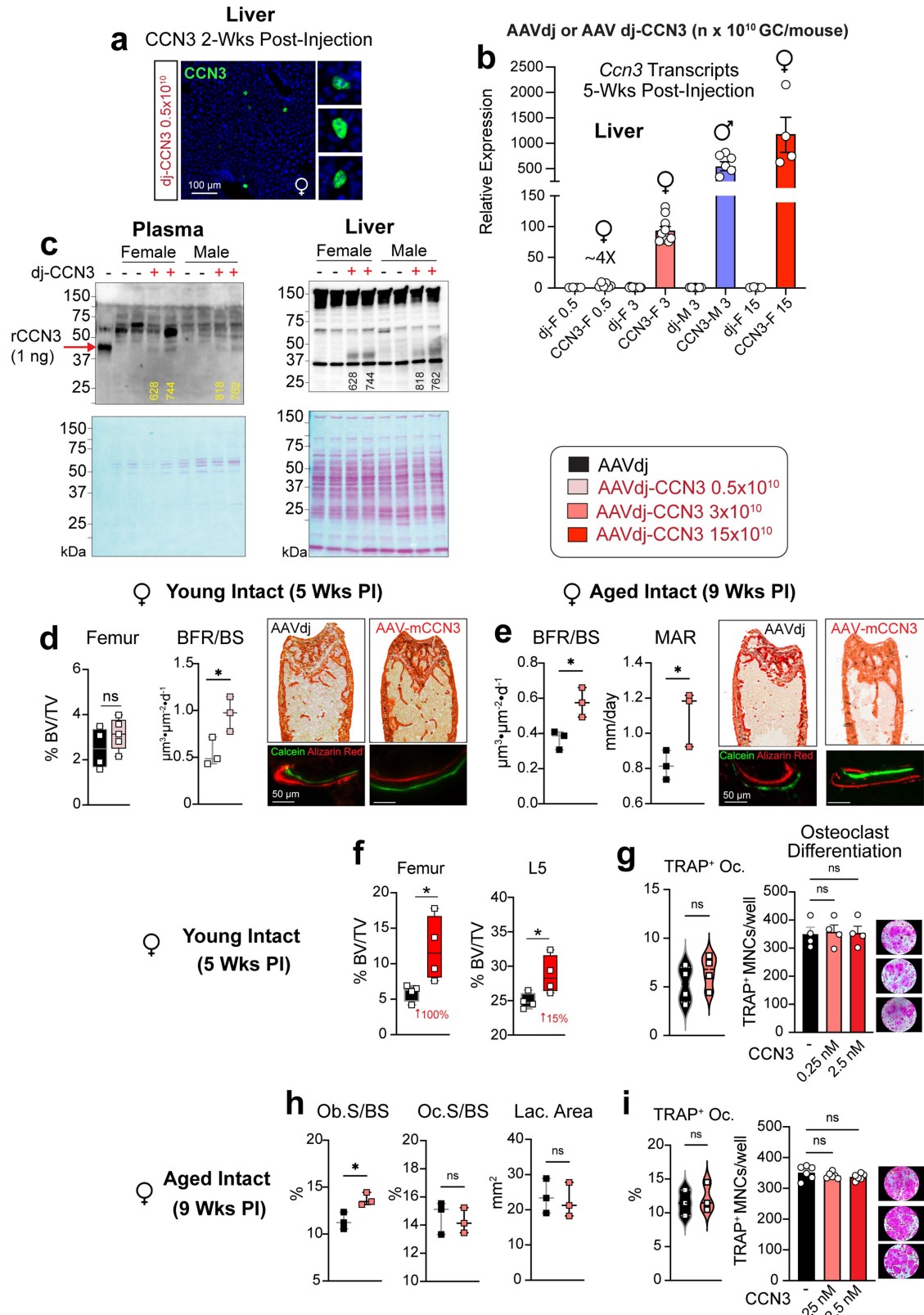

**Extended Data Fig. 9** | See next page for caption.

**Extended Data Fig. 9 | Ectopic Hepatic mCCN3 Expression in Control *Esr1*$^{fl/fl}$ Females Increase Bone Formation Without Affecting Bone Resorption.**
**a**, Expression of mCCN3 protein in female liver transduced with a low dose of AAVdj-CAG-*Ccn3* (0.5*10$^{10}$ GC/mouse) 2 weeks post-injection; panels to the right represent digitally magnified images of individual positive cells. Scale bar = 100 μm. **b**, Relative levels of *Ccn3* transcripts in liver tissue 5 weeks post-injection after transduction of 0.5*10$^{10}$, 3*10$^{10}$ and 15*10$^{10}$ GC/mouse of AAVdj-CAG-*Ccn3* viral vector and control AAV-empty vector (dj) into *Esr1*$^{fl/fl}$ male or female littermates, (N = 4–9 per group). **c**, Western blots of plasma and liver uncropped with Ponceau staining below used in Main Fig. 5d. The relative expression for *Ccn3* by qPCR is listed for each sample in lanes. **d**, Bone volume and dynamic histomorphometry measurements after Calcein and Alizarin red double labeling (5 days apart) of female mice transduced with the lowest dose of AAVdj-CAG-*Ccn3* viral vector compared to control vector (black) obtained in femurs from *Esr1*$^{fl/fl}$ control females. **e**, Bone formation rate and mineral apposition rate (MAR) from dynamic histomorphometry measurements with representative images of femur sections described above from aged *Esr1*$^{fl/fl}$ females (20–23 months of age) injected with AAVdj-CAG-CCN3. **f**, Bone volume of femur and L5 of female mice transduced with highest dose of AAVdj-CAG-*Ccn3* viral vector. **g**, TRAP+ osteocytes (%) in femurs from young females (left). In vitro differentiation of osteoclasts from bone marrow isolated from young intact females treated with vehicle or recombinant CCN3 (right). **h**, Osteoblast surface (Ob.S), osteoclast surface (Oc.S) per bone surface (BS), and silver nitrate staining for lacunae. **i**, TRAP+ osteocytes (%) in femurs from aged females (left). In vitro differentiation of isolated osteoclasts from aged intact females treated with vehicle or recombinant CCN3 (right). Unpaired Student's T test 2-tailed for low and high dose groups in panels e, d, g, h. *p < 0.05. ns = not significant. Error Bars ± SEM. Legend is provided above graphs. Abbreviations: MNCs mononucleated cells.

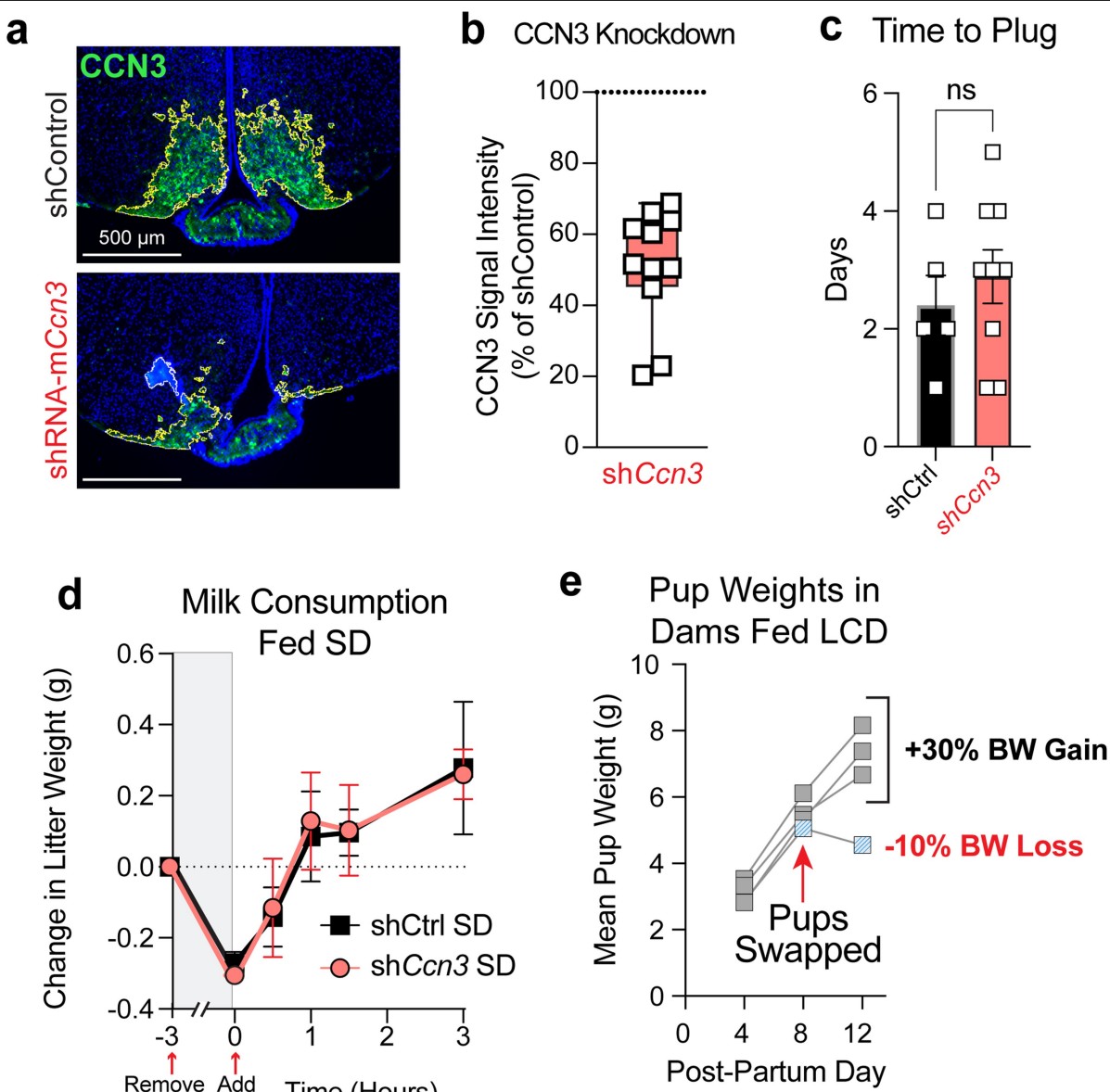

**Extended Data Fig. 10 | Sh-RNA-M*ccn3* Knockdown Does Not Impair the Fertility or Milk Provision of Female Mothers, But Limits Pup Growth.**
**a**, Representative images of brain sections and CCN3 staining quantification from dams injected with shControl (upper) or shRNA-m*Ccn3* (lower) and collected at 12DPP. Yellow border defines area in which CCN3 immunostaining intensity was quantified. Scale bars = 500 μm. **b**, Percent reduction in ARC CCN3 immunostaining intensity of shRNA-m*Ccn3* dams fed SD (N = 9) or LCD (N = 2) as compared to shRNA control dams fed SD (N = 5). **c**, The time interval

between mating (day 0) and observation of copulatory plug (Control N = 5, shRNA-m*Ccn3* N = 9). **d**, Milk consumption in litters from shCtrl (N = 3) and sh*Ccn3* (N = 9) as measured by weight recovery following 3-hour separation from lactating dams fed SD. **e**, Mean body weights for litters (N = 4 litters, N = 6 pups/litter) nursed by shCtrl mothers at 4, 8, and 12DPP (grey) or switched to an sh*Ccn3* mother beginning at 8DPP (blue and grey). Unpaired 2-tailed Student's T-test in panel c. Two-way ANOVA with repeated measures (Holm-Šidák's multiple-comparisons test) in panel d. Error Bars ± SEM.

# Reporting Summary

## Statistics

For all statistical analyses, confirm that the following items are present in the figure legend, table legend, main text, or Methods section.

| n/a | Confirmed | |
|---|---|---|
| ☐ | ☒ | The exact sample size (*n*) for each experimental group/condition, given as a discrete number and unit of measurement |
| ☐ | ☒ | A statement on whether measurements were taken from distinct samples or whether the same sample was measured repeatedly |
| ☐ | ☒ | The statistical test(s) used AND whether they are one- or two-sided<br>*Only common tests should be described solely by name; describe more complex techniques in the Methods section.* |
| ☐ | ☒ | A description of all covariates tested |
| ☐ | ☒ | A description of any assumptions or corrections, such as tests of normality and adjustment for multiple comparisons |
| ☐ | ☒ | A full description of the statistical parameters including central tendency (e.g. means) or other basic estimates (e.g. regression coefficient) AND variation (e.g. standard deviation) or associated estimates of uncertainty (e.g. confidence intervals) |
| ☐ | ☒ | For null hypothesis testing, the test statistic (e.g. *F*, *t*, *r*) with confidence intervals, effect sizes, degrees of freedom and *P* value noted<br>*Give P values as exact values whenever suitable.* |
| ☒ | ☐ | For Bayesian analysis, information on the choice of priors and Markov chain Monte Carlo settings |
| ☒ | ☐ | For hierarchical and complex designs, identification of the appropriate level for tests and full reporting of outcomes |
| ☒ | ☐ | Estimates of effect sizes (e.g. Cohen's *d*, Pearson's *r*), indicating how they were calculated |

*Our web collection on statistics for biologists contains articles on many of the points above.*

## Software and code

Policy information about availability of computer code

| | |
|---|---|
| Data collection | qPCR data was acquired using the CFX Maestro software (Bio-Rad, v4.1.2433.1219). Bulk RNA-seq reads were sequenced on Illumina's NovaSeq 6000, S4 flow cell. For histomorphometry, mounted sections were imaged with an ECHO REVOLVE R4 using FITC (Calcein) and TexasRed (Alizarin Red) channels by the Wash U Bone Core. Mosaic-tiled images of distal femurs were acquired at ×20 magnification with a Zeiss Axioplan Imager M1 microscope (Carl Zeiss MicroImaging) fitted with a motorized stage. The tiled images were stitched and converted to a single image using the Axiovision software (Carl Zeiss MicroImaGing). For SSCs MBH brain transplants, images of histological sections were taken using the Keyence B2-X800. For kidney transplants, brightfield images were taken using a Luminera Infinity-3 and quantified using ImageJ software. In vivo micro (μ) CT was performed using a Scanco Viva CT40 high-speed μCT preclinical scanner. For ex-vivo μCT, a Scanco Medical μCT 50 specimen scanner calibrated to a hydroxyapatite phantom or Bruker SkyScan1276 (Bruker Preclinical Imaging) was utilized. Standard best practices were used to quantify trabecular and cortical bone parameters44. Image acquisition of an implanted femur was captured with iPhone 13 Pro. Body composition to determine % lean and fat mass was obtained by dual-energy x-ray (DEXA, GE Lunar PIXImus). Femurs underwent a three-point bend test using the Instron E100 mechanical load frame. For brain RNAscope and immunohistochemistry, confocal images were acquired at the UCSF Nikon Imaging Center using a Nikon CSU-22 with an EMCCD camera and MicroManager v2.0gamma. for single-cell RNA-sequencing of ocSSCs, pooled libraries were sequenced on NovaSeq6000 (Illumina) to obtain 1–2 million 2 x 151 base-pair paired-end reads per cell. Flow cytometry and cell sorting were performed on a FACS Aria II cell sorter (BD Biosciences). |
| Data analysis | For histomorphometry, photoshop software removed the background in non-tissue areas for images of the proximal tibias. Blinded analyses were performed using two image-analysis software programs, Bioquant OSTEO software (Nashville, TN, USA) or ImageJ software. Images taken with iPhone 13 Pro were edited in Photoshop CC. For HFD experiment, Volumes of interest were evaluated using Scanco evaluation software. Representative 3D images were created using Scanco Medical mCT Ray v4.0 software. For the whole femur bone cultures, naloxone and S961 experiments, ccnn3 injections into WT mice, GOF CCN3 hepatic expression and LOF Ccn3 knowdown in ARC, reconstructed |

samples were analyzed using CT Analyser and CTvox software (Bruker). Confocal images were processed and quantified using ImageJ FIJI v1.52i and the Cell Counter plugin v2. For all bulk RNAseq samples, sequencing-generated reads were aligned to the mouse transcriptome (mm10) using Kallisto in gene mode47. Differential gene expression was evaluated using the likelihood-ratio test by Sleuth (qval <0.05)48. All heatmaps were generated with the top 50 female/male-biased genes obtained from 27-week-old mice and were generated in R49. For single cell RNAseq sequencing, sequenced data were demultiplexed using bcl2fastq2 2.18 (Illumina). Raw reads were further processed using a skewer for 3' quality trimming, 3' adaptor trimming, and removal of degenerate reads. Trimmed reads were then mapped to the mouse genome vM20 using STAR 2.4, and counts for gene and transcript reads were calculated using RSEM 1.2.21. Data were explored, and plots were generated using Scanpy v1.9. To select high-quality cells only, we excluded cells with fewer than 450 genes and genes detected in less than three cells were excluded. Cells with a mitochondrial gene content higher than 5%, ERCC content higher than 30%, and ribosomal gene content higher than 5% were excluded as well. Scrublet was then used to detect and remove residual duplicates. A total of 264 high-quality cells (122 control and 142 mutant mouse cells) were included in the final analysis. Raw counts per million (CPM) values were mean- and log-normalized, and then data were scaled to a maximum value of 10. Combat batch correction was applied to account for potential biases through minor differences in cell processing. Principal component (PC) 'elbow' heuristics were used to determine the number of PCs for clustering analysis with UMAP and Leiden Algorithm (leidenalg). Differential gene expression between Esr1fl/fl (wild type) and Esr1Nkx2-1Cre (mutant), as well as Leiden clusters, was calculated by the Wilcoxon-Rank-Sum test. Cell cycle status was assessed using the 'score_genes_cell_cycle' function with the updated gene list provided by Nestorowa et al.51. EnrichR was used to explore enrichment for pathways and ontologies of differentially expressed genes between wild-type and mutant groups52. Statistical tests, excluding RNA-Seq analyses, were performed using Prism 10 (GraphPad). Flow cytometry and cell sorting were analyzed using FlowJo software.

For manuscripts utilizing custom algorithms or software that are central to the research but not yet described in published literature, software must be made available to editors and reviewers. We strongly encourage code deposition in a community repository (e.g. GitHub). See the Nature Portfolio guidelines for submitting code & software for further information.

# Data

Policy information about availability of data

All manuscripts must include a data availability statement. This statement should provide the following information, where applicable:
- Accession codes, unique identifiers, or web links for publicly available datasets
- A description of any restrictions on data availability
- For clinical datasets or third party data, please ensure that the statement adheres to our policy

A data availability statement is included in the manuscript. All data generated or analyzed during this study will be included in the published article (and its supplementary information files). All raw data and processed data files for the bulk-RNA and sc-RNA Sequencing are publicly available at GEO under sample accession numbers GSE248882 and GSE241478.

# Research involving human participants, their data, or biological material

Policy information about studies with human participants or human data. See also policy information about sex, gender (identity/presentation), and sexual orientation and race, ethnicity and racism.

| Reporting on sex and gender | Male and Female |
|---|---|
| Reporting on race, ethnicity, or other socially relevant groupings | NA |
| Population characteristics | Different Age Groups and Different Biological Sex |
| Recruitment | NA |
| Ethics oversight | UC Davis IRB ID: 1997852-3, waived |

Note that full information on the approval of the study protocol must also be provided in the manuscript.

# Field-specific reporting

Please select the one below that is the best fit for your research. If you are not sure, read the appropriate sections before making your selection.

☒ Life sciences   ☐ Behavioural & social sciences   ☐ Ecological, evolutionary & environmental sciences

For a reference copy of the document with all sections, see nature.com/documents/nr-reporting-summary-flat.pdf

# Life sciences study design

All studies must disclose on these points even when the disclosure is negative.

| Sample size | Sample sizes were based on previous work from our lab and others (Correa et al., 2015, Herber et al., 2019, van Veen et al,. 2020, Ambrosi et al., 2021). For other studies statistical calculation was performed to determine sample size using open source software (G-power etc). |
|---|---|
| Data exclusions | Animals that unexpectedly became morbid during the course of parabiosis were excluded. |

| | |
|---|---|
| Data exclusions | Only animals that survived to final timepoints of experiments were included. During processing of single cell genomic data filtering of low quality cells was applied as described. |
| Replication | All data presented are biological replicates unless otherwise stated in the figure legends. Each experimental finding was reproduced in at least two independent experiments or at different concentrations, with the exception of parabiosis. |
| Randomization | Mice were drawn at random from a pool of littermate mice containing a roughly equal mix of Cre+ and Cregenotypes. Partitioning into control and experimental groups was determined by genotype. For experiments involving repeated measurements, a randomized balanced design was implemented such that a mix of control and experimental mice housed in the same cage received identical treatments during each trial. |
| Blinding | Measurements of micro CT  bone parameters, osmium stain, histology, histomorphometry, DEXA, qPCR, bulk/single RNA-Seq data and glucose tolerance test and plasma triglyceride assay were made by experimenters blinded to the genotype and treatment. |

# Reporting for specific materials, systems and methods

We require information from authors about some types of materials, experimental systems and methods used in many studies. Here, indicate whether each material, system or method listed is relevant to your study. If you are not sure if a list item applies to your research, read the appropriate section before selecting a response.

## Materials & experimental systems

| n/a | Involved in the study |
|---|---|
| ☐ | ☒ Antibodies |
| ☒ | ☐ Eukaryotic cell lines |
| ☒ | ☐ Palaeontology and archaeology |
| ☐ | ☒ Animals and other organisms |
| ☒ | ☐ Clinical data |
| ☒ | ☐ Dual use research of concern |
| ☒ | ☐ Plants |

## Methods

| n/a | Involved in the study |
|---|---|
| ☒ | ☐ ChIP-seq |
| ☐ | ☒ Flow cytometry |
| ☒ | ☐ MRI-based neuroimaging |

## Antibodies

| | |
|---|---|
| Antibodies used | ERa (EMD Millipore, #C1355 polyclonal rabbit, 1:750 dilution). Kiss1 (Abcam, #ab19028 polyclonal rabbit, 1:200 dilution). CCN3 (R&D Systems, #AF1976 polyclonal goat, 1:1000 dilution). HRP-conjugated secondary antibody for CCN3 (Invitrogen, ##A15999, 1:30000).Species-appropriate secondary Alexa Fluor-coupled antibodies (Invitrogen, #A-21447, #A10042, or #A-11055; 1:1000 dilution). For mouse SSC lineages, antibodies used were as follows: CD90.1 (Thermo Fisher, Cat# 47–0900), CD90.2 (Thermo Fisher, Cat#47–0902), CD105 (Thermo Fisher, Cat#13–1051), CD51 (BD Biosciences, Cat#551187), CD200 (Thermo Fisher Cat#MA5-17980), CD45 (BioLegend, Cat#103110), Ter119 (Thermo Fisher, Cat#15–5921), Tie2 (Thermo Fisher, Cat#14–5987) 6C3 (BioLegend, Cat#108312), and Streptavidin PE-Cy7 (Thermo Fisher, Cat#25–4317) as well as Sca-1 (Thermo Fisher, Cat#56-5981), CD45 (Thermo Fisher, Cat#11–0451), CD31 (Thermo Fisher, Cat#12–0311), CD140a (ThermoFisher, Cat#17–1401), CD24 (Thermo Fisher, Cat#47–0242). For human SSC isolation, antibodies used were as follows: CD45 (BioLegend, Cat#304029), CD235a (BioLegend, Cat#306612), CD31 (Thermo Fisher Scientific, Cat#13-0319), CD202b (TIE-2) (BioLegend, Cat#334204), streptavidin APC-AlexaFlour750 (Thermo Fisher, SA1027), CD146 (BioLegend, Cat#342010), PDPN (Thermo Fisher Scientific, Cat#17-9381), CD164 (BioLegend, Cat#324808), and CD73 (BioLegend, Cat#344016). |
| Validation | ERalpha antibody validated by the manufacturer to detect ERalpha in breast cancer cell lines. In addition hypothalamic ERalpha expression was absent in conditional ERalpha knockout mice.<br>Kiss 1 antibody validated by the manufacturer to dected Kiss 1. See https://www.abcam.com/kisspeptin-antibody-ab19028-references.html#top-608.<br>CCN3 antibody validated by the manufacturer to detect CCN3 in conditioned medium of Sf9y82 cells infected with a recom-binant baculovirus expressing CCN3 (NOV) in the sense orientation. Perbal, B. et al. (1999) Proc. Natl. Acad. Sci. USA 96:869.<br>All antibodies were used for flow cytometry and are validated, commercially available products that addtionally have been validated in previously published studies (e.g. PMID: 29748647 & PMID: 15967997).<br>FACS Sorting Antibodies - Mouse Listed Below  All antibodies were used for flow cytometry and are validated, commercially available products that addtionally have been validated in previously published studies (e.g. PMID: 29748647 & PMID: 15967997).<br>Anti-mouse Thy1.1 Cat#: 47-0900), CD90.1 (Thy-1.1) Monoclonal Antibody (HIS51), APC-eFluor 780, eBioscience™  Reported for flow cytometry recognizing mouse CD90.1<br>Anti-mouse Thy1.2 (Cat#: 47-0902), CD90.2 (Thy-1.2) Monoclonal Antibody (53-2.1), APC-eFluor 780, eBioscience™  Reported for flow cytometry recognizing mouse CD90.2<br>Anti-mouse CD105 Cat#: 13-1051), CD105 (Endoglin) Monoclonal Antibody (MJ7/18), Biotin, eBioscience™  Reported for flow cytometry recognizing mouse CD105<br>Anti-mouse CD51 Cat#: 551187), CD51 (Integrin alpha V) Monoclonal Antibody (RMV-7), PE, BD Biosciences,  Reported for flow cytometry recognizing mouse CD51<br>Anti-mouse CD200 Cat#: MA5-17980), CD200 Monoclonal Antibody (OX90), FITC, eBioscience™  Reported for flow cytometry recognizing mouse CD200<br>Anti-mouse Ter119 Cat#: 15-5921), TER-119 Monoclonal Antibody (TER-119), PE-Cyanine5, eBioscience™  Reported for flow cytometry recognizing mouse Ter119<br>Anti-mouse Tie2 Cat#: 14-5987), CD202b (TIE2) Monoclonal Antibody (TEK4), eBioscience™  Reported for flow cytometry recognizing mouse Tie2 |

Anti-mouse 6C3 Cat#: 108312), Ly-51 Antibody Monoclonal Antibody (6C3), Alexa Fluor® 647, BioLegend  Reported for flow cytometry recognizing mouse CD249

Anti-streptavidin-PE-Cy7 Cat# 25–4317), eBioscience™ Streptavidin PE-Cyanine7 Conjugate  Reported for flow cytometry recognizing Biotin

Anti-mouse Sca1 Cat# 56-5981), Ly-6A/E (Sca-1) Monoclonal Antibody (D7), Alexa Fluor™ 700, eBioscience™  Reported for flow cytometry recognizing mouse Ly-6A/E

Anti-mouse CD45 Cat#: 103110), CD45 Monoclonal Antibody (30-F11), PE-Cyanine5, BioLegend  Reported for flow cytometry recognizing mouse CD45

Anti-mouse CD45 Cat#: 11–0451), CD45 Monoclonal Antibody (30-F11), FITC, eBioscience™  Reported for flow cytometry recognizing mouse CD45

Anti-mouse CD31 Cat#: 12-0311), CD31 Monoclonal Antibody (390), PE, eBioscience™  Reported for flow cytometry recognizing mouse CD31

Anti- mouse CD140a Cat#17–1401), CD140a Monoclonal Antibody (APA5), eBioscience™  Reported for flow cytometry recognizing mouse CD140a

Anti-mouse CD24 Cat#47–0242), CD24 Monoclonal Antibody (M1/69), APC-eFluor780, eBioscience™  Reported for flow cytometry recognizing mouse CD24

FACS Sorting Antibodies -Human Listed Below  All antibodies were used for flow cytometry and are validated, commercially available products that addtionally have been validated in previously published studies (e.g. PMID: 29748647 & PMID: 15967997).

Mouse anti-human CD45 Cat#304029), CD45 Monoclonal Antibody (HI30), Pacific Blue™,, BioLegend  Reported for flow cytometry recognizing anti-human CD45

Mouse anti-human CD235ab Cat#306612), CD235ab Monoclonal Antibody (HIR2), Pacific Blue™, BioLegend  Reported for flow cytometry recognizing anti-human CD235a

Mouse anti-human CD31 Cat#13-0319), CD31 Monoclonal Antibody (WM-59), Biotin, eBioscience™  Reported for flow cytometry recognizing anti-human CD31

Anti-streptavidin-APC-AlexaFlour750 Cat#SA1027), Streptavidin, (APC-Alexa Fluor™ 750), Thermo Fisher  Reported for flow cytometry recognizing Biotin

Mouse anti-human CD202b Cat#334204), CD202b Monoclonal Antibody CD202b (33.1), Biotin-linked, BioLegend  Reported for flow cytometry recognizing anti-human CD202B

Mouse anti-human CD146 at#342010), CD46 Monoclonal Antibody (SHM-57), PE-Cy7, Biolegend  Reported for flow cytometry recognizing anti-human CD146

Mouse anti-human PDPN Cat#17-9381), PDPN Monoclonal Antibody (NZ 1.3), APC, Thermo Fisher  Reported for flow cytometry recognizing anti-human PDPN

Mouse anti-human CD164 Cat#324808), CD 164 Monoclonal Antibody (67D2), PE, BioLegend  Reported for flow cytometry recognizing anti-human CD164

Mouse anti-human CD73 Cat#344016), CD73 Monoclonal Antibody (Ad2), FITC, BioLegend  Reported for flow cytometry recognizing anti-human CD73

# Animals and other research organisms

Policy information about studies involving animals; ARRIVE guidelines recommended for reporting animal research, and Sex and Gender in Research

| Laboratory animals | The origin of the Esr1fl/fl allele (official allele: Esr1tm1Sakh) on a 129P2 background and used to generate Esr1Nkx2-1Cre mice are described1 and were maintained on CD-1;129P2 mixed background. Primer sequences used for genotyping are listed in Extended Data Table 1. Esr1Nkx2-1Cre-CAG-Luc,-GFP mice were generated by crossing male mice harboring the CAG-Luc-GFP allele (official allele: L2G85Chco/J) to female mice homozygous for the Esr1fl/fl allele, followed by an additional cross to generate Esr1fl/fl; Nkx2-1Cre; Luc-GFP colony, which was maintained on a mixed FVB/N, CD-1, 129P2, and C57BL/6 genetic background. Esr1ProdynorphinCre mice were generated by crossing homozygous Esr1fl/fl females to Prodynorphin-Cre (B6;129S-Pdyntm1.1(cre)Mjkr/LowlJ, purchased from JAX) males. Unless otherwise noted, mice were maintained on a 12h light/dark cycle with ad libitum access to a standard breeder chow diet (PicoLab 5058; LabDiet, 4kcal% fat, 0.8% Ca2+) and sterile water and housed under controlled and monitored rooms for temperature and humidity with a 12h light/dark cycle. Eighteen-month-old C57BL/6-aged female mice were obtained through the NIA Aged Rodent Colony Program, available to NIA Funded Projects. To create cohorts of ovariectomized (OVX) females, ovariectomy was performed at 4 months of age, followed by 4 weeks of surgical recovery. All animal procedures were performed in accordance with UCSF institutional guidelines under the Ingraham lab IACUC protocol of record. |
|---|---|
| Wild animals | No wild animals used. |
| Reporting on sex | In all studies, sex as a biological factor was considered and for nearly all analyses unless indicated both male and female mice were used. |

| Field-collected samples | No samples collected from the field. |
| --- | --- |
| Ethics oversight | Experiments were approved and performed in accordance with the guidelines of the UCSF Institutional Animal Care Committee (IACUC) or the UCD Animal Ethics Committee, the National Institutes of Health Guide for Care and Use of Laboratory Animals, and recommendations of the International Association for the Study of Pain. |

Note that full information on the approval of the study protocol must also be provided in the manuscript.

## Flow Cytometry

### Plots

Confirm that:

☐ The axis labels state the marker and fluorochrome used (e.g. CD4-FITC).

☐ The axis scales are clearly visible. Include numbers along axes only for bottom left plot of group (a 'group' is an analysis of identical markers).

☐ All plots are contour plots with outliers or pseudocolor plots.

☐ A numerical value for number of cells or percentage (with statistics) is provided.

### Methodology

| Sample preparation | Detailed sample preparation protocol is provided in methods section of the manuscript (PMID: 29748647 & PMID: 15967997). |
| --- | --- |
| Instrument | Flow cytometry was performed on FACS Aria II (BD Biosciences). Gating schemes were established with fluorescence-minus-one (FMO: staining with all fluorophores except one) controls and negative propidium iodide (PI) (Sigma-Aldrich, Cat#P4170) staining (1 mg/ml) was used as a measure for cell viability. |
| Software | FlowJo v10 was used to analyze FACS data. |
| Cell population abundance | Quantification of cell populations are provided as total number of tissue or percentage of reference population as described in the figure and figure legends. |
| Gating strategy | Gating Strategy is provided in previously published manuscripts (PMID: 29748647 & PMID: 15967997). |

☐ Tick this box to confirm that a figure exemplifying the gating strategy is provided in the Supplementary Information.

