## [Peer Review File · Nature]

Manuscript Title: A Maternal Brain Hormone That Builds Bone During Lactation

Reviewer Comments & Author Rebuttals

Reviewer Reports on the Initial Version:

Referees' comments:

Referee #1 (Remarks to the Author):

The manuscript by Babey et al. describes a mechanism by which specific neuron populations within the arcuate nucleus (ARC) of the brain, respond to reduced estrogen levels during post-partum lactation by producing CCN3. CCN3 then acts on osteochondral skeletal stem cells within the bone, which induces osteogenesis and new bone formation in female mice. The authors propose that the transient increase in CCN3 levels during lactation is necessary to maintain bone that is being actively being remodeled to release calcium to meet the demands for milk production.

The authors employ a series of sophisticated animal models to demonstrate that loss of ER α in ARC neuronal populations leads to the increase of a soluble factor that promotes bone formation. They articulate a novel mechanism to explain maintenance of bone mass through the release of CCN3 as a brain derived factor that induces osteogenesis. The experiments are generally well done and support the proposed model; however, there are several issues that the authors can address to solidify the conclusions they are making and the model that they are proposing.

Main issues:

- 1) The authors do not demonstrate whether circulating levels of CCN3 levels are increased in any of the models in which they have removed ER α in ARC neuronal populations (Fig. 1a: AAV2-Cre delivery ; Fig1b: various Cre drivers coupled with floxed ESR1). Given the authors observations that different levels of CCN3 can either promote (low CCN3 concentrations) or impair (high CCN3 levels), it would be important compare the system concentration of CCN3 achieved in vivo with the concentrations of recombinant mCCN3 used in vitro (Fig. 4).
- 2) Have the authors taken the plasma isolated from the ESR1floxed or Nkx2.1-Cre/ESR1floxed mice (Fig. 4d) and treated with an isotype or CCN3 neutralizing antibody. Would depletion/neutralization of CCN3 impair the elevated bone formation observed in this assay?
- 3) The data in Fig 3g suggests that CCN3 expression emerges when ER α is lost. Have the authors co-stained tissues performed a co-IF with Cre-specific antibodies and CCN3 to show that only the cells expressing Cre (and thus lacking ER α) are the ones that are CCN3 positive. This question is raised due to the fact that in lactating females, there appears to be ARC neurons that co-express ER α and CCN3 (Fig. 5g).
- 4) Do the authors have a mechanism linking the loss of ER α in ARC neurons with elevated CCN3 expression. Is this direct repression of CCN3 via ER α ?
- 5) The data in Fig. 2 indicates that brain derived factors induce the osteogenic capacity of ocSSCs. In Fig. 4, ocSSCs were shown to undergo osteogenic differentiation following exposure to low doses of CCN3. Did the authors mine the scRNAseq data from the ESR1floxed or Nkx2.1-Cre/ESR1floxed

ocSSCs to see if known receptors for CCN3 are differentially expressed? Do the authors know how CCN3 is activating the osteogenic response in ocSSCs (integrin receptors, Notch).

6) The data presented in Fig. 5 represent important efforts of the authors to demonstrate that knockdown of CCN3 in ARC neurons diminishes bone formation (Fig. 5e,f). The correlative data in Fig. 5g also suggests that CCN3 levels do increase in ARC neurons during lactation in wildtype mice. Conditional alleles of CCN3 (floxed CCN3) have been described in the literature (Tu et al. 2023. JCCS, 21:14). Would AAV2-Cre injection into the ARC region of CCN3^{flx/flx} cause deleterious bone loss during lactation (due to loss of a CCN3 mediated osteogenic program)? While the cleanest (and definitive) experiment would be to generate compound mice with CCN3^{flx/flx} and one of the Cre drivers used in Fig. 1b (Nkx2.1-Cre, Kiss-Cre, Pdyn-Cre), this would represent a significant investment by the authors and would not be feasible within the allotted time for revision.

7) The system that was used to engineer CCN3 expression from the liver (Fig. 5a) seems to induce a significant amount of CCN3 positivity. Have the authors quantified the circulating levels of CCN3 achieved in this experiment? This comes back to the idea of CCN3 levels and the impacts that this factor has on bone formation. Indeed, in agreement with the data that the authors show in Fig. 4, there are numerous reports in the literature that high levels of CCN3 are thought to impair bone formation. It would be interesting to compare CCN3 levels in the serum of mice that lack ER α in ARC neurons versus CCN3 levels in the mice described in Fig. 5a.

Minor comments:

The transition to the use of a high fat diet as a means to influence ARC function was rather abrupt. Some additional text explaining why a chronic high fat diet would lead to the modulation of ARC neurons (via what mechanism), and subsequent loss of bone formation in Nkx2.1-Cre/ESR1^{flxed} mice? This transition requires a clearer rationale.

Referee #2 (Remarks to the Author):

This paper builds on previous work from the Ingraham laboratory where they demonstrated that central estrogen signaling exerted an inhibitory effect on bone formation. Using an elegant set of studies, including parabiosis and bone explants, along with extensive in vitro studies, they identify brain-derived CCN3 as a key factor that is regulated by estrogen and drives the increase in bone formation when central estrogen action is inhibited. Their findings have considerable implications for our understanding of the role of the skeleton in maintaining calcium homeostasis during lactation. Moreover, CCN3 may represent a novel therapeutic for osteoporosis, and this certainly warrants further development.

Overall, the studies are exceptionally well done and the results support the conclusions. I do, however, have several issues for the authors to address:

1. Conceptually, the authors ignore the concept of “coupling” between bone resorption and formation. For reviews/perspectives on this, please see PMIDs 31553686, 22354850. The issue here is that when the combination of estrogen deficiency and high PTHrP drive bone resorption during lactation to mobilize calcium from the skeleton for breast milk, bone formation inevitably increases due to this coupling (see PMID: 26887676 for an exhaustive review of skeletal changes during pregnancy and lactation). As noted in the references suggested, this coupling is due both to release of growth factors from the bone matrix as well as osteoclast-derived coupling factors. So CCN3 likely is not solely responsible for the stimulation of bone formation when bone resorption goes up during estrogen deficiency/PTHrP excess following lactation. Rather, given the dual threat of estrogen deficiency and high PTHrP, it is likely that without CCN3, bone loss would be even greater, as this normal coupling mechanism would not be sufficient to prevent marked bone loss. In the Introduction, Discussion, and in Fig. 5i, this concept of physiological coupling that is likely augmented by CCN3 should be incorporated in order to provide a more complete picture.
2. [REDACTED]
3. The high fat diet data are remarkable. Was this just a lucky guess, or were there other clues suggesting that this would unlock the puzzle? Kudos to the authors, as this reviewer would not have come up with this approach.
4. An important question is what happens to CCN3 following ovariectomy? The latter is associated with marked bone loss, so it would be important to know whether CCN3 is upregulated following ovariectomy? If so, why is it not sufficient to prevent bone loss in that setting – admittedly, it may be modulating it, just as in lactation, but this is an important biological question that the authors should address.
5. When the authors refer to Figs 10 e, f – not sure what they mean in terms of “proportional coupling” as osteoclast numbers did not change?
6. Mice in mixed backgrounds were used, which can affect skeletal phenotypes. The authors should comment on what measures (eg, use of littermate controls, etc) were used to limit confounding effects from this.

Referee #3 (Remarks to the Author):

There are lots to like about this manuscript! First of all, the authors addressed a very significant biological question related to female health especially during the lactation period, which has certainly been understudied. Two, the biological mechanisms revealed by this work are conceptually novel, demonstrating an unexpected function of estrogen-ER signaling on bone remodeling and identifying CCN3 as a brain-derived osteogenic factor. Three, the authors assembled an impressive set of experimental approaches, some of which are quite bold, to nicely demonstrate a complicated model. This work, if published, will likely stimulate multiple related research fields, including the brain-body connection, the postpartum biology, the estrogen biology, the bone biology, and the CCN proteins. I have a few comments/suggestions for the authors to address to further improve this work and the manuscript.

1. CCN3 is proposed as a circulating factor, but I did not see any measurement of CCN3 in the circulation of mice or humans, even for the overexpression model with the liver/AAV-dj-CCN3 approach.
2. One of the highlights, to me, is the link of CCN3 in the ARC with the lactation biology, as CCN3 appears in the ARC only during the lactation period. This link would be further strengthened if the authors can show knockout or knockdown of CCN3 affects the bone remodeling in lactating female mice.
3. The in vitro results indicate that CCN3 at different concentrations may cause opposite effects. Fig. 4h used a single i.p. dose of mCCN3 for an in vivo study. How was this dose chosen? Does this dose elevate the local CCN3 level similar to those in the in vitro experiments that showed the osteogenic effect? An in vivo dose response experiment would be helpful.

Other points:

4. The uterine weights were used as an indirect indicator of circulating estrogen levels (ED Fig. 2c). Did the author directly measure estrogen levels?
5. In the ocSSCs scRNAseq data, is there any DEGs related to CCN3 receptor and/or downstream signaling? In addition, Bmp2 does appear to be upregulated in the ocSSCs from the KO mice. Could author discuss this?
6. The results from HFD feeding study are quite striking, and should deserve some more exploration/discussions. For example, did HFD feeding alter the calcium reservoir in the body, or other mechanisms that may regulate bone remodeling?
7. Fig. 3g is missing the male *Esr1^{fl/fl}* group.
8. The units used for bone parameters are not quite consistent throughout the manuscript, e.g. Fig. 4h (using "% Change").

Referee #4 (Remarks to the Author):

This manuscript describes a very important finding, that of brain to bone communication through a factor, CCN3, upon reduction/removal of estrogen to maintain bone mass.

However, there are several concerns:

- 1). What is the rationale for using a high fat diet to regulate factor production in the arcuate nucleus? Are there any references for this?
- 2). Much attention was focused on the effects of CCN3 on osteogenic stem cells. What are the effects on other bone cell types such as osteoblasts, osteoclasts, osteocytes? Whereas TRAP positive osteoclasts were measured as shown in EB Figures 6 and 10, these were quantitated after 17 and 6 weeks respectively which would reflect homeostasis and not early effects on osteoclasts.
- 3). The title states that CCN3 sustains bone in lactating females, however there is no data to support this statement. All that is shown in Figure 5g is the expression levels of CCN3 in the arcuate nucleus in lactating as compared to virgin mice. It would be important to also measure bone mass. Would CCN3 protect against bone loss due to high calcium demand? This would occur with lactating mice with large litter size or on a low calcium diet. I would propose leaving the lactation hypothesis out of the manuscript unless additional experiments can be performed.
- 4). Even though ovariectomy is associated with bone loss, the early effects are increased bone formation (Turner et al, JBMR 1987). Would this effect be observed in CCN3 knockdown/knockout mice?

Author Rebuttals to Initial Comments:

Overall Comments: We thank all referees for their gracious and constructive comments on improving the rigor and impact of our paper. Motivated by their suggestions, we expanded the scope of our study and added key data to significantly strengthen our revised manuscript. Specifically, we now show that **1) CCN3 can be detected in plasma** as a secreted circulatory protein, **2) CCN3 greatly improves bone fracture repair**, and finally, **3) CCN3 expression in the arcuate nucleus (ARC) is essential in lactating mothers** to preserve maternal bone mass and pup viability. Taken together, we believe that this study is paradigm-shifting and will motivate others to examine the bidirectional informational exchange in brain-body conduits. Based on Guillemin and Schally's discovery over 50 years ago, textbooks teach us that brain-derived peptide hormones are routed to the pituitary via the dedicated hypophyseal portal system. We, too, initially subscribed to this dogma and imagined that our brain-derived osteoanabolic hormone must act on bone via the anterior pituitary, even though it is well appreciated that larger peptide hormones such as leptin, prolactin, and now GLP-1 somehow make their way into the brain. As our study now shows, this **brain-body dialog is bidirectional**, and when needed, neurons "step up" to produce circulatory hormones during critical periods in mammalian female physiology. Our work will surely prompt some to ask if the brain controls other adaptive physiological responses through similar humoral mechanisms. Finally, we believe that our focus on unraveling fundamental mechanisms of female physiology creates a **blueprint for unlocking new biology** and potentially new therapeutic targets.

Below, we provide a detailed point-by-point response to each Referee's comments in blue text.

Referee #1: The manuscript by Babey et al. describes a mechanism by which specific neuron populations within the arcuate nucleus (ARC) of the brain, respond to reduced estrogen levels during post-partum lactation by producing CCN3. CCN3 then acts on osteochondral skeletal stem cells within the bone, which induces osteogenesis and new bone formation in female mice. The authors propose that the transient increase in CCN3 levels during lactation is necessary to maintain bone that is being actively being remodeled to release calcium to meet the demands for milk production.

The authors employ a series of sophisticated animal models to demonstrate that loss of ER α in ARC neuronal populations leads to the increase of a soluble factor that promotes bone formation. They articulate a novel mechanism to explain the maintenance of bone mass through the release of CCN3 as a brain-derived factor that induces osteogenesis. The experiments are generally well done and support the proposed model; however, there are several issues that the authors can address to solidify the conclusions they are making and the model that they are proposing.

Main issues:

1. The authors do not demonstrate whether circulating levels of CCN3 levels are increased in any of the models in which they have removed ER α in ARC neuronal populations (Fig. 1a: AAV2-Cre delivery; Fig1b: various Cre drivers coupled with floxed ESR1). Given the authors' observations that different levels of CCN3 can either promote (low CCN3 concentrations) or impair (high CCN3 levels), it would be important compare the system concentration of CCN3 achieved in vivo with the concentrations of recombinant mCCN3 used in vitro (Fig. 4).

This reviewer and others raise a critical question – can we show that CCN3 circulates in the blood? We worked diligently to detect circulating CCN3 from mice transduced with relatively high levels of AAVdj-CCN3; detection was only possible following heparin-agarose purification of CCN3, albeit at very low levels (see Fig 5d). This result underscores the

poor quality of existing anti-CCN3 antibodies, and as expected, we failed to quantify circulating CCN3 levels in mutant females using multiple ELISAs (Abcam, R&D Duo Systems). Further, ELISAs failed to detect CCN3 in plasma or liver extracts from female mice expressing high levels of hepatic CCN3 (**Res. Fig 1A**) or after acute injection of recombinant CCN3 (not shown). Western blotting using commercial anti-mouse CCN3 antibodies tells us why – these antibodies exhibit a high degree of non-specific binding to plasma proteins (**Extended Data Fig 9c**). While efforts to generate suitable anti-CCN3 antibodies are ongoing, as an alternative, we recently constructed an AAVdj-CCN3-3xFLAG version for in vivo expression studies, to correlate relative levels of circulating CCN3 with biological endpoints (**Res. Fig 1B**). In contrast to the poor specificity of anti-CCN3 antibodies, affinity-purified CCN3-3xFLAG from female plasma (transduced with AAVdj-CCN3) yields a single, specific CCN3 band (**mCCN3-3x**). While this reagent allows us to begin probing how efficiently CCN3 is secreted from the ARC (or hepatocytes) in different mouse lines or life stages, our goal to reliably measure endogenous CCN3 requires that we generate reliable anti-CCN3 antibodies or a bioassay. It should also be noted that the single Olink Proteomics detection platform for mice is currently limited to 98 proteins and does not include CCN3.

2. Have the authors taken the plasma isolated from the ESR1 floxed or Nkx2.1-Cre/ESR1 floxed mice (Fig. 4d) and treated with an isotype or CCN3 neutralizing antibody. Would depletion/neutralization of CCN3 impair the elevated bone formation observed in this assay?

We thank the reviewer for suggesting that we test the sufficiency of CCN3 in our whole femur cultures. Unfortunately, depleting CCN3 from mutant female plasma is not possible, given the poor quality of anti-mCCN3 antibodies. As mentioned above, efforts are underway to solve this issue. Heparin agarose

Response Figure 1

purification could be used to diminish CCN3 levels from mutant plasma, but we feel that any results are confounded by this non-selective strategy; in other words, more than CCN3 will bind. As an aside, we have started generating wild-type and inactive variants of human and mouse CCN3 (made in CHO cells) that can be tested in bone cultures.

3) The data in Fig 3g suggests that CCN3 expression emerges when ER α is lost. Have the authors co-stained tissues performed a co-IF with Cre-specific antibodies and CCN3 to show that only the cells expressing Cre (and thus lacking ER α) are the ones that are CCN3 positive. This question is raised due to the fact that in lactating females, there appears to be ARC neurons that co-express ER α and CCN3 (Fig. 5g).

This question as to whether CCN3 and ER α are mutually exclusive or can coexist in the same subset of ARC^{KISS1} neurons is insightful and important. The answer in mutant brains would appear to be NO, but YES in lactating brains. With further analyses of mutant and lactating brains using FISH for *Esr1*, *Ccn3*, and *Kiss1*, we find that *Ccn3* (and CCN3) marks nearly every *Kiss1* neuron in mutant females, consistent with the KISS1 neuronal origin of the high bone mass phenotype. In control females, every *Kiss1* neuron expresses *Esr1* (Fig. 3g,h). In lactating brains, all three markers coexist (Fig 6d). CCN3-positive neurons are notably absent in OVX females (Fig. 6d). These data, coupled with our other genetic KO lines, show that CCN3 is restricted to KISS1-ER α neurons. We posit that deleting ER α from KISS1 neurons in intact females recapitulates the E2-depleted lactational state and primes the upregulation of CCN3. Further, these data imply that secondary events following the loss of ER α -dependent signaling are ultimately needed to drive CCN3 expression in both mutant and lactating females. Until we determine how and what regulates CCN3 in the ARC, it remains possible that two independent mechanisms account for CCN3 expression in the mutant and lactating females - we have added these points to the revised discussion. As discussed below in **Pt 4**, understanding how CCN3 is upregulated in these two settings is a fascinating puzzle to be solved.

4) Do the authors have a mechanism linking the loss of ER α in ARC neurons with elevated CCN3 expression. Is this direct repression of CCN3 via ER α ?

We consider this to be the next central question in this project. Based on new data in lactating females, expression of CCN3 does not rely solely on loss of ER α (Fig 6b), as is the case in *Esr1*^{Nkx2.1-Cre} mutants. Knowing how CCN3 is activated in the ARC of lactating females will surely be of great interest. We have already scoured our list of DEGs in mutant ARC to look at receptor signaling and changes in key transcription factors. Of the latter, we found that NPAS4 is upregulated and NR4A2 is downregulated in the mutant ARC. Neither appears to co-express in ARC^{KISS1} neurons, making it unlikely that they have a direct role in regulating CCN3. We also explored the possibility that VDR might interface with ER α or

directly regulate CCN3, given that HFD feeding diminishes levels of its ligand, 25-hydroxyvitamin D (Belenchia et al., 2017 *J. Comp Med.*), and *Vdr* ARC expression was reported using a genetic VDR-reporter (Sisley, 2021 *J. Comp Neurol*). Unfortunately, in our hands, we could not convince ourselves that *Vdr* is expressed in the ARC, let alone in ARC^{KISS1} neurons. Hormone-dependent control of *Ccn3* is likely but is potentially complex and could easily originate from the many gatekeepers of calcium homeostasis, including bone, gut, mammary gland, and kidney. Finally, we remain open to the possibility that neuronal activity or the status of adjacent tanycytes influence *Ccn3* expression. These possibilities will be addressed in the next steps.

5) The data in Fig. 2 indicates that brain derived factors induce the osteogenic capacity of ocSSCs. In Fig. 4, ocSSCs were shown to undergo osteogenic differentiation following exposure to low doses of CCN3. Did the authors mine the scRNAseq data from the ESR1 floxed or Nkx2.1-Cre/ESR1 floxed ocSSCs to see if known receptors for CCN3 are differentially expressed? Do the authors know how CCN3 activates the osteogenic response in ocSSCs (integrin receptors, Notch).

We, too, would like to identify the CCN3 receptor in SSCs. Although scRNA-Seq of control and mutant SSCs failed to yield viable candidates, we combined these data with our published transcriptomic datasets from purified SSCs (PMID: 34381212, 34280086, 36310174) and mined them to generate a candidate list, yielding scores of candidate receptors (including integrin receptors). Each is now being tested by alpha-fold modeling with the CCN3-predicted monomer and tested using reconstituted cellular systems and viral-mediated knockdowns in cultured human SSCs. Identifying and validating the receptor for CCN3 *in vitro* and *in vivo* will be the subject of our next study.

6) The data presented in Fig. 5 represent important efforts of the authors to demonstrate that knockdown of CCN3 in ARC neurons diminishes bone formation (Fig. 5e,f). The correlative data in Fig. 5g also suggests that CCN3 levels do increase in ARC neurons during lactation in wildtype mice. Conditional alleles of CCN3 (floxed CCN3) have been described in the literature (Tu et al. 2023. *JCCS*, 21:14). Would AAV2-Cre injection into the ARC region of CCN3^{flx/flx} cause deleterious bone loss during lactation (due to loss of a CCN3 mediated osteogenic program)? While the cleanest (and definitive) experiment would be to generate compound mice with CCN3^{flx/flx} and one of the Cre drivers used in Fig. 1b (Nkx2.1-Cre, Kiss-Cre, Pdyn-Cre), this would represent a significant investment by the authors and would not be feasible within the allotted time for revision.

We appreciate the suggestion to use the floxed allele of *Ccn3* to eliminate expression in the ARC. However, as correctly pointed out by this Reviewer, this strategy would significantly extend the timeframe of our study. Unfortunately, we are still awaiting shipment of this line due to MTA issues despite our initial request over six months ago. In response to this suggestion, viral vectors were delivered before pregnancy by

stereotaxic surgery, to blunt subsequent ARC CCN3 expression during lactation (Fig 6e). In this cohort, we achieved 50-80% knockdown of CCN3 in the ARC. Even without a complete knockout of CCN3, the results are dramatic. While sh-m*Ccn3* in the ARC does not affect fertility (time to plug), fecundity (litter size), or milk provision (Fig 6f and Extended Data Fig. 10a-d), these dams experience a significant decline in bone mass on chow (0.8% Ca²⁺). When lactating shRNA-m*Ccn3* mothers were challenged postpartum with a low calcium diet (0.01% Ca²⁺), moms could not sustain their progeny (Fig 6h,i). This drop in viability was entirely dependent on the status of brain CCN3 in mothers as transfer of pups to an shRNA-m*Ccn3* mother resulted in 10% weight loss compared to 30% weight gain when nursed by control (shCtrl) dams (Extended Data Fig 10e). From an evolutionary viewpoint, we wonder if these conditions represent a turning point for mothers, whereby the energy to maintain progeny is sacrificed, and self-preservation is prioritized for the future when conditions improve.

7) The system that was used to engineer CCN3 expression from the liver (Fig. 5a) seems to induce a significant amount of CCN3 positivity. Have the authors quantified the circulating levels of CCN3 achieved in this experiment? This comes back to the idea of CCN3 levels and the impacts that this factor has on bone formation. Indeed, in agreement with the data that the authors show in Fig. 4, there are numerous reports in the literature that high levels of CCN3 are thought to impair bone formation. It would be interesting to compare CCN3 levels in the serum of mice that lack ER α in ARC neurons versus CCN3 levels in the mice described in Fig. 5a.

As we now show, plasma CCN3 was readily detected only after heparin-agarose plasma purification from mice harboring relatively high hepatic CCN3 expression. The low sensitivity of existing anti-mCCN3 antibodies makes it difficult to quantify circulating levels of CCN3 accurately (Fig 5d and Extended Data Fig. 9c), raising the possibility that this is a sensitivity issue with detection or that secretion from hepatocytes is less efficient. As mentioned above, in vivo dosing of CCN3-FLAG can circumvent this issue by determining the optimal dose of mouse / human CCN3 to maximize bone formation.

Minor comments:

8) The transition to the use of a high fat diet as a means to influence ARC function was rather abrupt. Some additional text explaining why a chronic high fat diet would lead to the modulation of ARC neurons (via what mechanism), and subsequent loss of bone formation in Nkx2.1-Cre/ESR1 floxed mice? This transition requires a clearer rationale.

We appreciate this comment, which was raised by all Reviewers. Part of our rationale was based on the idea that a chronic high-fat diet might modulate ARC neurons (Quennell et al., 2011, *Endocrinology*) and, therefore, alter their function and perhaps the bone phenotype. Low-grade inflammation due to HFD challenge could further impair the function of these neurons. We now include new text providing the

rationalization for this dietary challenge. We also wondered if a prolonged HFD challenge would skew stem cell niches away from bone formation (osteogenesis) toward fat accumulation in the *Esr1*^{Nkx2.1-Cre} mutants, which we knew to be extremely low. Surprisingly, bone mass and fat accumulation are uncoupled in this setting. Mutant bones still show sparse BMAT, while also retaining much of their strength (Fig 3a,b). Understanding how this phenomenon occurs in mutant females will be pursued in a follow-up study. Finally, as often occurs in the scientific process, the art of “just experimenting” often leads to the unexpected, as was the case here.

Referee #2:

This paper builds on previous work from the Ingraham laboratory where they demonstrated that central estrogen signaling exerted an inhibitory effect on bone formation. Using an elegant set of studies, including parabiosis and bone explants, along with extensive in vitro studies, they identify brain-derived CCN3 as a key factor that is regulated by estrogen and drives the increase in bone formation when central estrogen action is inhibited. Their findings have considerable implications for our understanding of the role of the skeleton in maintaining calcium homeostasis during lactation. Moreover, CCN3 may represent a novel therapeutic for osteoporosis, and this certainly warrants further development. Overall, the studies are exceptionally well done and the results support the conclusions. I do, however, have several issues for the authors to address:

1. Conceptually, the authors ignore the concept of “coupling” between bone resorption and formation. For reviews/perspectives on this, please see PMIDs 31553686, 22354850. The issue here is that when the combination of estrogen deficiency and high PTHrP drive bone resorption during lactation to mobilize calcium from the skeleton for breast milk, bone formation inevitably increases due to this coupling (see PMID: 26887676 for an exhaustive review of skeletal changes during pregnancy and lactation). As noted in the references suggested, this coupling is due both to release of growth factors from the bone matrix as well as osteoclast-derived coupling factors. So CCN3 likely is not solely responsible for the stimulation of bone formation when bone resorption goes up during estrogen deficiency/PTHrP excess following lactation. Rather, given the dual threat of estrogen deficiency and high PTHrP, it is likely that without CCN3, bone loss would be even greater, as this normal coupling mechanism would not be sufficient to prevent marked bone loss. In the Introduction, Discussion, and in Fig. 5i, this concept of physiological coupling that is likely augmented by CCN3 should be incorporated in order to provide a more complete picture.

We appreciate the reviewer’s request to address the physiological coupling concept in our revised text. This has now been added to our introduction and discussion with the suggested review reference by Sims and Martin 2020. Without CCN3 in the ARC, the usual mechanisms that maintain healthy coupling of bone formation and bone loss during lactation are insufficient to counteract high PTHrP and low E2. Thus, as the reviewer suggested, without CCN3 on board, bone loss is even more significant during lactation (Fig

6g). We have modified the text and discussion to clarify that CCN3 might be one of several mechanisms at play to preserve healthy coupling in lactating females. We also detect no differences in osteoclast number/bone surface, lacunar density/bone area (Fig 5j), or osteocytic osteolysis by TRAP staining (Extended Fig 9f,i). Further, osteoclastogenesis was unchanged after CCN3 treatment based on in vitro culturing of bone marrow monocytes harvested from 3-month-old or 24-month-old male mice. These data suggest that the primary effect of CCN3 is on osteochondral skeletal stem cells, osteoblasts, and bone formation. We believe that our new fracture repair data with a slow-release CCN3 gel (Fig 4i,j) provides additional evidence that CCN3 can improve fracture repair and promote healthy remodeling.

2. In terms of lactation, trabecular bone is primarily mobilized. The spine (rather than metaphyses of long bones) are the main sites for trabecular bone. However, the paper provides data exclusively on trabecular bone at the femur. Was the spine analyzed in any of the key experiments – especially when CCN3 is administered, elevated, or knocked down? This is important from a biological and potentially therapeutic standpoint (eg, if CCN3 had no effect on the spine, it would suggest that the biological story presented here is incomplete and would likely not be a useful therapeutic agent).

We agree that obtaining data from the trabecular-rich vertebrae is important. Our revised study shows a ~15-30% percent increase in the L5 %BV/TV in intact females, intact males, and OVX females after ectopic expression of CCN3. We would also note that *Esr1*^{Nkx2.1-Cre} mutant females exhibited a significant increase in L5 bone mass, as shown in the image to the right (taken from Fig 2g Herber, Krause, et al., 2019 *Nat Comm*). Given that the trabecular bone of the tibiae and femora undergoes significant loss during lactation (Lyu et al., 2012 *JBMR*), we would argue that assessing trabecular bone in the femur likely mimics the lactational effects on the spine.

3. The high-fat diet data are remarkable. Was this just a lucky guess, or were there other clues suggesting that this would unlock the puzzle? Kudos to the authors, as this reviewer would not have come up with this approach.

Please see our response above to **Rev 1 (Pt 8)**.

4. An important question is what happens to CCN3 following ovariectomy? The latter is associated with marked bone loss, so it would be important to know whether CCN3 is upregulated following ovariectomy? If so, why is it not sufficient to prevent bone loss in that setting – admittedly, it may be modulating it, just as in lactation, but this is an important biological question that the authors should address.

Thank you for raising this important question regarding the brain expression of CCN3 following ovariectomy. As we show in Fig 6d, CCN3-positive neurons reside in close proximity to tanycytes lining the 3rd ventricle in both mutant and lactating females but are notably absent in OVX females at one and two weeks post-surgery. This suggests that estrogen depletion with OVX is insufficient to induce CCN3 in $ARC^{ERa/Kiss1}$ neurons by itself, failing to mimic the loss of estrogen signaling in mutant mice or estrogen depletion in lactating mice.

5. When the authors refer to Figs 10 e, f – not sure what they mean in terms of “proportional coupling” as osteoclast numbers did not change?

We thank the reviewer for this suggestion. We have revised the text and removed the phrase "proportional coupling" from the text.

6. Mice in mixed backgrounds were used, which can affect skeletal phenotypes. The authors should comment on what measures (eg, use of littermate controls, etc) were used to limit confounding effects from this.

In all experiments, except where noted, $Esr1^{fl/fl}$ littermate controls were used to limit confounding effects from mice with mixed backgrounds. In three instances, we used aged C57BL/6 mice obtained from NIA – 1) in the fracture repair studies, as noted in the main figure (Fig 4i,j), 2) in whole bone culturing of aged female femurs (Extended Data Fig 8e), and 3) in the osteoclast differentiation assays (Extended Data Fig 9f,i).

Referee #3:

There are lots to like about this manuscript! First of all, the authors addressed a very significant biological question related to female health especially during the lactation period, which has certainly been understudied. Two, the biological mechanisms revealed by this work are conceptually novel, demonstrating an unexpected function of estrogen-ER signaling on bone remodeling and identifying CCN3 as a brain-derived osteogenic factor. Three, the authors assembled an impressive set of experimental approaches, some of which are quite bold, to nicely demonstrate a complicated model. This work, if published, will likely stimulate multiple related research fields, including the brain-body connection, the postpartum biology, the estrogen biology, the bone biology, and the CCN proteins. I have a few comments/suggestions for the authors to address to further improve this work and the manuscript.

1. CCN3 is proposed as a circulating factor, but I did not see any measurement of CCN3 in the circulation of mice or humans, even for the overexpression model with the liver/AAV-dj-CCN3 approach.

Please see our responses above to Rev 1, Pt1.

2. One of the highlights, to me, is the link of CCN3 in the ARC with the lactation biology, as CCN3 appears in the ARC only during the lactation period. This link would be further strengthened if the authors can show knockout or knockdown of CCN3 affects the bone remodeling in lactating female mice.

We thank the Reviewer for this comment and are also excited to link CCN3 in the ARC with lactation biology, building on earlier seminal work by Bonewald, Kovacs, Wysolmerski, Kronenberg, and others. As mentioned above, in response to **Rev 1, Pt 6**, the role of CCN3 as an anabolic brain hormone during lactation was shown after knockdown of CCN3 in adult virgin female ARCs, before pregnancy (**Fig 6**). While much more needs to be done in this space, our study and others should motivate further investigation into the adaptive responses mounted by females during this distinct life stage.

3. The in vitro results indicate that CCN3 at different concentrations may cause opposite effects. Fig. 4h used a single i.p. dose of mCCN3 for an in vivo study. How was this dose chosen? Does this dose elevate the local CCN3 level similar to those in the in vitro experiments that showed the osteogenic effect? An in vivo dose response experiment would be helpful.

As this Reviewer notes, the dosing of CCN3 remains to be thoroughly investigated, which we hope to do in multiple contexts over the next year. Doses for chronic i.p. injections into males and females were chosen based on our in vitro studies and the literature (Marchal et al., 2015 *PLoS One*). As mentioned above, until we develop a reliable and accurate ELISA for CCN3 levels and know that recombinant CCN3 faithfully recapitulates endogenously expressed CCN3 (PTMs), these concentrations are entirely empirical. We did detect a dose-dependent increase in the callus bone volume and strength in the stabilized fracture model carried out in aged male mice (**Fig 4i,j**). We also performed limited in vivo dose-response testing after ectopically expressing CCN3 in the liver. Increased bone mass and bone strength in young females and males were apparent at intermediate levels of CCN3 (3×10^{11} GC/ml) (**Fig. 5e,f**). Higher CCN3 doses (15×10^{11}) were similarly effective (**Extended Data Fig. 9e**). Even at exceedingly low levels of hepatic CCN3 expression, modest increases in bone formation were observed (**Extended Data Fig. 9d**).

Other points:

4. The uterine weights were used as an indirect indicator of circulating estrogen levels (ED Fig. 2c). Did the author directly measure estrogen levels?

In our previous publication (Herber, Krause, et al., 2019 *Nat Comm*), we detected no significant changes in serum sex steroids (E2, T) in younger (4-5 wks) *Esr1*^{Nkx2.1-Cre} and older (24 wks) females acutely deleted for *Esr1* by stereotaxic surgery, please refer to **Figure 2 (Link)** and **Extended Data Fig 4c, 6e (Link)**. While this seemed counterintuitive at the time, in retrospect, this is consistent with the induction of CCN3 in the ARC during lactation when estrogen is depleted.

5. In the ocSSCs scRNAseq data, is there any DEGs related to CCN3 receptor and/or downstream signaling?

Please see our Response above to **Rev 1, Pt 5**.

In addition, *Bmp2* does appear to be upregulated in the ocSSCs from the KO mice. Could author discuss this?

Yes, we do find *Bmp2* upregulated in ocSSCs harvested from mutant females. We hypothesize that this is likely a consequence of their activation (through CCN3). As SSCs have autocrine mechanisms to amplify their activity, i.e., bone anabolic/pro-regenerative response, our previous studies show that *Bmp2* is sufficient to reactivate quiescent dysfunctional and aged SSCs (Ambrosi et al. Nature 2021, Murphy et al. 2020 *Nat. Medicine*).

6. The results from HFD feeding study are quite striking, and should deserve some more exploration/discussions. For example, did HFD feeding alter the calcium reservoir in the body, or other mechanisms that may regulate bone remodeling?

Please see our response above to **Rev 1, Pt 8**.

7. Fig. 3g is missing the male *Esr1*^{fl/fl} group.

As requested, and now shown in **Extended Data Fig 7e**, CCN3 is not present in intact control male *Esr1*^{fl/fl} mice. Moreover, to date, we have found no condition in the *Esr1*^{Nkx2.1-Cre} mutant males when CCN3 is expressed, having looked at both young pre-pubertal and aged males.

8. The units used for bone parameters are not quite consistent throughout the manuscript, e.g. Fig. 4h (using “% Change”).

For all figures except for panels in Figure 4, we display fractional bone volume as %BV/TV. In the far right-hand panels in **Fig 4d, f**, the absolute bone volumes in the left-hand panels are replotted for paired femurs as a percent change of the contralateral femur with CCN3. In doing so, we wanted to show the effects of CCN3 compared to control plasma or baseline when bones were not cultured (baseline). As mentioned in the text, culturing long bones for five days leads to significant bone loss, as shown in **Extended Fig 8d, e**. For i.p. chronic injections of recombinant mCCN3, we used different cohorts of littermates done two months apart to increase our numbers. As often occurs, the baseline %BV/TV for these cohorts was statistically different ~9% versus 12% BV/TV. When coupled with the relatively modest (for females) or variable (for males) effects using this delivery, we chose to show these data a percent change from the %BV/TV baseline. These results prompted us to seek a more reliable method of CCN3 treatment – hence, we chose to pursue the viral vector delivery system in the liver. This method is less invasive than daily

injections and yields far more consistent and robust results, as shown in Fig 5. In the future, this delivery method will allow us to test variants of CCN3 in vivo quickly.

Referee #4:

This manuscript describes a very important finding, that of brain to bone communication through a factor, CCN3, upon reduction/removal of estrogen to maintain bone mass. However, there are several concerns:

1). What is the rationale for using a high-fat diet to regulate factor production in the arcuate nucleus? Are there any references for this?

Please see our response to **Rev 1, Pt 8**.

2). Much attention was focused on the effects of CCN3 on osteogenic stem cells. What are the effects on other bone cell types such as osteoblasts, osteoclasts, osteocytes? Whereas TRAP positive osteoclasts were measured as shown in EB Figures 6 and 10, these were quantitated after 17 and 6 weeks respectively which would reflect homeostasis and not early effects on osteoclasts.

Please see our Response to **Rev 2, Pt 1**.

3). The title states that CCN3 sustains bone in lactating females, however there is no data to support this statement. All that is shown in Figure 5g is the expression levels of CCN3 in the arcuate nucleus in lactating as compared to virgin mice. It would be important to also measure bone mass. Would CCN3 protect against bone loss due to high calcium demand? This would occur with lactating mice with large litter size or on a low calcium diet. I would propose leaving the lactation hypothesis out of the manuscript unless additional experiments can be performed.

The Reviewer is correct in asking for more solid data to support a role for CCN3 during lactation other than simply showing expression. As outlined above in response to **Rev 1, Pt 6**, our new data shown in Fig 6 offers a high level of confidence in the role of CCN3 during lactation. Our study supports a unique role for ARC^{ERa-KISS1} neurons in female physiology by meeting the demands of lactation on the maternal skeleton and ensuring progeny survival by producing a novel lactation-induced brain hormone (LIBH or CCN3) (Fig 6j).

4). Even though ovariectomy is associated with bone loss, the early effects are increased bone formation (Turner et al, JBMR 1987). Would this effect be observed in CCN3 knockdown/knockout mice?

We thank the referee for pointing out this study, which shows that the rate of bone formation increases in ovariectomized rats, consistent with high bone turnover state in early post-menopausal women (Fink et al., 2000 *Osteoporosis Int.*, Recker et al., 2018 *Bone*). At no time point does CCN3 appear in ARC neurons of OVX females at 1 week (Fig 5g) and 2 (data not shown) post-surgery. Our finding suggests that, at least in mice, brain-derived CCN3 does not account for the findings of this study.

Reviewer Reports on the First Version:

Referees' comments:

Referee #1 (Remarks to the Author):

The authors of the manuscript entitled "CCN3, A Lactation-Induced Brain Hormone That Builds Bone and Sustains Progeny in Mice" have provided a thorough and thoughtful response to the first round of reviews and have resubmitted a revised manuscript that is improved from the initial submission. This reviewer appreciates the attempts made by the authors to demonstrate that CCN3 can be detected in circulation, which is hampered by the existing quality of anti-CCN3 antibodies (issues with non-specific binding to plasma proteins). New data (Fig. 5d) demonstrates that CCN3 can be detected in the plasma (following heparin-agarose purification) when ectopically expressed at high levels in the liver. The authors describe their efforts to import mice harboring floxed CCN3, which are ongoing and are currently not available to address comments raised during the initial review. To provide evidence that CCN3 produced by ARC neurons modulates bone density, the authors have partially knocked down CCN3 through stereotactic delivery of CCN3 shRNAs in the brain. The new data included in Fig. 6(e-i) and Fig. S10 demonstrate that a 50-80% reduction of CC3 in ARC neurons correlated with a reduction in bone mass. In addition, loss of CCN3 in the brain caused a reduction in the weight of pups nursed by mothers on a low calcium diet. These data provide important evidence supporting the model proposed by the authors. The remaining points have been addressed or reasonably discussed and I support publication of the revised manuscript.

Referee #2 (Remarks to the Author):

The authors have done a nice job addressing my concerns. Overall, this is an outstanding contribution breaking new ground in our understanding of brain-bone cross-talk.

Sundeep Khosla

Referee #3 (Remarks to the Author):

The authors have adequately addressed my comments. I support the publication of this exciting work in Nature.

Referee #4 (Remarks to the Author):

The authors have responded adequately to my comments. They have added an exciting experiment showing that knockdown of CCN3 in the ARC has dramatic effects on lactation. Impressive findings.

It is unfortunate that a specific antibody to CCN3 has not been identified, but this is frequently the case for poorly studied or highly homologous proteins. Non-specificity of commercial antibodies is a serious issue, and assumption of such specificity should never be made. The authors have performed due diligence. I think the paper should be published without this information.

I will not list all the points listed above as I feel these have been adequately addressed.